SciPost Physics Lecture Notes 

# Quantum Thermodynamics

**Patrick P. Potts**

Department of Physics and Swiss Nanoscience Institute, University of Basel,
Klingelbergstrasse 82, 4056 Basel, Switzerland

patrick.potts@unibas.ch

## Abstract

The theory of quantum thermodynamics investigates how the concepts of heat, work, and temperature can be carried over to the quantum realm, where fluctuations and randomness are fundamentally unavoidable. These lecture notes provide an introduction to the thermodynamics of small quantum systems. It is illustrated how the laws of thermodynamics emerge from quantum theory and how open quantum systems can be modeled by Markovian master equations. Quantum systems that are designed to perform a certain task, such as cooling or generating entanglement are considered. Finally, the effect of fluctuations on the thermodynamic description is discussed.

# 1   Introduction

By investigating concepts such as heat, work, and temperature, the theory of thermodynamics was a driving force in the industrial revolution, enabling the improvement and development of technologies that reshaped society such as steam engines and refrigerators [1]. Quantum thermodynamics investigates heat, work, temperature, as well as related concepts in quantum systems. In analogy to macroscopic thermodynamics, this endeavor promises to enable the development and improvement of novel quantum- and nano-technologies. Furthermore, a good understanding of the thermodynamics of quantum systems is key for keeping the energetic footprint of any scalable quantum technology in check [2].

As the concepts investigated by quantum thermodynamics are very general, the theory is of relevance for essentially all scenarios involving open quantum systems. This makes the field extremely broad, including well established topics such as thermoelectricity [3,4], investigating how an electrical current arises from a temperature gradient, as well as very recent investigations into the energetics of single qubit gates [5]. The broad nature of the field implies that the quantum thermodynamics community brings together people with very different backgrounds.

These lecture notes are meant to provide an introduction to this diverse and fast-growing field, discussing fundamental concepts and illustrating them with simple examples. After a short introduction to basic concepts, macroscopic thermodynamics, and information theory will be reviewed in Sec. 1. Section 2 will address the topic of thermodynamic equilibrium. This will allow for understanding why a system may often be described by very few parameters, such as temperature and chemical potential, and it will give you a physical understanding of these fundamental parameters. Section 3 will then discuss how the laws of thermodynamics emerge from quantum mechanics, connecting these very general laws to the microscopic behavior of quantum systems. Applying the laws of thermodynamics to quantum systems will allow you to make general predictions on what happens when small systems are placed into contact with thermal reservoirs. This may improve your physical intuition for how systems in and out of equilibrium behave, which is an extremely valuable skill as a physicist. In Sec. 4, we will study quantum master equations as a tool to describe open quantum systems. This is an extremely useful and versatile tool that can be employed to model many physical systems, e.g., quantum dots, NV centers, and optical cavities. Section 5 will then investigate quantum thermal machines, exploring some of the tasks that can be achieved with open quantum systems. Such tasks include the production of work, refrigeration, and the creation of entanglement. Finally, in Sec. 6, we will consider fluctuations which become important in nano-scale systems. You will learn how thermodynamic considerations may result in equalities and inequalities that can be understood as extensions of the laws of thermodynamics, determining the behavior of fluctuations around average values.

**Further reading:**    In 2022, a textbook on quantum stochastic thermodynamics by Strasberg was published [6]. While similar in spirit to the material discussed in this course, the book has a stronger focus on fluctuations which are only covered in Sec. 6 in this course. Another good resource is the book published in 2019 by Deffner and Campbell [7]. Compared to these lecture notes, it has a stronger focus on information-theoretic concepts and thermodynamic cycles. A slightly older textbook on the topic was published in 2009 by Gemmer, Michel, and Mahler [8]. While being a valuable resource for many of the concepts, it predates a number of important works that are central to how the topic is presented here. In addition, in 2019 a book giving a snapshot of the field was published, providing numerous short reviews on different topics by over 100 contributing authors [9]. Furthermore, a number of excellent reviews focusing on different aspects of quantum thermodynamics are provided by Refs. [3, 4, 10–14]. These resources are complemented by the focus issue in Ref. [15].

## 1.1 Basic concepts

In this section, we introduce some basic concepts that are used throughout the lecture notes. We set $\hbar = 1$ throughout. Note that this section should mainly act as a reminder (with the possible exception of Sec. 1.3) and to set the notation. If a concept is completely new to you, I suggest to read up on it in the respective references, which provide a far more detailed introduction.

### 1.1.1 Linear algebra

Because linear algebra provides the mathematical description of quantum mechanics, we review some basic concepts here. In a format accessible for physicists, a more detailed introduction can be found in most textbooks on quantum mechanics, see, e.g., Sec. 1 in the book by Shankar [16] or Sec. 2.1 in the book by Nielsen and Chuang [17].

**Vectors in Hilbert spaces:**    A (pure) quantum state can be described by a vector in a complex Hilbert space $\mathcal{H}$, which is a vector space equipped with an inner product. Such vectors may be denoted by

$$|\psi\rangle, \quad |\varphi\rangle, \quad a|\psi\rangle + b|\varphi\rangle, \tag{1}$$

where $a$ and $b$ are complex numbers. For reasons that will become clear later, we also call a vector a *ket*. Any vector can be written using a set of linearly independent basis vectors $|j\rangle$ as

$$|\psi\rangle = \sum_j \psi_j |j\rangle. \tag{2}$$

**Linear operators:**    Vectors can be manipulated by operators. A linear operator $\hat{A} : \mathcal{H} \to \mathcal{H}'$ maps a vector from a Hilbert space, $\mathcal{H}$, onto another vector, potentially in another Hilbert space $\mathcal{H}'$, i.e., $\hat{A}|\psi\rangle$ is a vector in $\mathcal{H}'$ for any $|\psi\rangle \in \mathcal{H}$. Furthermore, a linear operator obeys

$$\hat{A}(a|\psi\rangle + b|\varphi\rangle) = a\hat{A}|\psi\rangle + b\hat{A}|\varphi\rangle. \tag{3}$$

**Dual vectors:**    For each vector, a dual vector (an element of the dual space $\mathcal{H}^*$) exists given by the Hermitian conjugate of the original vector

$$\langle\psi| = |\psi\rangle^\dagger. \tag{4}$$

A dual vector is also called a *bra*. The notation introduced in Eqs. (1) and (4) is called Dirac or bra-ket notation.

**Inner product:**  The inner product between two vectors $|\psi\rangle$ and $|\varphi\rangle$ is denoted by

$$\langle\psi|\varphi\rangle \in \mathbb{C}, \tag{5}$$

and is a complex number. Note that this makes any dual vector a linear operator mapping vectors onto the Hilbert space of complex numbers, i.e., $\langle\psi| : \mathcal{H} \to \mathbb{C}$. With the inner product, we may define an orthonormal basis $|j\rangle$ as one fulfilling

$$\langle i|j\rangle = \delta_{i,j}, \tag{6}$$

where $\delta_{i,j}$ denotes the Kronecker delta. Using Eq. (2), we may express any vector through an orthonormal basis. The inner product between two vectors may then be evaluated as

$$\langle\psi|\varphi\rangle = \left(\sum_j \psi_j^* \langle j|\right)\left(\sum_k \varphi_k |k\rangle\right) = \sum_j \psi_j^* \varphi_j. \tag{7}$$

**Composite systems:**  When describing a composite system that is made of multiple subsystems, a tensor product space can be used. In the case of two subsystems (a bi-partite system), the total Hilbert space is then given by $\mathcal{H} = \mathcal{H}_A \otimes \mathcal{H}_B$, where $\mathcal{H}_A$ and $\mathcal{H}_B$ are the Hilbert spaces describing subsystems A and B, and $\otimes$ denotes the Kronecker product.

An orthonormal basis can be constructed from orthonormal bases of the subsystems as

$$|a, b\rangle = |a\rangle \otimes |b\rangle, \tag{8}$$

where $|a\rangle$ and $|b\rangle$ are the basis states of the subsystems.

Sometimes it is necessary to discard one of the subsystems. This is achieved through the partial trace. For an operator acting on the composite system $\hat{C} : \mathcal{H} \to \mathcal{H}$, the partial trace over the subsystem B is defined as

$$\text{Tr}_B\{\hat{C}\} = \sum_{a,a',b} |a\rangle\langle a'|\langle a, b|\hat{C}|a', b\rangle, \tag{9}$$

which is an operator that acts on subsystem A. In particular, if we consider a pure quantum state describing the composite system $|\Psi\rangle$, then the partial trace $\text{Tr}_B\{|\Psi\rangle\langle\Psi|\}$ is the reduced (possibly mixed) state of subsystem A. The reduced state fully describes subsystem A if no information on subsystem B can be obtained.

**Row and column vectors:**  Every complex vector space of finite dimension $n$ is isomorphic to $\mathbb{C}^n$. What this means is that as long as we consider finite-dimensional Hilbert spaces, we may use conventional row and column vectors. For instance, a set of orthonormal basis vectors may be identified by column vectors with the $j$-th entry equal to one and all others equal to zero. General vectors can then be expressed through Eq. (2), i.e.,

$$|j\rangle \equiv \begin{pmatrix} \vdots \\ 0 \\ 1 \\ 0 \\ \vdots \end{pmatrix}, \quad \langle j| \equiv \begin{pmatrix} \cdots & 0 & 1 & 0 & \cdots \end{pmatrix}, \quad \Rightarrow \quad |\psi\rangle \equiv \begin{pmatrix} \psi_0 \\ \psi_1 \\ \psi_2 \\ \vdots \end{pmatrix}, \quad \langle\psi| \equiv \begin{pmatrix} \psi_0^* & \psi_1^* & \psi_2^* & \cdots \end{pmatrix}. \tag{10}$$

**Matrices and the resolution of the identity:**   An important operator is the identity operator $\mathbb{1}$ defined by

$$\mathbb{1}\,|\psi\rangle = |\psi\rangle \qquad \forall \;\; |\psi\rangle\,. \tag{11}$$

Using any orthonormal basis, the identity may be written

$$\mathbb{1} = \sum_j |j\rangle\langle j|\,. \tag{12}$$

This equation is called the *resolution of the identity* and it is heavily used in deriving various results in quantum mechanics. With its help, we may express an operator in the basis of $|j\rangle$ as

$$\hat{A} = \sum_{j,k} |j\rangle\langle j|\hat{A}|k\rangle\langle k| = \sum_{j,k} A_{jk}\,|j\rangle\langle k|\,, \tag{13}$$

where $A_{jk} = \langle j|\hat{A}|k\rangle$ are the matrix elements of $\hat{A}$. Indeed, with the help of the row and column vectors in Eq. (10), we may identify

$$\hat{A} \equiv \begin{pmatrix} A_{00} & A_{01} & \cdots \\ A_{10} & A_{11} & \\ \vdots & & \ddots \end{pmatrix}, \qquad \hat{A}^\dagger \equiv \begin{pmatrix} A_{00}^* & A_{10}^* & \cdots \\ A_{01}^* & A_{11}^* & \\ \vdots & & \ddots \end{pmatrix}, \tag{14}$$

and we recover the usual prescription for the Hermitian conjugate of matrices.

**Superoperators:**   We may consider operators as vectors in a different vector space, the so-called Liouville space. The Hilbert-Schmidt inner product between two operators $\hat{A}$ and $\hat{B}$ is defined as $\mathbf{Tr}\{\hat{A}^\dagger\hat{B}\}$. With this inner product, Liouville space is itself a Hilbert space. An operator that acts on a vector in Liouville space is called a superoperator to distinguish it from operators that act on states $|\psi\rangle$. Superoperators thus act on operators the same way as operators act on states. Throughout these lecture notes, superoperators are denoted by calligraphic symbols, e.g., $\mathcal{L}$, while operators can be identified by their hat, e.g., $\hat{A}$ (with a few exceptions such as the identity operator $\mathbb{1}$).

As discussed above, we may write a vector in Liouville space as a column vector. Starting from the matrix representation of an operator $\hat{A}$, c.f. Eq. (14), this can be achieved by stacking its columns. Any superoperator may then be written as a matrix. This procedure is called vectorization (see Ref. [18] for more details) and it can be highly useful when using numerics to compute the behavior of open quantum systems.

### 1.1.2   The density matrix

The density matrix describes the state of a quantum system and is thus a central object throughout these lecture notes and in quantum theory in general. A more detailed introduction can be found in the book by Nielsen and Chuang [17], see Sec. 2.4.

The state of a quantum system is described by a positive semi-definite operator with unit trace acting on the Hilbert space $\mathcal{H}$. Positive semi-definite means that all eigenvalues of the density matrix are larger or equal to zero and unit trace means that the eigenvalues add up to one. The density matrix may be written as

$$\hat{\rho} = \sum_j p_j\,|\psi_j\rangle\langle\psi_j|\,, \qquad \text{with} \quad \langle\psi_j|\psi_j\rangle = 1, \quad p_j \geq 0, \quad \sum_j p_j = 1. \tag{15}$$

We may interpret the density matrix as the system being in the pure state $|\psi_j\rangle$ with probability $p_j$. If multiple $p_j$ are non-zero, this describes a scenario of incomplete knowledge.

**Measurements and expectation values:**   In quantum mechanics, any measurement can be described using a positive operator valued measure (POVM) [17]. A POVM is a set of positive, semi-definite operators $\hat{M}_j$ that obey $\sum_j M_j = \mathbb{1}$. The index $j$ labels the outcome of the measurement and it is obtained with probability $p_j = \text{Tr}\{\hat{M}_j \hat{\rho}\}$ for a quantum state $\hat{\rho}$. Note that the positive semi-definiteness ensures $p_j \geq 0$, since $\langle \psi | \hat{M}_j | \psi \rangle \geq 0$ for any $|\psi\rangle$. The fact that the $\hat{M}_j$ sum to the identity ensures that $\sum_j p_j = 1$.

Of particular interest are projective measurements, where $\hat{M}_j^2 = \hat{M}_j$, i.e., the operators $\hat{M}_j$ are projectors. We may create a projective measurement from any operator $\hat{A} = \sum_j a_j |j\rangle\langle j|$, by using the eigenbasis of the operator to define $\hat{M}_j = |j\rangle\langle j|$. In this case, we say that a projective measurement of the operator $\hat{A}$ gives the value $a_j$ with probability $p_j = \langle j | \hat{\rho} | j \rangle$. The average value of the measurement outcomes is given by

$$\langle \hat{A} \rangle \equiv \text{Tr}\{\hat{A}\hat{\rho}\} = \sum_j a_j p_j. \tag{16}$$

We also call this the average value of $\hat{A}$. After a projective measurement with outcome $a_j$, the system is collapsed into the state $|j\rangle\langle j|$. Repeating the same projective measurement twice thus necessarily gives the same outcome. While projective measurements are widely used in quantum theory, they represent an idealization that strictly speaking cannot be implemented in the laboratory as they would violate the third law of thermodynamics [19].

**Classical mixture vs quantum superposition:**   Let us consider the toss of a coin, where we identify with $|0\rangle$ the outcome tails and with $|1\rangle$ the outcome heads. The outcome of such a coin toss (before observation) is described by the density matrix

$$\hat{\rho} = \frac{1}{2}\left(|0\rangle\langle 0| + |1\rangle\langle 1|\right) = \mathbb{1}, \tag{17}$$

which is equal to the identity matrix, also called the maximally mixed state because each outcome is equally likely. Such a mixture of states is completely classical and merely reflects a scenario of incomplete knowledge, in this case the outcome of the coin toss.

We now compare the state in Eq. (17) to a quantum superposition provided by the pure state $|+\rangle = (|0\rangle + |1\rangle)/\sqrt{2}$

$$\hat{\rho} = |+\rangle\langle +| = \frac{1}{2}\left(|0\rangle\langle 0| + |1\rangle\langle 1| + |0\rangle\langle 1| + |1\rangle\langle 0|\right). \tag{18}$$

In contrast to the maximally mixed state, this quantum state is pure and we thus have perfect knowledge of the system. Performing a projective measurement on this state in the basis $|\pm\rangle$ will with certainty result in $+$. However, the state describes a coherent superposition of the states $|0\rangle$ and $|1\rangle$. This coherent superposition is distinguished from a mixture by the off-diagonal elements $|0\rangle\langle 1|$ and $|1\rangle\langle 0|$ (also called coherences).

**Time evolution of an isolated system:**   The time-evolution of a system that is isolated from its environment is given by

$$\hat{\rho}(t) = \hat{U}(t)\hat{\rho}(0)\hat{U}^\dagger(t), \qquad\qquad \hat{U}(t) = \mathcal{T}e^{-i\int_0^t dt' \hat{H}(t')}, \tag{19}$$

where $\hat{H}(t)$ is the Hamiltonian of the system and $\mathcal{T}$ denotes the time-ordering operator. The time-ordered exponential appearing in Eq. (19) can be written as

$$\mathcal{T}e^{-i\int_0^t dt' \hat{H}(t')} = \lim_{\delta t \to 0} e^{-i\delta t \hat{H}(t)} e^{-i\delta t \hat{H}(t-\delta t)} e^{-i\delta t \hat{H}(t-2\delta t)} \cdots e^{-i\delta t \hat{H}(2\delta t)} e^{-i\delta t \hat{H}(\delta t)} e^{-i\delta t \hat{H}(0)}, \tag{20}$$

such that the time argument in the Hamiltonian on the right-hand side increases from the right to the left of the expression. Each exponential in the product can be understood as the time-evolution by the infinitesimal time-step $\delta t$ [20].

Note that in quantum mechanics, a system that is isolated from its environment is denoted as a closed system. This can be confusing in the field of quantum thermodynamics because in thermodynamics, a closed system traditionally refers to a system that can exchange energy but cannot exchange matter with its environment. To avoid confusion, I use the term *isolated system* throughout these lecture notes.

### 1.1.3 Second quantization

Many scenarios considered in this course feature the transport of electrons or photons through a quantum system. Such scenarios are most conveniently described using the formalism of second quantization, see Sec. 1 in the Book by Bruus and Flensberg [21] for a more detailed introduction.

We first introduce the commutator and anti-commutator which play important roles for bosons and fermions respectively

$$
\begin{aligned}
&\text{Commutator:} && [\hat{A}, \hat{B}] = \hat{A}\hat{B} - \hat{B}\hat{A}, \\
&\text{Anti-commutator:} && \{\hat{A}, \hat{B}\} = \hat{A}\hat{B} + \hat{B}\hat{A}.
\end{aligned}
\tag{21}
$$

**Bosons:**   For a single bosonic mode, a central object is the creation operator $\hat{a}^\dagger$, which creates a boson in this mode. Its Hermitian conjugate $\hat{a}$ denotes the annihilation operator, removing a boson from the mode. These operators obey the canonical commutation relations

$$
[\hat{a}, \hat{a}^\dagger] = 1, \qquad\qquad [\hat{a}, \hat{a}] = [\hat{a}^\dagger, \hat{a}^\dagger] = 0.
\tag{22}
$$

The state without any bosons is denoted the vacuum state $|0\rangle$ and by definition it is annihilated by the annihilation operator, $\hat{a}|0\rangle = 0$. The state with $n$ bosons (also called a Fock state) can be written with help of the creation operator as

$$
|n\rangle \equiv \frac{\left(\hat{a}^\dagger\right)^n}{\sqrt{n!}} |0\rangle, \qquad\qquad \langle n|m\rangle = \delta_{n,m}.
\tag{23}
$$

With these definitions, we find the action of the creation and annihilation operators on the Fock states

$$
\hat{a}|n\rangle = \sqrt{n}|n-1\rangle, \qquad\qquad \hat{a}^\dagger|n\rangle = \sqrt{n+1}|n+1\rangle, \qquad\qquad \hat{a}^\dagger\hat{a}|n\rangle = n|n\rangle.
\tag{24}
$$

The operator $\hat{n} \equiv \hat{a}^\dagger\hat{a}$ is called the number operator.

When dealing with multiple bosonic modes, we introduce subscripts on the annihilation operators $\hat{a}_j$. The canonical commutation relations then read

$$
[\hat{a}_j, \hat{a}_k^\dagger] = \delta_{j,k}, \qquad\qquad [\hat{a}_j, \hat{a}_k] = [\hat{a}_j^\dagger, \hat{a}_k^\dagger] = 0.
\tag{25}
$$

The number states can be written as

$$
|n_0, n_1, \cdots\rangle = |\mathbf{n}\rangle \equiv \prod_j \frac{\left(\hat{a}_j^\dagger\right)^{n_j}}{\sqrt{n_j!}} |0\rangle, \qquad\qquad \hat{a}_j^\dagger\hat{a}_j|\mathbf{n}\rangle = n_j|\mathbf{n}\rangle.
\tag{26}
$$

**Fermions:**   For a single fermionic mode, the creation and annihilation operators are denoted as $\hat{c}^\dagger$ and $\hat{c}$ respectively. In contrast to bosonic operators, they obey canonical anti-commutation relations

$$\{\hat{c}, \hat{c}^\dagger\} = 1, \qquad\qquad \{\hat{c}, \hat{c}\} = \{\hat{c}^\dagger, \hat{c}^\dagger\} = 0. \qquad (27)$$

The latter relation directly implies that $\hat{c}^2 = (\hat{c}^\dagger)^2 = 0$. A fermionic mode may thus at most be occupied by a single fermion. This is known as the Pauli exclusion principle. Just like for bosons, the vacuum state $|0\rangle$ is annihilated by the annihilation operator, $\hat{c}|0\rangle = 0$. The Fock states can then be written as

$$|1\rangle = \hat{c}^\dagger|0\rangle, \qquad\qquad \hat{c}|1\rangle = |0\rangle, \qquad\qquad \hat{c}^\dagger\hat{c}|n\rangle = n|n\rangle, \qquad (28)$$

where $n = 0, 1$ and $\hat{c}^\dagger\hat{c}$ is called the number operator just like for bosons. Note that due to the Pauli exclusion principle, the occupied state $|1\rangle$ is annihilated by the creation operator $\hat{c}^\dagger|1\rangle = 0$, implying a symmetry between creation and annihilation operators. Indeed, we may call $\hat{c}^\dagger$ the annihilation operator for a *hole* which has the vacuum state $|1\rangle$.

Multiple modes are denoted by an index and obey the canonical anti-commutation relations

$$\{\hat{c}_j, \hat{c}_k^\dagger\} = \delta_{j,k}, \qquad\qquad \{\hat{c}_j, \hat{c}_k\} = \{\hat{c}_j^\dagger, \hat{c}_k^\dagger\} = 0. \qquad (29)$$

Note that operators for different modes *anti*-commute, implying a sign change when exchanging two fermions. Just as for bosons, the number states can be written as

$$|n_0, n_1, \cdots\rangle = |\mathbf{n}\rangle \equiv \prod_j \left(\hat{c}_j^\dagger\right)^{n_j} |0\rangle, \qquad\qquad \hat{c}_j^\dagger\hat{c}_j|\mathbf{n}\rangle = n_j|\mathbf{n}\rangle. \qquad (30)$$

Note however that do to the anti-commutation relations, the order with which $\hat{c}_j^\dagger$ act on the vacuum in the last expression matters.

### 1.1.4   Exact and inexact differentials

Exact and inexact differentials play an important role in thermodynamics. They are introduced in most textbooks on thermodynamics, see for instance App. A of the book by Hentschke [22].

We recall that the differential of a differentiable function $f(x, y)$ of two variables is denoted by

$$df(x, y) = \partial_x f(x, y)dx + \partial_y f(x, y)dy, \qquad (31)$$

where $\partial_x$ denotes the partial derivative with respect to $x$, keeping $y$ fixed. The integral of such a differential along any curve $\gamma$ is determined solely by its endpoints

$$\int_\gamma df(x, y) = f(x_f, y_f) - f(x_i, y_i), \qquad (32)$$

where the subscript $i$ ($f$) denotes the initial (final) values of the curve $\gamma$.

We now consider a differential form

$$dg(x, y) = a(x, y)dx + b(x, y)dy. \qquad (33)$$

We call this an exact differential if and only if

$$\partial_y a(x, y) = \partial_x b(x, y). \qquad (34)$$

In that case, we may write $a(x, y) = \partial_x g(x, y)$ and $b(x, y) = \partial_y g(x, y)$ and $g(x, y)$ is a differentiable function. An exact differential is therefore a differential form that can be

written as the differential of a function. We call the differential form in Eq. (33) an inexact differential if it cannot be written as the differential of a function. In this case, $\partial_y a(x, y) \neq \partial_x b(x, y)$. Importantly, the integral of an inexact differential generally does depend on the path of integration and not just its endpoints.

As an example, we consider the exact differential

$$dz = y\,dx + x\,dy, \tag{35}$$

which is the differential of the function $z(x, y) = xy$. The differential in Eq. (35) can be written as the sum of two inexact differentials

$$đq = y\,dx, \qquad\qquad đw = x\,dy, \tag{36}$$

where $đ$ denotes an inexact differential. We may now integrate these differentials along two different curves with the same endpoints

$$\gamma_1 : (0,0) \to (1,0) \to (1,1), \qquad\qquad \gamma_2 : (0,0) \to (0,1) \to (1,1). \tag{37}$$

Integrals along these paths may be evaluated using

$$
\begin{aligned}
\int_{\gamma_1} dg(x, y) &= \int_0^1 dx\, a(x, 0) + \int_0^1 dy\, b(1, y), \\
\int_{\gamma_2} dg(x, y) &= \int_0^1 dy\, a(0, y) + \int_0^1 dx\, b(x, 1),
\end{aligned}
\tag{38}
$$

where we made use of Eq. (33). A simple calculation results in

$$\int_{\gamma_1} đq = \int_{\gamma_2} đw = 1, \qquad \int_{\gamma_2} đq = \int_{\gamma_1} đw = 0. \tag{39}$$

As expected, we recover

$$\int_{\gamma_1} dz = \int_{\gamma_2} dz = z(1,1) - z(0,0) = 1. \tag{40}$$

## 1.2 Macroscopic thermodynamics

Here we briefly summarize some important aspects of the traditional theory of thermodynamics which applies at macroscopic scales, where fluctuations may safely be neglected. There are many textbooks on this topic, see for instance the books by Callen [23] or Hentschke [22].

Among physical theories, Thermodynamics takes a rather peculiar role because it is generally valid for all systems (at the macroscopic scale) and it does not provide numerical results but rather general equalities and inequalities which constrain physically allowed processes. Indeed, the theory of thermodynamics has been called the *village witch* among physical theories [12]. The main assumption in thermodynamics is that we can describe a macroscopic body by very few variables such as temperature $T$, pressure $p$, volume $V$, internal energy $U$, and entropy $S$, see Fig. 1. Given these macroscopic variables, thermodynamics provides relations and constraints between them.

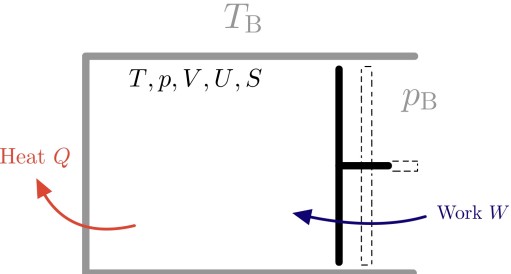

Figure 1: A gas in a container of volume $V$, at temperature $T$, with pressure $p$. The internal energy of the gas is given by $U$ and its entropy by $S$. Using a piston, the volume of the gas can be changed and work $W$ can be performed on the system. Energy may also be exchanged with the environment in the form of heat $Q$. Note that the temperature $T_B$ and pressure $p_B$ of the environment may differ from the temperature and pressure of the system.

**The first law of thermodynamics:**    The first law of thermodynamics may be written in the form

$$dU = đW - đQ, \tag{41}$$

The first law states that energy is conserved and that energy changes come in two forms: heat $đQ$ and work $đW$. Quite generally, work is a useful form of energy that is mediated by macroscopic degrees of freedom. This is in contrast to heat, which is a more inaccessible form of energy mediated by microscopic degrees of freedom. Note that these quantities are inexact differentials, i.e., they may depend on the path taken when changing thermodynamic variables. Here we use a sign convention where work is positive when increasing the energy of the body while heat is positive when it flows out of the body and into the environment.

The first law of thermodynamics prevents a perpetuum mobile of the first kind, i.e., a machine that produces the work it needs to run itself indefinitely. In the presence of any type of losses, such a machine needs to produce work out of nothing, violating Eq. (41). An example would be a lamp that is powered by a solar cell, illuminated only by the lamp itself, see Fig. 2.

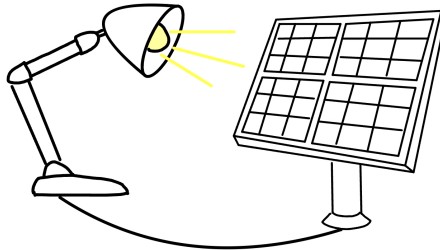

Figure 2: Perpetuum mobile of the first kind. A lamp illuminates a solar panel which in turn powers the lamp. Without any hidden energy source, such a device cannot work because of unavoidable losses. At long times, we may set $dU = 0$. Furthermore, we split the work $đW = đW_L - đW_S$, where $W_L$ denotes the work exerted on the lamp by the solar panel and $W_S$ denotes the work exerted on the solar panel by the lamp. In the presence of losses within the solar panel (any other loss mechanism results in similar conclusions), it provides strictly less work to the lamp than it received, i.e., $đW_L < đW_S$. By Eq. (41), this would necessitate another source of energy, either in the form of work or in the form of heat with $đQ < 0$.

Work may come in different forms. For instance, electrical work (also called chemical work) is produced when charges are moved from a region with lower voltage to a region with

higher voltage, which is what happens when charging a battery. While this type of work will be the focus of later parts of these lecture notes, here we focus on the traditional example of mechanical work which is produced by changing the volume of a gas using, e.g., a piston as illustrated in Fig. 1. In this case, we know from classical mechanics that work is given by $dW = -p_{\mathrm{B}}dV$, where $p_{\mathrm{B}}$ denotes the external pressure exerted by the piston. Furthermore, the heat flowing into the body can be related to the entropy change of the surrounding environment $dQ = T_{\mathrm{B}}dS_{\mathrm{B}}$, where the subscript $B$ stands for "bath´´. This relation follows from describing the environment as being in thermal equilibrium at temperature $T_{\mathrm{B}}$ and we will revisit this later. We stress that in these expressions for heat and work, the quantities of the surrounding environment appear (note that $dV = -dV_{\mathrm{B}}$).

**The second law of thermodynamics:**   The second law of thermodynamics is provided by the Clausius inequality [24]

$$d\Sigma = dS + dS_{\mathrm{B}} = dS + \sum_{\alpha} \frac{dQ_{\alpha}}{T_{\alpha}} \geq 0, \tag{42}$$

where we included multiple reservoirs at respective temperatures $T_{\alpha}$ that may exchange heat $dQ_{\alpha}$ with the body. The second law states that the total entropy, which is the sum of the entropy of the body $S$ and the entropy of the environment $S_{\mathrm{B}} = \sum_{\alpha} S_{\alpha}$, can only increase. This change in total entropy is called entropy production $\Sigma$.

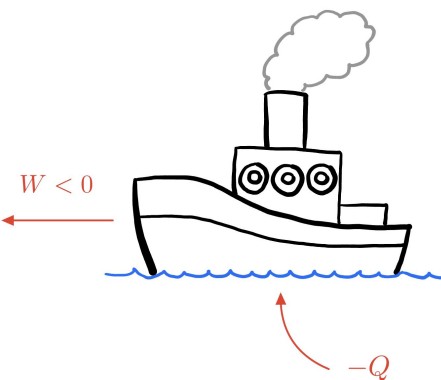

Figure 3: Perpetuum mobile of the second kind. A boat moves across the sea, being powered by the heat absorbed from the surrounding water. To move in the water, the boat needs to perform work against the friction force, i.e., $dW < 0$. At long times, we may neglect any changes in the internal energy or the entropy of the system, i.e., $dS = dU = 0$. The work required to move against the friction force must then be provided by heat coming from the environment, requiring $dQ < 0$. Since the environment only has a single temperature, this violates the second law of thermodynamics given in Eq. (42) which requires $dQ/T \geq 0$.

The second law of thermodynamics prevents a perpetuum mobile of the second kind, i.e., a machine that runs indefinitely by extracting heat out of an equilibrium environment. In this case, there is only one temperature $T$ describing the environment. Furthermore, for long times we may neglect $U$ as well as $S$ since they remain finite while the produced work $-W$ increases indefinitely. The first law then requires $W = Q$ which when inserted in Eq. (42) provides $W/T \geq 0$, preventing any production of work. An example would be given by a boat that moves (performing work against the friction force of the water) by extracting heat from the surrounding water, see Fig. 3. Note that dissipating work (i.e., turning work into heat, $W = Q \geq 0$) is perfectly allowed by the laws of thermodynamics and indeed is what happens when an incandescent light bulb is glowing.

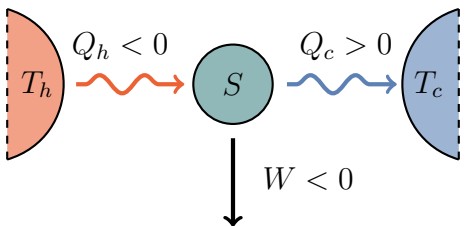

Figure 4: Working principle of a heat engine. Heat from a hot reservoir $Q_h$ is partly converted into work $W$ and partly dissipated into the cold reservoir $Q_c$. The dissipation ensures that the entropy production is positive, as required by the second law of thermodynamics, and limits the efficiency to be below the Carnot efficiency, see Eq. (44).

In addition to this example, the second law of thermodynamics provides important limitations on physically allowed processes. Consider a heat engine, where the temperature gradient between two thermal reservoirs at temperatures $T_h > T_c$ is exploited to produce work, see Fig. 4. Let us further consider a long-time scenario, where any changes in the system may be neglected compared to the heat and work which continuously increase. The first and second laws of thermodynamics then read

$$W = Q_h + Q_c, \qquad\qquad \frac{Q_h}{T_h} + \frac{Q_c}{T_c} \geq 0. \qquad (43)$$

These equations have a few remarkable consequences. First, for $W = 0$ it is straightforward to show that $-Q_h = Q_c \geq 0$, i.e, heat flows out of the hot and into the cold reservoir. Second, either $Q_h$, $Q_c$, or both have to be positive, i.e., it is not possible to extract heat simultaneously from both reservoirs. This is equivalent to the argument preventing a perpetuum mobile of the second kind. A heat engine, defined by $-W > 0$ will thus require an influx of heat from the hot reservoir $Q_h < 0$, providing the necessary energy, as well as a heat outflux into the cold reservoir $Q_c > 0$ ensuring the increase in entropy. The second law of thermodynamics then implies the well-known result that the efficiency of such a heat engine is upper bounded by the Carnot efficiency

$$\eta \equiv \frac{-W}{-Q_h} \leq 1 - \frac{T_c}{T_h}, \qquad (44)$$

where the efficiency can be understood as the ratio between the desired output $(-W)$ divided by the resource that is used $(-Q_h)$.

**Thermodynamic processes and cycles:**   In a thermodynamic process, the state of a body is changed by, e.g., changing its volume $V$ or pressure $p$. Such changes can be applied under different conditions. For instance, in an *adiabatic* process, the state is changed without exchanging heat with the environment. Similarly, processes at constant temperature $T$ or entropy $S$ are called *isothermal* and *isentropic* processes respectively. Note that the word *adiabatic* is used differently in thermodynamics and in quantum mechanics. In quantum mechanics, it refers to processes that are sufficiently slow such that a system in its ground state remains in its groundstate throughout the process. For this reason, I will try to avoid the word adiabatic. An important class of thermodynamic processes are *reversible processes*, which obey $\Sigma = 0$. This generally requires that the body is in thermal equilibrium with its environment at all times. Reversible processes therefore have to be very slow and always are an idealization of any real process. As we will see below, reversible processes are very useful to derive thermodynamic relations.

By combining different thermodynamic processes, we can design thermodynamic cycles. After a completed cycle, the body is in the same state but since $đQ$ and $đW$ are inexact differentials, we generally have

$$\oint dU = 0, \qquad\qquad \oint đQ = \oint đW \neq 0. \tag{45}$$

Thus, it is possible to turn heat into work in a thermodynamic cycle. Prominent examples of thermodynamic cycles are the Otto cycle, providing the underlying principle behind car engines, and the Carnot engine, which is an idealized, reversible engine that reaches the Carnot efficiency given in Eq. (44).

**Maxwell relations:**   Let us consider the first law of thermodynamics for a reversible process. As the body is in equilibrium with the environment throughout the process, we may identify $T = T_B$, $p = p_B$ and because $d\Sigma = 0$, we have $dS = -dS_B$. The first law thus reads

$$dU = TdS - pdV = \partial_S U dS + \partial_V U dV. \tag{46}$$

Since this is an exact differential, the last expression holds no matter if the process is reversible or not. However, the identification of heat and work with the first and second term on the right-hand side is only justified for reversible processes. Since $U$ is a smooth function, partial derivatives with respect to different variables commute and we find the Maxwell relation

$$\partial_S \partial_V U = \partial_V T = -\partial_S p. \tag{47}$$

Other Maxwell relations may be obtained similarly from other thermodynamic potentials such as the free energy $F = U - TS$. The free energy plays a particularly important role as it captures the capacity of a body to do work (in contrast to $U$ capturing the capacity to provide heat and work). The differential of the free energy reads

$$dF = -SdT - pdV = \partial_T F dS + \partial_V F dV, \tag{48}$$

from which we may derive the Maxwell relation

$$-\partial_T \partial_V F = \partial_V S = \partial_T p. \tag{49}$$

**Equations of state:**   In order to obtain quantitative predictions, the theory of thermodynamics has to be supplemented by *equations of state*. For instance, a monatomic ideal gas with $N$ atoms obeys the relations

$$U = \frac{3}{2} N k_B T, \qquad\qquad pV = N k_B T, \tag{50}$$

where $k_B$ denotes the Boltzmann constant. With the help of these equations, we may then derive non-trivial quantitative results such as the change in entropy when changing temperature and volume

$$S(T, V) - S(T_0, V_0) = k_B \ln\left[\left(\frac{T}{T_0}\right)^{3/2} \frac{V}{V_0}\right]. \tag{51}$$

## 1.3   Information theory

As we will see below, entropy is closely related to concepts from information theory. Here we provide a brief introduction to these concepts, for a more detailed discussion see for instance the original paper by Shannon [25] or the book by Cover and Thomas [26].

**Self-information:** Let us consider a random variable $X$ with outcomes $x_1, \cdots, x_n$ occuring with probabilities $p_1, \cdots, p_n$. Note that in this section, we use a different symbol for the random variable $X$ and its outcomes $x_j$ as is customary in mathematics. The *self-information* or *surprisal* of an outcome $x_j$ is defined as

$$I_j = -\ln p_j. \tag{52}$$

It quantifies the information that is gained when observing outcome $x_j$. The self-information has the following properties:

i) The less probable an event is, the more surprising it is and the more information we gain

$$\lim_{p_j \to 1} I_j = -\ln 1 = 0, \qquad \qquad \lim_{p_j \to 0} I_j = \infty. \tag{53}$$

ii) The information of uncorrelated events is additive. Let $p_j^X$ and $p_k^Y$ denote the probabilities that the independent random variables $X$ and $Y$ take on the values $x_j$ and $y_k$ respectively. It then holds that

$$p_{j,k} = p_j^X p_k^Y, \qquad \qquad I_{j,k} = -\ln p_{j,k} = -\ln p_j^X - \ln p_k^Y = I_j^X + I_k^Y, \tag{54}$$

where $p_{j,k}$ denotes the joint probability of observing outcomes $x_j$ and $y_k$.

We note that with Eq. (52), information is quantified in *nats* since we are using the natural logarithm. To quantify information in *bits*, one should use the logarithm of base two instead.

**Shannon entropy:** The Shannon entropy is defined as the average self-information

$$H(X) = \sum_j p_j I_j = -\sum_j p_j \ln p_j. \tag{55}$$

It quantifies the average information gain when observing the random variable $X$, or, equivalently, the lack of information before observation. The practical importance of the Shannon entropy is highlighted by the *noiseless coding theorem* which states that storing a random variable taken from a distribution with Shannon entropy $H$ requires at least $H$ nats on average (if $H$ is defined using $\log_2$, then it requires at least $H$ bits). Any further compression results in loss of information. This theorem can be illustrated using an example taken from Nielsen and Chuang [17]:

Consider a source that produces the symbols $A$, $B$, $C$, and $D$ with respective probabilities $1/2$, $1/4$, $1/8$, and $1/8$. If we do not use any compression, we need two bits of storage per symbol, e.g.,

$$A \to 00, \quad B \to 01, \quad C \to 10, \quad D \to 11. \tag{56}$$

However, the symbols provided by the source may be encoded using less bits per symbol on average by taking into account the probability distribution of the source. The basic idea is to use less bits for the more frequent symbols. For instance, we may use the following encoding

$$A \to 0, \quad B \to 10, \quad C \to 110, \quad A \to 111. \tag{57}$$

Note that we cannot use $B \to 1$ because then we can no longer determine if $110$ means $C$ or if it means $BBA$. With this compression, we find the average number of bits per symbol to be $1/2 \cdot 1 + 1/4 \cdot 2 + 1/8 \cdot 3 + 1/8 \cdot 3 = 7/4$. This equals the Shannon entropy (in bits)

$$H = -\frac{1}{2}\log_2 \frac{1}{2} - \frac{1}{4}\log_2 \frac{1}{4} - \frac{1}{8}\log_2 \frac{1}{8} - \frac{1}{8}\log_2 \frac{1}{8} = \frac{7}{4}. \tag{58}$$

It turns out that any compression algorithm that uses less bits results in the loss of information.

**Von Neumann entropy:**    The quantum version of the Shannon entropy is given by the von Neumann entropy

$$S_{\mathrm{vN}}[\hat{\rho}] = -\mathrm{Tr}\{\hat{\rho}\ln\hat{\rho}\} = -\sum_j p_j \ln p_j, \tag{59}$$

where $\hat{\rho} = \sum_j p_j |j\rangle\langle j|$. As a generalization of the Shannon entropy to quantum systems, it describes the lack of knowledge we have over a quantum system given a state $\hat{\rho}$. The von Neumann entropy is minimized for pure states and maximized for the maximally mixed state given by the identity matrix

$$S_{\mathrm{vN}}[|\psi\rangle\langle\psi|] = 0, \qquad\qquad S_{\mathrm{vN}}[\mathbb{1}/d] = \ln d, \tag{60}$$

where $d$ denotes the dimension of the Hilbert space.

**Relative entropy:**    The relative entropy, also known as the *Kullback-Leibler divergence* [27] is useful to quantify information when a random variable is described by an erroneous distribution. For two distributions $\{p_j\}$ and $\{q_j\}$, the relative entropy is defined as

$$D_{\mathrm{KL}}[\{p_j\}||\{q_j\}] = \sum_j p_j \ln\frac{p_j}{q_j}. \tag{61}$$

By definition, $D_{\mathrm{KL}} \geq 0$ where equality is only achieved if the two distributions are the same. To shed some light onto the interpretation of this quantity, let us assume that a random variable $X$ is distributed according to $\{p_j\}$ but we describe it by the erroneous distribution $\{q_j\}$. The information gain that we attribute to the outcome $x_j$ then reads $-\ln q_j$. On average, this gives

$$-\sum_j p_j \ln q_j = -\sum_j p_j \ln p_j + \sum_j p_j \ln\frac{p_j}{q_j} = H(X) + D_{\mathrm{KL}}[\{p_j\}||\{q_j\}]. \tag{62}$$

The left-hand side of this equation provides the information we assign to the random variable on average, the assumed information. The right-hand side provides the actual average information, given by the Shannon entropy, plus the relative entropy. The actual information is thus strictly smaller than the assumed information and the discrepancy is quantified by the relative entropy. We can think of the relative entropy as an *information loss* due to the erroneous description.

**Quantum relative entropy:**    The generalization of the relative entropy to the quantum scenario is defined as [28]

$$S[\hat{\rho}||\hat{\sigma}] = \mathrm{Tr}\{\hat{\rho}\ln\hat{\rho} - \hat{\rho}\ln\hat{\sigma}\}. \tag{63}$$

Just as its classical counterpart, is always non-negative $S[\hat{\rho}||\hat{\sigma}] \geq 0$ and it quantifies the discrepancy between the assumed and the actual lack of information when a system in state $\hat{\rho}$ is erroneously described by the state $\hat{\sigma}$.

**Mutual information:**    The mutual information is a measure that characterizes the mutual dependence of two random variables. It tells us how much information one random variable has on the other and it is defined as

$$I_{\mathrm{MI}}(X;Y) = D_{\mathrm{KL}}[\{p_{j,k}\}||\{p_j^X p_k^Y\}] = H(X) + H(Y) - H(X,Y). \tag{64}$$

To extend this definition to the quantum regime, we consider a bi-partite system in a Hilbert space $\mathcal{H} = \mathcal{H}_{\mathrm{A}} \otimes \mathcal{H}_{\mathrm{B}}$. The quantum mutual information in a quantum state $\hat{\rho}_{\mathrm{AB}}$ is defined as

$$I_{\mathrm{MI}}[\mathrm{A};\mathrm{B}] = S[\hat{\rho}_{AB}||\hat{\rho}_{\mathrm{A}} \otimes \hat{\rho}_{\mathrm{B}}] = S_{\mathrm{vN}}[\hat{\rho}_{\mathrm{A}}] + S_{\mathrm{vN}}[\hat{\rho}_{\mathrm{B}}] - S_{\mathrm{vN}}[\hat{\rho}_{\mathrm{AB}}], \tag{65}$$

where we introduced $\hat{\rho}_A = \text{Tr}_B\{\hat{\rho}_{AB}\}$ and $\hat{\rho}_B = \text{Tr}_A\{\hat{\rho}_{AB}\}$. Since the quantum relative entropy is a non-negative quantity, so is the mutual information. Furthermore, the mutual information is bounded from above by a corollary of the Araki–Lieb inequality [29]

$$I_{\text{MI}}[A; B] \leq 2\min\{S_{\text{vN}}[\hat{\rho}_A], S_{\text{vN}}[\hat{\rho}_B]\} \leq 2\min\{d_A, d_B\}, \tag{66}$$

where $d_\alpha$ denotes the dimension of the Hilbert space $\mathcal{H}_\alpha$. The last inequality follows from the upper bound on the von Neumann entropy given in Eq. (60).

## 2 Thermodynamic equilibrium

Equilibrium states may be characterized by a few intensive variables such as temperature $T$, chemical potential $\mu$, and pressure $p$. In equilibrium, no net currents flow and the system is stationary, i.e., observables become time-independent. Note the similarity to the main assumption in macroscopic thermodynamics, see Sec. 1.2. This is no coincidence. Indeed, in macroscopic thermodynamics, systems are assumed to be in local equilibrium (i.e., can be described by a thermal state with variables that may differ from the environment).

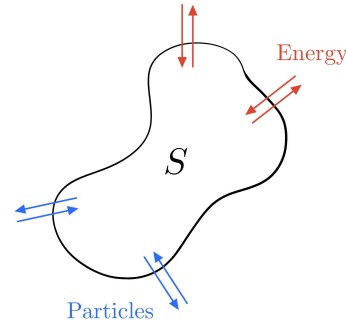

Figure 5: System exchanging energy and particles with the environment.

### 2.1 The grand-canonical ensemble

Throughout these lecture notes, we consider a system that may exchange both energy and particles with its environmnet as sketched in Fig. 5. In equilibrium, such a system is described by the grand-canonical Gibbs state

$$\hat{\rho}_G = \frac{e^{-\beta(\hat{H} - \mu\hat{N})}}{Z}, \qquad\qquad Z = \text{Tr}\left\{e^{-\beta(\hat{H} - \mu\hat{N})}\right\}, \tag{67}$$

where $\beta = 1/k_B T$ denotes the inverse temperature, $\hat{H}$ is the Hamiltonian, $\hat{N}$ the particle-number operator, and $Z$ is called the partition function. For equilibrium states, we may generally identify the relevant quantities in macroscopic thermodynamics as average values

$$U \equiv \langle\hat{H}\rangle, \qquad S \equiv k_B S_{\text{vN}}[\hat{\rho}]. \tag{68}$$

Because the Gibb's state is of central importance for much that follows, we now discuss three different ways of motivating why it provides an adequate description for thermodynamic equilibrium.

### 2.1.1   Subsystem of an isolated system

The first motivation considers the system of interest to be a small part of a very large "super-system" that is described as an isolated system, see Fig. 6. This motivation for the Gibbs state can be found in many textbooks, see e.g., Landau & Lifshitz [30] or Schwabl [31]. We start by assuming that the supersystem has a fixed energy and particle number given by $E_{\text{tot}}$ and $N_{\text{tot}}$ respectively. In this case, it is described by the so-called micro-canonical ensemble

$$\hat{\rho}_{\text{tot}} = \sum_{E,N} p_{\text{tot}}(E,N) |E,N\rangle\langle E,N|, \tag{69}$$

where $p_{\text{tot}}(E,N)$ is nonzero only if the energy and particle number are close to $E_{\text{tot}}$ and $N_{\text{tot}}$, i.e., for $E_{\text{tot}} \leq E < E_{\text{tot}} + \delta E$ and $N_{\text{tot}} \leq N < N_{\text{tot}} + \delta N$. For energies and particle numbers within these shells, each state is equally likely in the microcanonical ensemble, i.e.,

$$p_{\text{tot}}(E,N) = \frac{\Theta(E - E_{\text{tot}})\Theta(N - N_{\text{tot}})\Theta(E_{\text{tot}} + \delta E - E)\Theta(N_{\text{tot}} + \delta N - N)}{\Omega(E_{\text{tot}}, N_{\text{tot}})\delta E \delta N}, \tag{70}$$

where $\Omega(E_{\text{tot}}, N_{\text{tot}})$ denotes the density of states and $\Theta(x)$ the Heaviside theta function which is equal to one for $x > 0$ and zero otherwise. Being a macroscopic supersystem, the number of states contributing to the microcanonical ensemble is assumed to be large

$$\Omega(E_{\text{tot}}, N_{\text{tot}})\delta E \delta N \gg 1. \tag{71}$$

At the same time, for energy and particle numbers to be well defined, we require $\delta E \ll E_{\text{tot}}$ and $\delta N \ll N_{\text{tot}}$. This also justifies to assume that the density of states is constant in the energy and particle number shells that contribute to the micro-canonical ensemble. Fur future reference, we note that the von Neumann entropy of the microcanoncial ensemble is given by the logarithm of the number of contributing states [see also Eq. (60)]

$$S_{\text{vN}}(\hat{\rho}_{\text{tot}}) = \ln[\Omega(E_{\text{tot}}, N_{\text{tot}})\delta E \delta N]. \tag{72}$$

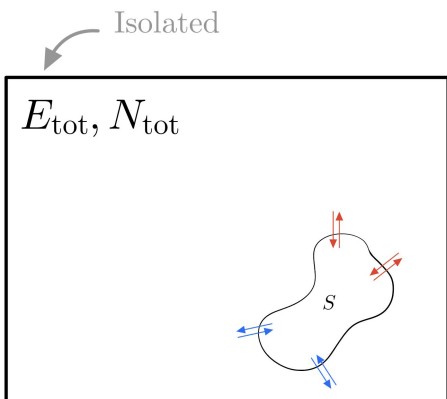

Figure 6: System of interest as a small part of a large, isolated "supersystem". The energy and particle number in the supersystem are fixed to $E_{\text{tot}}$ and $N_{\text{tot}}$ respectively. The reduced state of the system can then be shown to be the Gibbs state.

Since our system of interest is part of the supersystem, we may write the Hamiltonian and particle-number operator as

$$\hat{H}_{\text{tot}} = \hat{H}_{\text{S}} + \hat{H}_{\text{B}} + \hat{V}, \qquad\qquad \hat{N}_{\text{tot}} = \hat{N}_{\text{S}} + \hat{N}_{\text{B}}, \tag{73}$$

where the subscript B stands for "bath" and describes the remainder of the supersystem. We now make a weak coupling approximation that is crucial to obtain the Gibbs state: we assume that $\hat{V}$ is sufficiently weak such that we may neglect the coupling energy and the energy and particle eigenstates of the supersystem approximately factorize

$$|E, N\rangle \simeq |E_S, N_S\rangle \otimes |E_B, N_B\rangle . \tag{74}$$

The density matrix in Eq. (69) may then be written as

$$\hat{\rho}_{\text{tot}} = \sum_{E_S, N_S} \sum_{E_B, N_B} p_{\text{tot}}(E_S + E_B, N_S + N_B) |E_S, N_S\rangle\langle E_S, N_S| \otimes |E_B, N_B\rangle\langle E_B, N_B| . \tag{75}$$

We are interested in the reduced state of the system obtained by taking the partial trace over the bath

$$\hat{\rho}_S = \text{Tr}_B\{\hat{\rho}_{\text{tot}}\} = \sum_{E_B, N_B} \langle E_B, N_B| \hat{\rho}_{\text{tot}} |E_B, N_B\rangle = \sum_{E_S, N_S} p_S(E_S, N_S) |E_S, N_S\rangle\langle E_S, N_S| . \tag{76}$$

The probability for the system to be in a state with energy $E_S$ and particle number $N_S$ may be written as

$$p_S(E_S, N_S) = \sum_{E_B, N_B} p_{\text{tot}}(E_S + E_B, N_S + N_B) = \frac{\Omega_B(E_{\text{tot}} - E_S, N_{\text{tot}} - N_S)\delta E \delta N}{\Omega(E_{\text{tot}}, N_{\text{tot}})\delta E \delta N} , \tag{77}$$

where the numerator in the last expression denotes the number of states in the bath, i.e., the number of terms in the sum $\sum_{E_B, N_B} p_{\text{tot}}(E_S + E_B, N_S + N_B)$ which are non-vanishing.

In analogy to Eq. (72), the entropy of the bath is given by

$$S_B(E, N) = k_B \ln[\Omega_B(E, N)\delta E \delta N] . \tag{78}$$

We note that the factor of $k_B$ appears here because this is the thermodynamic, not the von Neumann entropy. Since the system is considered to be small, we may expand the bath entropy around $E_{\text{tot}}$ and $N_{\text{tot}}$ to obtain

$$\begin{aligned}
S_B(E_{\text{tot}} - E_S, N_{\text{tot}} - N_S) &\simeq S_B(E_{\text{tot}}, N_{\text{tot}}) - E_S \partial_E S_B(E, N_{\text{tot}})|_{E=E_{\text{tot}}} \\
&\quad - N_S \partial_N S_B(E_{\text{tot}}, N)|_{N=N_{\text{tot}}} \\
&= S_B(E_{\text{tot}}, N_{\text{tot}}) - \frac{E_S - \mu N_S}{T} ,
\end{aligned} \tag{79}$$

where we identified $\partial_E S_B(E, N) = 1/T$ and $\partial_N S_B(E, N) = -\mu/T$. Inserting the last expression into Eq. (77) results in

$$p_S(E_S, N_S) = \frac{\Omega_B(E_{\text{tot}}, N_{\text{tot}})\delta E \delta N}{\Omega(E_{\text{tot}}, N_{\text{tot}})\delta E \delta N} e^{-\beta(E_S - \mu N_S)} = \frac{e^{-\beta(E_S - \mu N_S)}}{Z} , \tag{80}$$

where the last equality follows from the normalization of the state. Since this is exactly the probability for the Gibbs state given in Eq. (67), we have shown that a small subsystem of a large and isolated supersystem is described by the Gibbs state if the supersystem is described by the microcanonical ensemble (i.e., has well defined energy and particle number) and if the system is weakly coupled to the remainder of the supersystem.

### 2.1.2 Jaynes' maximum entropy principle

The second motivation for the Gibbs state that we discuss is based on Jaynes' maximum entropy principle [32, 33]. The basic idea behind this principle is that an adequate description of the physical system should only encode the information that we actually have. As discussed above, equilibrium states may be described by intensive variables such as temperature and chemical potential. Equivalently, they may be described by the conjugate extensive quantities provided by the average energy $\bar{E}$ and the average particle number $\bar{N}$. Following Jaynes', we take $\bar{E}$ and $\bar{N}$ as the prior data, i.e., the knowledge that we have about the state. Jaynes' crucial insight was that in order to avoid encoding any other knowledge in the state, the entropy, a quantifier for lack of knowledge, should be maximized. Mathematically, the Gibbs state is then obtained by maximizing the von Neumann entropy under the constraints

$$\bar{E} = \langle \hat{H} \rangle, \qquad\qquad \bar{N} = \langle \hat{N} \rangle. \tag{81}$$

To perform this maximization, we start by considering states of the form

$$\hat{\rho} = \sum_n p_n |E_n, N_n\rangle\langle E_n, N_n|, \quad \hat{H} |E_n, N_n\rangle = E_n |E_n, N_n\rangle, \quad \hat{N} |E_n, N_n\rangle = N_n |E_n, N_n\rangle. \tag{82}$$

This form may be motivated by demanding that equilibrium states do not change in time, which implies [see also Eq. (19)]

$$\hat{U}(t)\hat{\rho}\hat{U}^\dagger(t) = \hat{\rho} \quad \Longleftrightarrow \quad [\hat{H}, \hat{\rho}] = 0, \tag{83}$$

and that there are no coherent superpositions of states with different number of particles, which implies $[\hat{N}, \hat{\rho}] = 0$. The latter assumption may be motivated by a particle superselection rule [34] and it does not apply when considering scenarios with a vanishing chemical potential $\mu = 0$. We now want to maximize the von Neumann entropy

$$S_{\text{vN}}[\hat{\rho}] = -\sum_n p_n \ln p_n, \tag{84}$$

under the following three constraints

$$\sum_n p_n = 1, \qquad \sum_n p_n E_n = \bar{E}, \qquad \sum_n p_n N_n = \bar{N}, \tag{85}$$

where the first constraint simply ensures the normalization of the state. Such constrained maximization problems can be solved by the method of Lagrange multipliers [35]. To this end, we define the function

$$\mathcal{L} = -\sum_n p_n \ln p_n - \lambda_0 \left( \sum_n p_n - 1 \right) - \lambda_1 \left( \sum_n p_n E_n - \bar{E} \right) - \lambda_2 \left( \sum_n p_n N_n - \bar{N} \right). \tag{86}$$

The solution to the maximization procedure is the distribution $\{p_n\}$ that obeys

$$\frac{\partial \mathcal{L}}{\partial p_n} = -\ln p_n - 1 - \lambda_0 - \lambda_1 E_n - \lambda_2 N_n = 0. \tag{87}$$

We now identify

$$\lambda_1 = \beta, \qquad \lambda_2 = -\beta\mu, \qquad e^{\lambda_0 + 1} = Z. \tag{88}$$

The solution of Eq. (87) then reads

$$p_n = \frac{e^{-\beta(E_n - \mu N_n)}}{Z}, \tag{89}$$

recovering the Gibbs state given in Eq. (67). The Lagrange multipliers or, equivalently, the quantities $\beta$, $\mu$, and $Z$ are fully determined by the constraints given in Eq. (85).

Introducing the *grand potential* which plays a similar role to the free energy

$$\Phi_G = \langle \hat{H} \rangle - k_B T S_{vN}[\hat{\rho}] - \mu \langle \hat{N} \rangle, \tag{90}$$

we may write Eq. (86) as

$$\mathcal{L} = -\beta \Phi_G - \lambda_0 \left( \sum_n p_n - 1 \right) + \beta(\bar{E} - \mu \bar{N}). \tag{91}$$

The last term may be dropped as it does not depend on $p_n$. The last equation then implies that maximizing the von Neumann entropy given the average energy and average particle number is mathematically equivalent to minimizing the grand potential (the only constraint left, corresponding to the Lagrange multiplier $\lambda_0$, ensures the normalization of the state).

### 2.1.3 Complete passivity

The final motivation we provide for the Gibbs state is based on the notion of *passive states* [36, 37]. These are states from which no work can be extracted. To define passive states, we consider an isolated system that obeys the time-evolution given in Eq. (19). For an isolated system with a time-dependent Hamiltonian, we interpret the total energy change of the system as work. This can be motivated by considering the time-dependence of the Hamiltonian as mediated by a classical field describing a macroscopic degree of freedom. In analogy to a piston compressing a gas, the energy change mediated by this macroscopic degree of freedom should be considered as work. We now consider a cyclic work protocol of duration $\tau$, where the Hamiltonian is changed over time $\hat{H}(t)$ such that

$$\hat{H} \equiv \hat{H}(0) = \hat{H}(\tau). \tag{92}$$

The work extracted during this protocol reads

$$W_{ex} = \text{Tr}\{\hat{H}\hat{\rho}\} - \text{Tr}\{\hat{H}\hat{U}\hat{\rho}\hat{U}^\dagger\}, \tag{93}$$

where $\hat{\rho}$ denotes the initial state of the system and the unitary $\hat{U} \equiv \hat{U}(\tau)$ is determined by the time-dependent Hamiltonian, see Eq. (19). We note that with a suitable Hamiltonian, any unitary operator can be generated.

A passive state can now be defined as a state $\tau$ obeying

$$\text{Tr}\{\hat{H}(\hat{\tau} - \hat{U}\hat{\tau}\hat{U}^\dagger)\} \leq 0 \quad \forall \hat{U}, \tag{94}$$

which implies that no work can be extracted from passive states, no matter how the time-dependent Hamiltonian is designed. Passive states are of the form

$$\hat{\tau} = \sum_n p_n |E_n\rangle\langle E_n|, \quad \text{with} \quad E_0 \leq E_1 \leq E_2 \cdots \quad p_0 \geq p_1 \geq p_2 \cdots \tag{95}$$

Remarkably, taking multiple copies of a passive state may result in a state that is no longer passive. This suggests the definition of *completely passive states* [36]: a state $\hat{\tau}$ is completely passive iff

$$\text{Tr}\{\hat{H}_n(\tau^n - \hat{U}\hat{\tau}^n\hat{U}^\dagger)\} \leq 0 \quad \forall \hat{U}, n, \tag{96}$$

where

$$\hat{H}_n = \hat{H} \oplus \hat{H} \oplus \cdots \oplus \hat{H}, \qquad \hat{\tau}^n = \hat{\tau} \otimes \hat{\tau} \otimes \cdots \otimes \hat{\tau}. \tag{97}$$

Here we introduced the Kronecker sum as $\hat{A} \oplus \hat{B} = \hat{A} \otimes \mathbb{1} + \mathbb{1} \otimes \hat{B}$. This implies that it is not possible to extract work from completely passive states even if we take multiple copies and let them interact during the work protocol. Remarkably, the only completely passive states are Gibbs states in the canonical ensemble (assuming no degeneracy in the ground state) [36, 38, 39]

$$\hat{\rho}_{\mathrm{c}} = \frac{e^{-\beta \hat{H}}}{\mathrm{Tr}\{e^{-\beta \hat{H}}\}}. \tag{98}$$

In complete analogy, the grand-canonical Gibbs state is completely passive with respect to $\hat{H} - \mu \hat{N}$, i.e., the average $\langle \hat{H} - \mu \hat{N} \rangle$ cannot be lowered by any unitary, even when taking multiple copies. If we further assume a particle-superselection rule [34], preventing the creation of coherent superpositions of different particle numbers as long as $\mu \neq 0$, then the grand-canonical Gibbs state in Eq. (67) is completely passive since $[\hat{U}, \hat{N}] = 0$.

## 2.2 Equivalence of ensembles in the thermodynamic limit

We have already encountered the grand-canonical, the canonical, as well as the micro-canonical ensemble. Here we discuss these ensembles in a bit more detail and argue that they all become equivalent in the thermodynamic limit, which is the limit of large systems relevant for macroscopic thermodynamics.

**Microcanonical ensemble**

$$\hat{\rho}_{\mathrm{M}} = \sum_{\substack{\bar{E} \leq E < \bar{E}+\delta E \\ \bar{N} \leq N < \bar{N}+\delta N}} \frac{1}{\Omega(\bar{E}, \bar{N})\delta E \delta N} |E, N\rangle\langle E, N|, \tag{99}$$

where $\Omega(\bar{E}, \bar{N})$ denotes the density of states that we have encountered in Eq. (71). In the microcanonical ensemble, both the particle number as well as the energy is fixed, where fixed means that all states within an energy (particle number) shell of thickness $\delta E$ ($\delta N$) contribute. Within this shell, the state is assumed to be completely mixed, i.e., all states have equal weight. As such, the state maximizes the von Neumann entropy for fixed $\bar{E}$ and $\bar{N}$.

**Canonical ensemble**

$$\hat{\rho}_{\mathrm{c}} = \sum_{\bar{N} \leq N < \bar{N}+\delta N} \sum_{E} \frac{e^{-\beta E}}{Z_{\mathrm{c}}} |E, N\rangle\langle E, N| = \frac{e^{-\beta \hat{H}_{\bar{N}}}}{\mathrm{Tr}\{e^{-\beta \hat{H}_{\bar{N}}}\}}, \tag{100}$$

where $\hat{H}_{\bar{N}}$ denotes the Hamiltonian projected onto the relevant particle-number subspace

$$\hat{H}_{\bar{N}} = \sum_{\bar{N} \leq N < \bar{N}+\delta N} |N\rangle\langle N| \hat{H} |N\rangle\langle N|, \qquad |N\rangle = \sum_{E} |E, N\rangle. \tag{101}$$

In the canonical ensemble, only the particle number is fixed. The average energy $\langle \hat{H} \rangle$ is determined by the temperature of the environment. The canonical ensemble is the adequate equilibrium state when energy (but not particles) can be exchanged with the environment. The canonical Gibbs state maximizes the von Neumann entropy for a fixed particle number $\bar{N}$ and an average energy given by $\langle \hat{H} \rangle = \bar{E}$. Furthermore, the canonical Gibbs state minimizes the free energy

$$F = \langle \hat{H} \rangle - k_{\mathrm{B}} T S_{\mathrm{vN}}[\hat{\rho}], \tag{102}$$

when the particle number $\bar{N}$ is fixed.

**Grand-canonical ensemble**   The grand-canonical Gibbs state is given in Eq. (67). As discussed above, it maximizes the von Neumann entropy for given average values of energy and particle number. This is equivalent to minimizing the grand potential given in Eq. (90). The grand-canonical ensemble is the adequate description when both energy and particles may be exchanged with the environment. The average energy and particle number are then determined by the temperature and chemical potential of the environment respectively.

**Thermodynamic limit and equivalence of ensembles**   The thermodynamic limit is the limit of large systems which formally can be defined as

$$V \to \infty, \qquad\qquad \frac{\bar{E}}{V} = cst. \qquad\qquad \frac{\bar{N}}{V} = cst. \tag{103}$$

This can be achieved by setting $V \propto x$, $\bar{E} \propto x$, $\bar{N} \propto x$, and letting $x \to \infty$. In the thermodynamic limit, all ensembles become equivalent [40] because relative fluctuations around average values vanish. Here, we illustrate this for the canonical ensemble. In this ensemble, the probability for the system to take on a certain energy obeys

$$p(E) \propto e^{-\beta E} \Omega(E, \bar{N}). \tag{104}$$

As it turns out, this function becomes highly peaked in the thermodynamic limit. This can be seen from the relative fluctuations

$$\frac{\langle \hat{H}^2 \rangle - \langle \hat{H} \rangle^2}{\langle \hat{H} \rangle^2} = -\frac{1}{\langle \hat{H} \rangle} \partial_\beta \langle \hat{H} \rangle = \frac{k_B T^2}{\langle \hat{H} \rangle^2} C \propto \frac{1}{x}, \tag{105}$$

where we introduced the heat capacity

$$C = \partial_T \langle \hat{H} \rangle, \tag{106}$$

which quantifies the change of energy upon a change of temperature. The last proportionality in Eq. (105) follows because the heat capacity is an extensive quantity and is thus proportional to $x$ for large systems. Equation (105) implies that in the thermodynamic limit, the relative fluctuations vanish. The probability distribution for the system to take on a given relative energy may then be approximated as a Gaussian distribution (see Ref. [23] for a discussion on higher moments)

$$p(E/\bar{E}) \simeq \frac{\bar{E}}{\sqrt{2\pi k_B T^2 C}} e^{-\frac{(E/\bar{E}-1)^2}{2k_B T^2 C/\bar{E}^2}}, \tag{107}$$

which tends to a Dirac delta distribution $\delta(E/\bar{E} - 1)$ in the thermodynamic limit.

# 3   The laws of thermodynamics

In this section, we discuss how the laws of thermodynamics emerge in a quantum mechanical framework.

## 3.1   The general scenario

The general scenario we consider consists of a system coupled to multiple reservoirs which are in local thermal equilibrium as sketched in Fig. 7. This is described by the Hamiltonian

$$\hat{H}_{\text{tot}}(t) = \hat{H}_S(t) + \sum_\alpha \left( \hat{H}_\alpha + \hat{V}_\alpha \right), \tag{108}$$

where the system is labeled by the subscript $S$ and the reservoirs are labeled by the index $\alpha$. the term $\hat{V}_\alpha$ denotes the coupling between the system and reservoir $\alpha$. Because the system and reservoirs together comprise an isolated system, the time evolution of the total density matrix is given by

$$\hat{\rho}_{\text{tot}}(t) = \hat{U}(t)\hat{\rho}_{\text{tot}}(0)\hat{U}^\dagger(t) \qquad \Leftrightarrow \qquad \partial_t \hat{\rho}_{\text{tot}}(t) = -i[\hat{H}_{\text{tot}}(t), \hat{\rho}_{\text{tot}}(t)], \qquad (109)$$

where the time-evolution operator is given by

$$\hat{U}(t) = \mathcal{T}e^{-i\int_0^t dt' \hat{H}_{\text{tot}}(t')}, \qquad (110)$$

with $\mathcal{T}$ denoting the time-ordering operator (see Sec. 1.1.2) and we allow the system Hamiltonian to be time-dependent while we assume the reservoir Hamiltonians as well as the coupling terms to be time-independent.

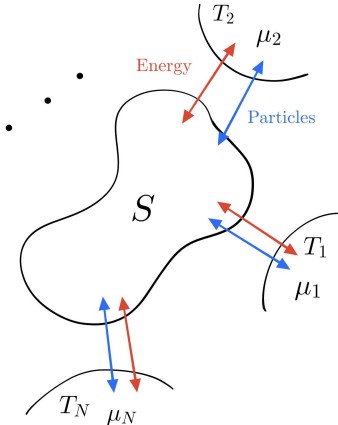

Figure 7: General scenario. A system can exchange both energy and particles with $N$ thermal reservoirs, described by the temperatures $T_\alpha$ and the chemical potentials $\mu_\alpha$.

As seen in Sec. 1.2, the energy flows between the system and the reservoirs are of crucial importance in thermodynamics. This motivates considering the mean energy change of reservoir $\alpha$

$$\partial_t \langle \hat{H}_\alpha \rangle = \text{Tr}\left\{ \hat{H}_\alpha \partial_t \hat{\rho}_{\text{tot}} \right\} = \partial_t \langle \hat{H}_\alpha - \mu_\alpha \hat{N}_\alpha \rangle + \mu_\alpha \partial_t \langle \hat{N}_\alpha \rangle = J_\alpha + P_\alpha. \qquad (111)$$

Here we divided the energy change into a part that we call the heat current $J_\alpha$ and a part that we call the power $P_\alpha$ that enters reservoir $\alpha$. Throughout these notes, the sign convention is such that energy flows are positive when flowing toward the body that their index refers to. To motivate the separation of energy flows into heat and work, we introduce the concept of entropy production in the quantum regime.

## 3.2 Entropy production

The standard extension of entropy to the quantum regime is the von Neumann entropy. However, under unitary time-evolution, the von Neumann entropy is constant and thus

$$\partial_t S_{\text{vN}}[\hat{\rho}_{\text{tot}}] = -\partial_t \text{Tr}\{\hat{\rho}_{\text{tot}}(t)\ln \hat{\rho}_{\text{tot}}(t)\} = 0. \qquad (112)$$

The von Neumann entropy of the total system can thus not tell us anything about how energy flows between the system and the reservoirs. To make progress, we consider an effective description based on local thermal equilibrium [i.e., the reservoirs are described by the Gibbs state given in Eq. (67)]

- True description: $\hat{\rho}_{\text{tot}}(t)$

- Effective description: $\hat{\rho}_{\text{S}}(t) \bigotimes_\alpha \hat{\tau}_\alpha$

where $\hat{\rho}_{\text{S}}(t) = \text{Tr}_{\text{B}}\{\hat{\rho}_{\text{tot}}(t)\}$ and we denote the Gibbs state describing reservoir $\alpha$ as

$$\hat{\tau}_\alpha = \frac{e^{-\beta_\alpha(\hat{H}_\alpha - \mu_\alpha \hat{N}_\alpha)}}{Z_\alpha}. \tag{113}$$

The effective description thus contains all information on the system but neglects any changes to the reservoirs as well as the correlations that may build up between system and reservoirs. Such an effective description is often the best one can do in an experiment, where one might have control over the microscopic degrees of freedom of the system only.

We now consider the quantum relative entropy between the true state $\hat{\rho}_{\text{tot}}(t)$ and our effective description

$$S[\hat{\rho}_{\text{tot}}\|\hat{\rho}_{\text{S}} \bigotimes_\alpha \hat{\tau}_\alpha] = S_{\text{vN}}[\hat{\rho}_{\text{S}}] + \sum_\alpha \beta_\alpha \langle \hat{H}_\alpha - \mu_\alpha \hat{N}_\alpha \rangle + \sum_\alpha \ln Z_\alpha - S_{\text{vN}}[\hat{\rho}_{\text{tot}}], \tag{114}$$

where averages are with respect to the total state $\langle \hat{X} \rangle = \text{Tr}\{\hat{X}\hat{\rho}_{\text{tot}}(t)\}$ and we used $\ln \hat{\tau}_\alpha = -\beta_\alpha(\hat{H}_\alpha - \mu_\alpha \hat{N}_\alpha) - \ln Z_\alpha$ as well as $\ln(\hat{A} \otimes \hat{B}) = \ln(\hat{A}) \otimes \mathbb{1} + \mathbb{1} \otimes \ln(\hat{B})$. As discussed in Sec. 1.3, Eq. (114) may be interpreted as the information that is lost due to our effective description. We now introduce the *entropy production rate* [41]

$$\dot{\Sigma} = k_{\text{B}} \partial_t S[\hat{\rho}_{\text{tot}}\|\hat{\rho}_{\text{S}} \bigotimes_\alpha \hat{\tau}_\alpha] = k_{\text{B}} \partial_t S_{\text{vN}}[\hat{\rho}_{\text{S}}] + \sum_\alpha \frac{J_\alpha}{T_\alpha}. \tag{115}$$

The entropy production rate can thus be interpreted as the rate at which information is lost by our local equilibrium description, due to the buildup of correlations between system and environment as well as changes to the reservoirs. Note that it is not guaranteed to be positive. Finite size effects as well as non-Markovian dynamics can result in a negative entropy production rate (a backflow of information from the reservoirs). However, as we will see later, for infinitely large and memoryless reservoirs, the entropy production rate is ensured to be positive at all times as information is irretrievably lost when one can only access the system alone.

Note that Eq. (115) motivates the interpretation of $J_\alpha$ as a heat flow, such that the entropy production associated to reservoir $\alpha$ is given by the usual expression for reservoirs which remain in thermal equilibrium. Interestingly, we may refine our effective description by using time-dependent temperatures and chemical potentials [42, 43]

$$\tau_\alpha(t) = \frac{e^{-\beta_\alpha(t)(\hat{H}_\alpha - \mu_\alpha(t)\hat{N}_\alpha)}}{Z_\alpha}, \tag{116}$$

such that

$$\text{Tr}\{\hat{H}_\alpha \hat{\rho}_{\text{tot}}(t)\} = \text{Tr}\{\hat{H}_\alpha \hat{\tau}_\alpha(t)\}, \qquad \text{Tr}\{\hat{N}_\alpha \hat{\rho}_{\text{tot}}(t)\} = \text{Tr}\{\hat{N}_\alpha \hat{\tau}_\alpha(t)\}. \tag{117}$$

In this case, one may show that

$$k_{\text{B}} \partial_t S_{\text{vN}}[\hat{\tau}_\alpha(t)] = \frac{J_\alpha(t)}{T_\alpha(t)}, \tag{118}$$

and the entropy production rate reduces to

$$\dot{\Sigma} = k_{\text{B}} \partial_t S[\hat{\rho}_{\text{tot}}\|\hat{\rho}_{\text{S}} \bigotimes_\alpha \hat{\tau}_\alpha(t)] = k_{\text{B}} \partial_t S_{\text{vN}}[\hat{\rho}_{\text{S}}] + k_{\text{B}} \sum_\alpha \partial_t S_{\text{vN}}[\hat{\tau}_\alpha(t)]$$

$$= k_{\text{B}} \partial_t S_{\text{vN}}[\hat{\rho}_{\text{S}}] + \sum_\alpha \frac{J_\alpha(t)}{T_\alpha(t)}. \tag{119}$$

## 3.3 The first law of thermodynamics

To derive the first law of thermodynamics, we consider the change in the average energy of system and reservoirs

$$
\begin{aligned}
\partial_t \langle \hat{H}_{\mathrm{tot}}(t) \rangle &= \mathrm{Tr}\left\{ [\partial_t \hat{H}_{\mathrm{tot}}(t)] \hat{\rho}_{\mathrm{tot}} \right\} + \mathrm{Tr}\left\{ \hat{H}_{\mathrm{tot}}(t) \partial_t \hat{\rho}_{\mathrm{tot}} \right\} \\
&= \partial_t \langle \hat{H}_{\mathrm{S}}(t) \rangle + \partial_t \sum_\alpha \langle \hat{H}_\alpha \rangle + \partial_t \sum_\alpha \langle \hat{V}_\alpha \rangle.
\end{aligned}
\tag{120}
$$

Using Eq. (72), the last term on the first line can be shown to vanish. With the help of the last equation and Eq. (111), we may then derive the first law of thermodynamics (note that the energy flows are defined to be positive when they enter the location corresponding to their index)

$$
\partial_t \langle \hat{H}_{\mathrm{S}}(t) \rangle = P_{\mathrm{S}}(t) - \sum_\alpha [J_\alpha(t) + P_\alpha(t)] - \partial_t \sum_\alpha \langle \hat{V}_\alpha \rangle, \qquad P_{\mathrm{S}} \equiv \langle \partial_t \hat{H}_{\mathrm{S}} \rangle, \tag{121}
$$

where $P_{\mathrm{S}}$ denotes the power entering the system due to some external classical drive that renders $\hat{H}_{\mathrm{S}}$ time-dependent. The term due to the coupling energy $\langle \hat{V}_\alpha \rangle$ can be neglected when the coupling is weak, which is a common assumption for open quantum systems. Relaxing this assumption and considering the thermodynamics of systems that are strongly coupled to the environment is an exciting ongoing avenue of research [44–47].

## 3.4 The second law of thermodynamics

Let us consider an initial state which is a product state of the form

$$
\hat{\rho}_{\mathrm{tot}}(0) = \hat{\rho}_{\mathrm{S}}(0) \bigotimes_\alpha \hat{\tau}_\alpha, \tag{122}
$$

i.e., we assume our effective description to be exact at $t = 0$. In this case, Eq. (114) can be written as

$$
\Sigma \equiv k_{\mathrm{B}} S\left[ \hat{\rho}_{\mathrm{tot}}(t) \| \hat{\rho}_{\mathrm{S}}(t) \bigotimes_\alpha \hat{\tau}_\alpha \right] = k_{\mathrm{B}} S_{\mathrm{vN}}[\hat{\rho}_{\mathrm{S}}(t)] - k_{\mathrm{B}} S_{\mathrm{vN}}[\hat{\rho}_{\mathrm{S}}(0)] + \sum_\alpha \frac{Q_\alpha}{T_\alpha}, \tag{123}
$$

where the heat is defined as

$$
Q_\alpha = \mathrm{Tr}\left\{ (\hat{H}_\alpha - \mu_\alpha \hat{N}_\alpha) \hat{\rho}_{\mathrm{tot}}(t) \right\} - \mathrm{Tr}\left\{ (\hat{H}_\alpha - \mu_\alpha \hat{N}_\alpha) \hat{\rho}_{\mathrm{tot}}(0) \right\}, \tag{124}
$$

and we used that $S[\hat{\rho}_{\mathrm{tot}}(0) \| \hat{\rho}_{\mathrm{S}}(0) \bigotimes_\alpha \hat{\tau}_\alpha] = 0$. Since it is expressed as a quantum relative entropy, we have

$$
\Sigma \geq 0. \tag{125}
$$

From an information point of view, this inequality tells us that if our effective description is true at $t = 0$, then it can only be worse at later times. To understand how our description becomes worse, we follow Refs. [41, 48] and write the entropy production as

$$
\Sigma = I_{\mathrm{MI}}[\mathrm{S}; \mathrm{B}] + S[\hat{\rho}_{\mathrm{B}}(t) \| \bigotimes_\alpha \hat{\tau}_\alpha], \tag{126}
$$

where we introduced the reduced state of the environment $\hat{\rho}_{\mathrm{B}}(t) = \mathrm{Tr}_{\mathrm{S}}\{\hat{\rho}_{\mathrm{tot}}(t)\}$ and the mutual information between system and environment is given by [c.f. (65)]

$$
I_{\mathrm{MI}}[\mathrm{S}; \mathrm{B}] = S_{\mathrm{vN}}[\hat{\rho}_{\mathrm{S}}(t)] + S_{\mathrm{vN}}[\hat{\rho}_{\mathrm{B}}(t)] - S_{\mathrm{vN}}[\hat{\rho}_{\mathrm{tot}}(t)]. \tag{127}
$$

The first term in Eq. (126) describes the correlations between the system and the environment while the second term describes the departure of the environment from the initial product of

Gibbs states. As was recently shown, the departure of the environment from its initial state provides the dominant contribution to the entropy production [48]. This can be seen from the upper bound of the mutual information $I_{\mathrm{MI}}[\mathbf{S};\mathbf{B}] \leq 2d_{\mathbf{S}}$ [c.f. Eq. (66)], where $d_{\mathbf{S}}$ denotes the dimension of the Hilbert space of the system. In the long-time limit, or under steady state conditions, the mutual information therefore saturates and can no longer contribute to $\Sigma$. Any entropy production due to persisting heat currents, see Eq. (123), can thus be associated with the environment departing further and further from the initial state with well defined temperatures and chemical potentials.

As mentioned above, the entropy production rate $\dot{\Sigma}$ is not always guaranteed to be positive (i.e., $\Sigma$ is not necessarily a monotonously increasing function of time). However, at small times, Eq. (125) ensures that the entropy production rate is also positive. Furthermore, as we will see in the next section, the entropy production is positive for infinitely large and memory-less reservoirs which couple weakly to the system. We note that with a time-dependent effective description, see Eq. (116), Eqs. (123) and (125) still hold.

## 3.5 The zeroth law of thermodynamics

For completeness, we will also briefly mention the zeroth and the third law of thermodynamics, although they will play a less important role throughout these notes. The zeroth law is usually phrased as: *If two systems are both in equilibrium with a third one, then they are in equilibrium with each other.* Being equipped with definitions for temperature and chemical potential, then the zeroth law implies: *Systems in equilibrium with each other have the same temperature and chemical potential.*

When a small system is coupled to a large equilibrium reservoir, then the reduced system state is expected to tend to [47]

$$\lim_{t\to\infty} \hat{\rho}_{\mathbf{S}}(t) = \frac{\mathrm{Tr}_{\mathbf{B}}\left\{e^{-\beta(\hat{H}_{\mathrm{tot}} - \mu\hat{N}_{\mathrm{tot}})}\right\}}{\mathrm{Tr}\left\{e^{-\beta(\hat{H}_{\mathrm{tot}} - \mu\hat{N}_{\mathrm{tot}})}\right\}} \xrightarrow[\mathrm{coupling}]{\mathrm{weak}} \frac{e^{\beta(\hat{H}_{\mathbf{S}} - \mu\hat{N}_{\mathbf{S}})}}{Z}. \tag{128}$$

As expected, in equilibrium, the system is thus characterized by the same temperature and chemical potential as the reservoir. We note that equilibrium may only be reached if $\hat{H}_{\mathbf{tot}}$ is time-independent and if there is only a single temperature and a single chemical potential characterizing the environment. The exact range of validity of the first equality in Eq. (128) is a subject of ongoing research [47].

## 3.6 The third law of thermodynamics

The formulation of the third law of thermodynamics that we consider here is known as Nernst's unattainability principle [49]. In its modern formulation, it reads: *It is impossible to cool a system to its ground state (or create a pure state) without diverging resources.* These resources may be time, energy (in the form of work), or control complexity [50]. When a diverging amount of work is available, one may cool a system to the ground state by increasing the energy of all excited states by an infinite amount. Then, the equilibrium state at any finite temperature reduces to the ground state which can thus be obtained by equilibration with any thermal reservoir. When an infinite amount of time is available, this process may be performed reversibly, reducing the work cost to a finite amount. Loosely speaking, infinite control complexity allows one to parallelize this cooling process and reach the ground state using only a finite amount of time and work [50].

Existing proofs of the third law of thermodynamics for quantum systems [50, 51] use the framework of the resource theory of thermodynamics [13] and are not directly applicable to the scenario considered in these notes.

# 4 Markovian master equations

In this section, we consider Markovian master equations as a description for the reduced state of the system $\hat{\rho}_S$. Markovianity implies that no memory effects of the environment are taken into account. For instance, a particle that is emitted from the system will not be reabsorbed by the system before losing all memory of the emission event. In principle, memory effects are always present but if the coupling between system and environment is weak, these effects can often safely be ignored. There are numerous references that discuss Markovian master equations going substantially beyond these notes, see for instance Refs. [52–56].

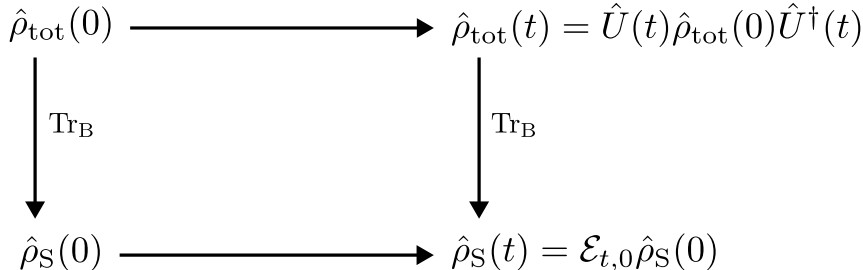

Figure 8: Time evolution of total and reduced density matrices. The total density matrix evolves unitarily, according to Eq. (109). The reduced density matrix at time $t$ can either be obtained by tracing out the environment from $\hat{\rho}_{tot}(t)$, or by evolving $\hat{\rho}_S(0)$ by the universal dynamical map $\mathcal{E}_{t,0}$

As illustrated in Fig. 8, the time-evolution of the reduced density matrix can be described by a *universal dynamical map* (UDM). A UDM $\mathcal{E}$ is a linear map which transforms a density matrix into another density matrix. Furthermore, it is independent of the density matrix it acts upon. In its most general form, a UDM can be written as

$$\mathcal{E}\hat{\rho} = \sum_j \hat{K}_j \hat{\rho} \hat{K}_j^{\dagger}, \qquad \sum_j \hat{K}_j^{\dagger} \hat{K}_j = \mathbb{1},$$

where the operators $\hat{K}_j$ are called Kraus operators.

We say that a system obeys Markovian time-evolution if it is described by a *divisible* UDM, i.e.,

$$\hat{\rho}(t) = \mathcal{E}_{t,t_0}\hat{\rho} = \mathcal{E}_{t,t_1}\mathcal{E}_{t_1,t_0}\hat{\rho},$$

For any intermediate time $t_0 < t_1 < t$. Furthermore, we note that a differential equation is a Markovian master equation (i.e., results in Markovian time-evolution of a density matrix) if and only if it can be written in the form

$$\begin{aligned}
\partial_t \hat{\rho}(t) &= -i[\hat{H}(t), \hat{\rho}(t)] + \sum_k \gamma_k(t)\left[\hat{L}_k(t)\hat{\rho}(t)\hat{L}_k^{\dagger}(t) - \frac{1}{2}\{\hat{L}_k^{\dagger}(t)\hat{L}_k(t), \hat{\rho}(t)\}\right] \\
&= -i[\hat{H}(t), \hat{\rho}(t)] + \sum_k \gamma_k(t)\mathcal{D}[\hat{L}_k(t)]\hat{\rho}(t),
\end{aligned} \tag{129}$$

where $\hat{H}(t)$ is Hermitian, $\gamma_k(t) \geq 0$, and the operators $\hat{L}_k$ are referred to as Lindblad jump operators. For a proof, see Ref. [53]. This form of the master equation is also called GKLS form, after Gorini, Kosakowski, Sudarshan [57], and Linblad [58] who considered the time-independent case. In the rest of this chapter, we provide detailed derivations for Markovian master equations that describe our general scenario introduced in Sec. 3.1.

## 4.1 Nakajima-Zwanzig superoperators

To derive Markovian master equations, we use the Nakajima-Zwanzig projection operator approach [59, 60] following Ref. [61]. To this end, we introduce the superoperators

$$\mathcal{P}\hat{\rho}_{\text{tot}} = \text{Tr}_{\text{B}}\{\hat{\rho}_{\text{tot}}\}\bigotimes_{\alpha}\hat{\tau}_{\alpha} = \hat{\rho}_{\text{S}}\bigotimes_{\alpha}\hat{\tau}_{\alpha}, \qquad \mathcal{Q} = \mathbb{1} - \mathcal{P}. \tag{130}$$

Note that these are projectors as $\mathcal{P}^2 = \mathcal{P}$. Further, note that we are interested in the time-evolution of $\mathcal{P}\hat{\rho}_{\text{tot}}(t)$, which provides us with an effective description of the form discussed in Sec. 3.2. We consider the general scenario discussed in Sec. 3.1, i.e., the Hamiltonian is given by Eq. (108), but we move to an interaction picture (which we denote by a tilde instead of a hat)

$$\tilde{\rho}_{\text{tot}}(t) = \hat{U}_0^{\dagger}(t)\hat{\rho}_{\text{tot}}(t)\hat{U}_0(t), \tag{131}$$

determined by the unitary operator

$$\hat{U}_0(t) = \mathcal{T}e^{-i\int_0^t dt'[\hat{H}_{\text{S}}(t')+\sum_{\alpha}\hat{H}_{\alpha}]}, \tag{132}$$

with $\mathcal{T}$ denoting time-ordering. In the interaction picture, the time-evolution of the total density matrix is determined by

$$\partial_t\tilde{\rho}_{\text{tot}}(t) = -i\sum_{\alpha}\left[\tilde{V}_{\alpha}(t),\tilde{\rho}_{\text{tot}}(t)\right] = \mathcal{V}(t)\tilde{\rho}_{\text{tot}}(t), \tag{133}$$

where we used Eq. (109) as well as

$$\partial_t\hat{U}_0(t) = -i\left[\hat{H}_{\text{S}}(t)+\sum_{\alpha}\hat{H}_{\alpha}\right]\hat{U}_0(t), \qquad \partial_t\hat{U}_0^{\dagger}(t) = i\hat{U}_0^{\dagger}(t)\left[\hat{H}_{\text{S}}(t)+\sum_{\alpha}\hat{H}_{\alpha}\right], \tag{134}$$

and the coupling Hamiltonian in the interaction picture is given by $\tilde{V}_{\alpha}(t) = \hat{U}_0^{\dagger}(t)\hat{V}_{\alpha}\hat{U}_0(t)$, in analogy to Eq. (131). Finally, we have expressed the commutator in Eq. (133) with the help of the superoperator $\mathcal{V}$. In the following, we will assume $\mathcal{P}\mathcal{V}(t)\mathcal{P} = 0$. This is not a restriction as it can always be ensured by adding some terms to $\hat{H}_{\text{S}}$ and subtracting them from $\hat{V}_{\alpha}$, as we will see below. We can then write

$$\partial_t\mathcal{P}\tilde{\rho}_{\text{tot}}(t) = \mathcal{P}\mathcal{V}(t)\tilde{\rho}_{\text{tot}}(t) = \mathcal{P}\mathcal{V}(t)\mathcal{Q}\tilde{\rho}_{\text{tot}}(t),$$
$$\partial_t\mathcal{Q}\tilde{\rho}_{\text{tot}}(t) = \mathcal{Q}\mathcal{V}(t)\tilde{\rho}_{\text{tot}}(t) = \mathcal{Q}\mathcal{V}(t)\mathcal{P}\tilde{\rho}_{\text{tot}}(t) + \mathcal{Q}\mathcal{V}(t)\mathcal{Q}\tilde{\rho}_{\text{tot}}(t), \tag{135}$$

where we used $\mathcal{P} + \mathcal{Q} = \mathbb{1}$. The formal solution to the second equation is given by

$$\mathcal{Q}\tilde{\rho}_{\text{tot}}(t) = \mathcal{G}(t,0)\mathcal{Q}\tilde{\rho}_{\text{tot}}(0) + \int_0^t ds\,\mathcal{G}(t,s)\mathcal{Q}\mathcal{V}(s)\mathcal{P}\tilde{\rho}_{\text{tot}}(s), \tag{136}$$

where we introduced the propagator

$$\mathcal{G}(t,s) = \mathcal{T}e^{\int_s^t \mathcal{Q}\mathcal{V}(\tau)d\tau}. \tag{137}$$

We now assume factorizing initial conditions

$$\tilde{\rho}_{\text{tot}}(0) = \hat{\rho}_{S}(0)\bigotimes_{\alpha}\hat{\tau}_{\alpha}, \tag{138}$$

such that $\mathcal{P}\tilde{\rho}_{\text{tot}}(0) = \tilde{\rho}_{\text{tot}}(0)$ and $\mathcal{Q}\tilde{\rho}_{\text{tot}}(0) = 0$. Inserting Eq. (136) into Eq. (135), we find

$$\partial_t\mathcal{P}\tilde{\rho}_{\text{tot}}(t) = \int_0^t ds\,\mathcal{P}\mathcal{V}(t)\mathcal{G}(t,s)\mathcal{Q}\mathcal{V}(s)\mathcal{P}\tilde{\rho}_{\text{tot}}(s). \tag{139}$$

We note that this expression is still exact (for the given initial conditions). Since it explicitly depends on $\mathcal{P}\tilde{\rho}_{\text{tot}}$ at previous times, it contains memory effects and does not constitute a Markovian master equation.

## 4.2 Born-Markov approximations

We now make a weak coupling approximation. If the coupling between system and reservoirs are proportional to $\hat{V}_\alpha \propto r$, with $r \ll 1$, the propagator in Eq. (137) obeys $\mathcal{G}(t,s) = 1 + \mathcal{O}(r)$, resulting in

$$\partial_t \mathcal{P} \tilde{\rho}_{\text{tot}}(t) = \mathcal{P} \int_0^t ds\, \mathcal{V}(t)\mathcal{V}(s)\mathcal{P}\tilde{\rho}_{\text{tot}}(s) + \mathcal{O}(r^3), \tag{140}$$

where we again used $\mathcal{P}\mathcal{V}(t)\mathcal{P} = 0$. The last equation implies

$$\partial_t \tilde{\rho}_S(t) = -\int_0^t ds \sum_\alpha \text{Tr}_B\left\{\left[\tilde{V}_\alpha(t),\left[\tilde{V}_\alpha(t-s),\tilde{\rho}_S(t-s)\bigotimes_\alpha \hat{\tau}_\alpha\right]\right]\right\}, \tag{141}$$

where we substituted $s \to t-s$ and we made use of $\text{Tr}_B\{\tilde{V}_\delta(t)\tilde{V}_\gamma(s)\tilde{\rho}_S(s)\otimes_\alpha \hat{\tau}_\alpha\} = 0$ for $\delta \neq \gamma$. This is similar to the assumption $\mathcal{P}\mathcal{V}(t)\mathcal{P} = 0$ and can always be ensured by an appropriate redefinition of the terms in the Hamiltonian as shown below. We note that Eq. (141) is often obtained by assuming that $\tilde{\rho}_{\text{tot}}(t) = \tilde{\rho}_S(t)\otimes_\alpha \hat{\tau}_\alpha$ at all times, the so-called Born approximation. Here we do not make such an assumption. In agreement with the discussion in the previous section, we consider $\tilde{\rho}_S(t) \otimes_\alpha \hat{\tau}_\alpha$ to be an effective description, which only keeps track of the system and neglects changes in the environment state as well as correlations between system and environment.

In addition to the weak coupling approximation, we now make a Markov approximation. To this end, we assume that the integrand in Eq. (141) decays on a time-scale $\tau_B$ (the bath-correlation time, more on this below). If this time-scale is short enough, which is the case for large, memory-less environments, we can assume $\tilde{\rho}_S(t-s)$ to approximately remain constant and replace its time-argument in Eq. (141) by $t$. Furthermore, using the same argumentation, we can extend the integral to infinity obtaining

$$\partial_t \tilde{\rho}_S(t) = -\int_0^\infty ds \sum_\alpha \text{Tr}_B\left\{\left[\tilde{V}_\alpha(t),\left[\tilde{V}_\alpha(t-s),\tilde{\rho}_S(t)\bigotimes_\alpha \hat{\tau}_\alpha\right]\right]\right\}. \tag{142}$$

This equation is Markovian, i.e., it is local in time and does not depend explicitly on the initial conditions. However, it is not in GKLS form and does not in general preserve the positivity of the density matrix, i.e., eigenvalues may become negative. The approximations that result in Eq. (142) are usually called the *Born-Markov* approximations. For a more formal application of these approximations, see Refs. [62, 63]. Note that under the Born-Markov approximations, the effect induced by different reservoirs is additive.

To make progress, we write the coupling Hamiltonian in the general form

$$\hat{V}_\alpha = \sum_k \hat{S}_{\alpha,k} \otimes \hat{B}_{\alpha,k} = \sum_k \hat{S}_{\alpha,k}^\dagger \otimes \hat{B}_{\alpha,k}^\dagger, \tag{143}$$

where we used the Hermiticity of $\hat{V}_\alpha$ in the second equality. We note that the operators $\hat{S}_{\alpha,k}$ and $\hat{B}_{\alpha,k}$ are not necessarily Hermitian. Inserting Eq. (143) into Eq. (142), we find after some algebra

$$\partial_t \tilde{\rho}_S(t) = \sum_\alpha \sum_{k,k'} \int_0^\infty ds \left\{ C_{k,k'}^\alpha(s)\left[\tilde{S}_{\alpha,k'}(t-s)\tilde{\rho}_S(t)\tilde{S}_{\alpha,k}^\dagger(t) - \tilde{S}_{\alpha,k}^\dagger(t)\tilde{S}_{\alpha,k'}(t-s)\tilde{\rho}_S(t)\right] \right.$$
$$\left. + C_{k,k'}^\alpha(-s)\left[\tilde{S}_{\alpha,k'}(t)\tilde{\rho}_S(t)\tilde{S}_{\alpha,k}^\dagger(t-s) - \tilde{\rho}_S(t)\tilde{S}_{\alpha,k}^\dagger(t-s)\tilde{S}_{\alpha,k'}(t)\right]\right\}, \tag{144}$$

where we introduced the bath-correlation functions

$$C_{k,k'}^\alpha(s) = \text{Tr}\left\{\tilde{B}_{\alpha,k}^\dagger(s)\hat{B}_{\alpha,k'}\hat{\tau}_\alpha\right\}, \tag{145}$$

and we used $[C_{k,k'}^{\alpha}(s)]^* = C_{k',k}^{\alpha}(-s)$. These bath-correlation functions are usually peaked around $s = 0$ and decay over the time-scale $\tau_{\mathrm{B}}$ (indeed, this is how $\tau_{\mathrm{B}}$ is defined). If this time-scale is short, the integrand in Eq. (144) decays quickly and the Markov assumption performed above is justified. Note that it is important that this approximation is made in the interaction picture, where $\tilde{\rho}_{\mathrm{S}}$ varies slowly (in the Schrödinger picture, $\hat{\rho}_{\mathrm{S}}$ tends to oscillate with frequencies given by the differences of the eigenvalues of $\hat{H}_{\mathrm{S}}$).

While Eq. (144) is in general still not in GKLS form, it happens to reduce to GKLS form when $C_{k,k'}^{\alpha}(s) \propto \delta_{k,k'}$ and

$$\left[\hat{S}_{\alpha,k}, \hat{H}_{\mathrm{S}}\right] = \omega_{\alpha,k}\hat{S}_{\alpha,k} \qquad \tilde{S}_{\alpha,k}(t) = e^{-i\omega_{\alpha,k}t}\hat{S}_{\alpha,k}, \tag{146}$$

which implies that $\hat{S}_{\alpha,k}$ are ladder operators for the Hamiltonian $\hat{H}_{\mathrm{S}}$, removing the energy $\omega_{\alpha,k}$ from the system. Before we consider such an example in detail, we briefly justify two assumptions made in the derivation above. To this end, we note that if

$$\mathrm{Tr}\left\{\hat{B}_{\alpha,k}\hat{\tau}_{\alpha}\right\} = 0, \tag{147}$$

then it is straightforward to show that

$$\mathcal{P}\mathcal{V}(t)\mathcal{P} = 0, \qquad \mathrm{Tr}\left\{\tilde{V}_{\delta}(t)\tilde{V}_{\gamma}(s)\tilde{\rho}_{\mathrm{S}}(t) \otimes_{\alpha} \hat{\tau}_{\alpha}\right\} = 0, \qquad \forall \delta \neq \gamma. \tag{148}$$

In case Eq. (147) is not true, we may define

$$\hat{B}_{\alpha,k}' = \hat{B}_{\alpha,k} - \mathrm{Tr}\left\{\hat{B}_{\alpha,k}\hat{\tau}_{\alpha}\right\}, \tag{149}$$

which do fulfill $\mathrm{Tr}\{\hat{B}_{\alpha,k}'\hat{\tau}_{\alpha}\} = 0$. Defining

$$\hat{V}_{\alpha}' = \sum_{k}\hat{S}_{\alpha,k} \otimes \hat{B}_{\alpha,k}', \qquad \hat{H}_{\mathrm{S}}'(t) = H_{\mathrm{S}}(t) + \sum_{\alpha,k}\mathrm{Tr}\left\{\hat{B}_{\alpha,k}\hat{\tau}_{\alpha}\right\}\hat{S}_{\alpha,k}, \tag{150}$$

we have $\hat{H}_{\mathrm{tot}}(t) = H_{\mathrm{S}}'(t) + \sum_{\alpha}(\hat{H}_{\alpha} + \hat{V}_{\alpha}')$, i.e., the same total Hamiltonian but now in a form such that Eq. (147) holds, ensuring our assumptions given in Eq. (148).

### 4.3 Example: equilibration of a quantum dot

We now consider an example provided by a spinless, single-level quantum dot tunnel-coupled to a single fermionic reservoir, see Fig. 9. In this case, Eq. (144) happens to already be in GKLS form. The Hamiltonian of system and environment is then given by

$$\hat{H}_{\mathrm{tot}} = \hat{H}_{\mathrm{S}} + \hat{H}_{\mathrm{B}} + \hat{V}, \tag{151}$$

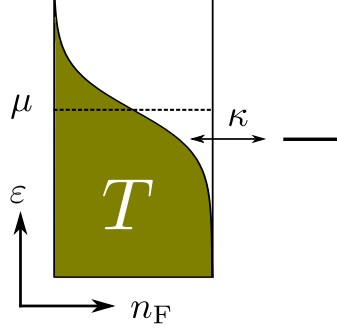

Figure 9: Spinless, single-level quantum dot tunnel-coupled to a fermionic reservoir. the reservoir is characterized by a temperature $T$ and a chemical potential $\mu$. The coupling between the reservoir and the quantum dot is given by $\kappa$. In the steady state, the quantum dot is in equilibrium with the reservoir.

with

$$\hat{H}_\text{S} = \varepsilon_\text{d}\hat{d}^\dagger\hat{d}, \qquad \hat{H}_\text{B} = \sum_q \varepsilon_q \hat{c}_q^\dagger \hat{c}_q, \qquad \hat{V} = \sum_q \left(g_q \hat{d}\hat{c}_q^\dagger - g_q^* \hat{d}^\dagger \hat{c}_q\right). \tag{152}$$

Here the reservoir is modeled as a collection of non-interacting fermions and the coupling Hamiltonian describes tunneling of single electrons between the system and the reservoir (the minus sign arises from the fermionic anti-commutation relations).

We note that for fermions, there is no tensor product structure because operators on the environment and on the system may *anti*-commute. Strictly speaking, the derivation in the last section is thus not valid. However, using a Jordan-Wigner transform, one can map the fermionic system onto a spin system where such a tensor-product structure is provided [54]. After tracing out the reservoir, the spin operators can then be replaced by fermionic operators again. For the tunneling Hamiltonian that we often use to describe the system-environment coupling for fermions, this procedure is equivalent to replacing the tensor product by the usual product between fermionic operators in the derivation.

### 4.3.1  The master equation

Comparing the coupling Hamiltonian to Eq. (143), we may write $\hat{V} = \hat{S}_0\hat{B}_0 + \hat{S}_1\hat{B}_1$ with

$$\hat{S}_0 = \hat{d}, \qquad \hat{S}_1 = \hat{d}^\dagger, \qquad \hat{B}_0 = \sum_q g_q \hat{c}_q^\dagger, \qquad \hat{B}_1 = -\sum_q g_q^* \hat{c}_q. \tag{153}$$

We further find

$$[\hat{d}, \hat{H}_\text{S}] = \varepsilon_\text{d}\hat{d} \qquad \Rightarrow \qquad \tilde{S}_0(t) = e^{-i\varepsilon_\text{d}t}\hat{d}, \quad \tilde{S}_1(t) = e^{i\varepsilon_\text{d}t}\hat{d}^\dagger = \tilde{S}_0^\dagger(t), \tag{154}$$

and similarly

$$\tilde{B}_0(t) = \sum_q g_q e^{i\varepsilon_q t}\hat{c}_q^\dagger, \qquad\qquad \tilde{B}_1(t) = -\sum_q g_q^* e^{-i\varepsilon_q t}\hat{c}_q. \tag{155}$$

These expressions result in the bath-correlation functions

$$\begin{aligned}
C_{0,0}(s) &= \sum_q |g_q|^2 e^{-i\varepsilon_q s}\,\text{Tr}\left\{\hat{c}_q\hat{c}_q^\dagger\hat{\tau}_\text{B}\right\} = \sum_q |g_q|^2 e^{-i\varepsilon_q s}[1 - n_\text{F}(\varepsilon_q)] \\
&= \int_{-\infty}^{\infty} d\omega\, e^{-i\omega s}\rho(\omega)[1 - n_\text{F}(\omega)], \\
C_{1,1}(s) &= \sum_q |g_q|^2 e^{i\varepsilon_q s}\,\text{Tr}\left\{\hat{c}_q^\dagger\hat{c}_q\hat{\tau}_\text{B}\right\} = \sum_q |g_q|^2 e^{i\varepsilon_q s} n_\text{F}(\varepsilon_q) \\
&= \int_{-\infty}^{\infty} d\omega\, e^{i\omega s}\rho(\omega)n_\text{F}(\omega),
\end{aligned} \tag{156}$$

where we introduced the Fermi-Dirac occupation

$$n_\text{F}(\omega) = \frac{1}{e^{\frac{\omega - \mu}{k_\text{B}T}} + 1}, \tag{157}$$

and made use of $\text{Tr}\{\hat{c}_q^\dagger\hat{c}_{q'}\hat{\tau}_\text{B}\} = 0$ for $q \neq q'$. Furthermore, it is easy to show that $C_{0,1}(s) = C_{1,0}(s) = 0$ since $\text{Tr}\{(\hat{c}_q^\dagger)^2\hat{\tau}_\text{B}\} = \text{Tr}\{(\hat{c}_q)^2\hat{\tau}_\text{B}\} = 0$. Finally, in Eq. (156) we introduced the spectral density

$$\rho(\omega) = \sum_q |g_q|^2 \delta(\varepsilon_q - \omega), \tag{158}$$

which will be treated as a continuous function. This is justified whenever the summands in Eq. (156) are sufficiently smooth such that the sums can be thought of as Riemann sums that approximate integrals of smooth functions.

Using Eq. (154), we may re-write the master equation in Eq. (144) as

$$
\begin{aligned}
\partial_t \tilde{\rho}_S(t) = \int_{-\infty}^{\infty} ds \, \{ C_{0,0}(s) e^{i\varepsilon_d s} \mathcal{D}[\hat{d}] \tilde{\rho}_S(t) + C_{1,1}(s) e^{-i\varepsilon_d s} \mathcal{D}[\hat{d}^{\dagger}] \tilde{\rho}_S(t) \} \\
- \frac{1}{2} \int_{-\infty}^{\infty} ds \, \mathrm{sign}(s) [ C_{0,0}(s) e^{i\varepsilon_d s} - C_{1,1}(s) e^{-i\varepsilon_d s} ] [\hat{d}^{\dagger}\hat{d}, \tilde{\rho}_S(t)],
\end{aligned}
\tag{159}
$$

where

$$
\mathcal{D}[\hat{A}]\hat{\rho} = \hat{A}\hat{\rho}\hat{A}^{\dagger} - \frac{1}{2}\{\hat{A}^{\dagger}\hat{A}, \hat{\rho}\}.
\tag{160}
$$

With the help of the bath-correlation functions given in Eq. (156), we may evaluate the integrals over $s$ to find the transition rates

$$
\begin{aligned}
\int_{-\infty}^{\infty} ds \, C_{0,0}(s) e^{i\varepsilon_d s} = \int_{-\infty}^{\infty} d\omega \int_{-\infty}^{\infty} ds \, e^{-i(\omega-\varepsilon_d)s} \rho(\omega)[1-n_F(\omega)] = \kappa[1-n_F(\varepsilon_d)], \\
\int_{-\infty}^{\infty} ds \, C_{1,1}(s) e^{-i\varepsilon_d s} = \int_{-\infty}^{\infty} d\omega \int_{-\infty}^{\infty} ds \, e^{i(\omega-\varepsilon_d)s} \rho(\omega) n_F(\omega) = \kappa n_F(\varepsilon_d),
\end{aligned}
\tag{161}
$$

where we used

$$
\int_{-\infty}^{\infty} ds \, e^{-i(\omega-\varepsilon_d)s} = 2\pi\delta(\omega-\varepsilon_d),
\tag{162}
$$

and introduced $\kappa \equiv 2\pi\rho(\varepsilon_d)$.

We furthermore find the so-called Lamb shift

$$
\begin{aligned}
-\frac{1}{2} \int_{-\infty}^{\infty} ds \, \mathrm{sign}(s) [ C_{0,0}(s) e^{i\varepsilon_d s} - C_{1,1}(s) e^{-i\varepsilon_d s} ] \\
= -i\mathrm{Im} \int_0^{\infty} ds \, [ C_{0,0}(s) e^{i\varepsilon_d s} - C_{1,1}(s) e^{-i\varepsilon_d s} ] = iP \int_{-\infty}^{\infty} d\omega \, \frac{\rho(\omega)}{\omega-\varepsilon_d},
\end{aligned}
\tag{163}
$$

where we made use of $C_{j,j}^*(s) = C_{j,j}(-s)$ as well as the identity

$$
\lim_{t\to\infty} \int_0^t ds \, e^{i\omega s} = \pi\delta(\omega) + iP\left(\frac{1}{\omega}\right),
\tag{164}
$$

with $P$ denoting the Cauchy principal value [i.e., $P(1/\omega)$ is equal to $1/\omega$ except at $\omega = 0$, where the principal value vanishes].

Finally, inserting Eqs. (161) and (163) back into Eq. (159), we find the Markovian master equation in the Schrödinger picture

$$
\begin{aligned}
\partial_t \hat{\rho}_S(t) = -i[\hat{H}_S, \hat{\rho}_S(t)] + e^{-i\hat{H}_S t} [\partial_t \tilde{\rho}_S(t)] e^{i\hat{H}_S t} \\
= -i[\bar{\varepsilon}_d \hat{d}^{\dagger}\hat{d}, \hat{\rho}_S(t)] + \kappa[1-n_F(\varepsilon_d)]\mathcal{D}[\hat{d}]\hat{\rho}_S(t) + \kappa n_F(\varepsilon_d)\mathcal{D}[\hat{d}^{\dagger}]\hat{\rho}_S(t),
\end{aligned}
\tag{165}
$$

where the renormalized dot energy reads

$$
\bar{\varepsilon}_d = \varepsilon_d + P \int_{-\infty}^{\infty} d\omega \, \frac{\rho(\omega)}{\varepsilon_d-\omega}.
\tag{166}
$$

The reservoir thus has two effects: Through the dissipative part of the master equations, it describes electrons entering ($\mathcal{D}[\hat{d}^\dagger]$) and leaving ($\mathcal{D}[\hat{d}]$) the quantum dot. Note that to enter the dot, electrons in the reservoir have to be available, corresponding to the factor $n_\text{F}$ while to leave the dot, empty states have to be available corresponding to the factor $1-n_\text{F}$. In addition, the energy level of the quantum dot is renormalized.

Note that when taking the ratio between the rates of entering and leaving the quantum dot, we obtain the Boltzmann factor

$$e^{\beta(\varepsilon_\text{d}-\mu)} = \frac{\kappa[1-n_\text{F}(\varepsilon_\text{d})]}{\kappa n_\text{F}(\varepsilon_\text{d})}. \tag{167}$$

This condition is known as *local detailed balance* and it generally holds for transition rates that are induced by reservoirs in thermal equilibrium. It ensures that in equilibrium, the system state tends to a Gibbs state, see also Eq. (173) below.

As discussed above, the Markovian approximation is justified if the bath-correlation functions decay on a time scale which is much faster than the time over which $\tilde{\rho}_\text{S}$ varies. In the limiting case where both $\rho(\omega)$ as well as $n_\text{F}(\omega)$ are independent of $\omega$, the bath-correlation functions become proportional to a Dirac delta function and the environment becomes truly *memoryless* (i.e., $\tau_\text{B} \to 0$). In practice, it is sufficient for $\tau_\text{B}$ to be much shorter than any relevant time-scale of the system. In energy, this translates to the condition that the functions $\rho(\omega)$ as well as $n_\text{F}(\omega)$ are flat around the relevant energies of the system. For the present system $\tilde{\rho}_\text{S}$ changes on the time-scale $1/\kappa$. The Markov approximation is then valid as long as $\kappa\tau_\text{B} \ll 1$. In energy space, this requires $\rho(\omega)$ as well as $n_\text{F}(\omega)$ to be approximately constant in the interval $\omega \in [\varepsilon_\text{d}-\kappa, \varepsilon_\text{d}+\kappa]$. The spectral density depends on the details of the reservoir and may or may not fulfill this condition depending on the specific scenario. For the Fermi-Dirac occupation to be sufficiently flat for the Markov approximation, the following condition has to hold

$$\kappa \ll \max\{k_\text{B}T, |\varepsilon_\text{d}-\mu|\}. \tag{168}$$

At low temperatures, the Fermi-Dirac occupation becomes a step function. Therefore, the Markovian approximation is not justified at low temperatures if the dot level $\varepsilon_\text{d}$ is close to the chemical potential. For a more detailed discussion on the validity of the Born-Markov approximations, see appendix B.1 of Ref. [64].

### 4.3.2  Solving the master equation

To solve the master equation in Eq. (165), we write the density matrix of the system as

$$\hat{\rho}_\text{S}(t) = p_0(t)|0\rangle\langle 0| + p_1(t)|1\rangle\langle 1|, \qquad\qquad p_0(t) + p_1(t) = 1, \tag{169}$$

where $|0\rangle$ denotes the empty state and $|1\rangle$ the full state. Here we used the fact that we cannot have a superposition of states with a different number of electrons in the system due to a particle superselection rule [34]. Using these basis states, the fermionic operators can be cast into

$$\hat{d} = |0\rangle\langle 1|, \qquad \hat{d}^\dagger = |1\rangle\langle 0|, \qquad \hat{d}^\dagger\hat{d} = |1\rangle\langle 1|, \qquad p_1(t) = \text{Tr}\{\hat{d}^\dagger\hat{d}\hat{\rho}_\text{S}(t)\}. \tag{170}$$

The master equation in Eq. (165) can then be reduced to

$$\partial_t p_1(t) = -\kappa[1-n_\text{F}(\varepsilon_\text{d})]p_1(t) + \kappa n_\text{F}(\varepsilon_\text{d})p_0(t) = -\kappa[p_1(t)-n_\text{F}(\varepsilon_\text{d})]. \tag{171}$$

This equation shows that a full dot is emptied with rate $\kappa[1-n_\text{F}(\varepsilon_\text{d})]$ whereas an empty dot is filled with rate $\kappa n_\text{F}(\varepsilon_\text{d})$. The solution to this differential equation reads

$$p_1(t) = p_1(0)e^{-\kappa t} + n_\text{F}(\varepsilon_\text{d})(1-e^{-\kappa t}). \tag{172}$$

The occupation probability thus exponentially goes toward the equilibrium value $n_F(\varepsilon_d)$. The time-scale with which this happens is given by $1/\kappa$. In equilibrium, the system is described by the thermal state with temperature and chemical potential equal to those of the reservoir, as demanded by the zeroth law of thermodynamics, and we find

$$\lim_{t\to\infty}\hat{\rho}_S(t) = \frac{e^{-\beta(\varepsilon_d-\mu)\hat{d}^\dagger\hat{d}}}{\text{Tr}\{e^{-\beta(\varepsilon_d-\mu)\hat{d}^\dagger\hat{d}}\}}. \tag{173}$$

This result is closely related to the local detailed balance condition in Eq. (167).

### 4.3.3 Energy flows and the first law

In addition to the state of the quantum dot, we are interested in the energy flow between the dot and the reservoir. To this end, we consider the change in the energy of the system

$$\partial_t\langle\hat{H}_S\rangle = \text{Tr}\{\varepsilon_d\hat{d}^\dagger\hat{d}\,\partial_t\hat{\rho}_S\} = (\varepsilon_d-\mu)\partial_t\langle\hat{d}^\dagger\hat{d}\rangle + \mu\partial_t\langle\hat{d}^\dagger\hat{d}\rangle = -J_B(t) - P_B(t). \tag{174}$$

To identify the heat current and the power, we used that the change in the number of electrons in the system is minus the change in the number of electrons in the reservoir, more on this in Sec. 4.5. Using $p_1 = \langle\hat{d}^\dagger\hat{d}\rangle$, we find for the heat current and power that flow into the reservoir

$$J_B(t) = -(\varepsilon_d-\mu)\partial_t\langle\hat{d}^\dagger\hat{d}\rangle = (\varepsilon_d-\mu)\kappa e^{-\kappa t}[p_1(0) - n_F(\varepsilon_d)],$$
$$P_B(t) = -\mu\partial_t\langle\hat{d}^\dagger\hat{d}\rangle = \mu\kappa e^{-\kappa t}[p_1(0) - n_F(\varepsilon_d)]. \tag{175}$$

We thus find that if the dot starts out in a non-equilibrium state, there is an exponentially decreasing energy flow which can be divided into power and heat. The power flows into the reservoir whenever $p_1(t) > n_F(\varepsilon_d)$. The heat flow additionally depends on the sign of $\varepsilon_d-\mu$: electrons entering the reservoir above the chemical potential ($\varepsilon_d-\mu > 0$) heat up the reservoir, while electrons entering below the chemical potential ($\varepsilon_d-\mu < 0$) cool it down. This can be understood intuitively by noting that at zero temperature, all states below the chemical potential are occupied while states above the chemical potential are empty. Electrons entering the reservoir below the chemical potential thus bring the reservoir closer to the zero-temperature distribution.

### 4.3.4 Entropy and the second law

For the second law, we need to consider the entropy of the quantum dot given by

$$S_{vN}[\hat{\rho}_S(t)] = -p_0(t)\ln p_0(t) - p_1(t)\ln p_1(t), \quad \partial_t S_{vN}[\hat{\rho}_S(t)] = -\dot{p}_1(t)\ln\frac{p_1(t)}{1-p_1(t)}, \tag{176}$$

where the dot denotes the time-derivative and we used $p_0 + p_1 = 1$ to compute the derivative. The entropy production rate given in Eq. (115) can then be expressed as

$$\dot{\Sigma}(t) = k_B\partial_t S_{vN}[\hat{\rho}_S(t)] + \frac{J_B}{T} = \frac{k_B J_B(t)}{(\varepsilon_d-\mu)}\left[\beta(\varepsilon_d-\mu) + \ln\frac{p_1(t)}{1-p_1(t)}\right], \tag{177}$$

where we used $-\dot{p}_1 = J_B/(\varepsilon_d-\mu)$ which follows from Eq. (175). Using the equality in Eq. (167), we find

$$\dot{\Sigma}(t) = \frac{k_B J_B(t)}{(\varepsilon_d-\mu)}\ln\left(\frac{p_1(t)[1-n_F(\varepsilon_d)]}{[1-p_1(t)]n_F(\varepsilon_d)}\right)$$
$$= k_B\kappa\left(p_1(t)[1-n_F(\varepsilon_d)] - [1-p_1(t)]n_F(\varepsilon_d)\right)\ln\left(\frac{p_1[1-n_F(\varepsilon_d)]}{[1-p_1]n_F(\varepsilon_d)}\right) \geq 0, \tag{178}$$

where we used $J_B = (\varepsilon_d - \mu)\kappa(p_1 - n_F)$ which follows from Eqs. (171) and (175). the positivity of $\dot{\Sigma}$ can be shown by writing it in the form $(x - y)(\ln x - \ln y) \geq 0$ which is ensured to be positive because the logarithm is a monotonously increasing function. Using the solution in Eq. (172), we can explicitly write the entropy production rate as

$$\dot{\Sigma}(t) = k_B \kappa e^{-\kappa t} \delta_0 \ln \frac{(e^{-\kappa t}\delta_0 + n_F)(1 - n_F)}{(1 - n_F - e^{-\kappa t}\delta_0)n_F}, \tag{179}$$

where we introduced $\delta_0 = p_1(0) - n_F$ and we suppressed the argument of the Fermi-Dirac occupation for ease of notation. We thus find that for this Markovian master equation, the entropy production rate is indeed always positive, as anticipated above, and exponentially decreases in time such that in equilibrium, no entropy is produced as expected.

## 4.4 Obtaining GKLS form

We now return to Eq. (144). As mentioned above, this equation is not in GKLS form. In this section, we consider additional approximations that bring it into GKLS form. To this end, we first write

$$\tilde{S}_{\alpha,k}(t) = \sum_j e^{-i\omega_j t} \hat{S}_{\alpha,k}^j, \tag{180}$$

i.e., we write the system operators in Eq. (144) in a Fourier series. While this can always be done, it is particularly simple when the system Hamiltonian is time-independent. In this case, we may introduce its eigenstates as $\hat{H}_S |E_a\rangle = E_a |E_a\rangle$. We may then find the operators $\hat{S}_{\alpha,k}^j$ by multiplying $\hat{S}_{\alpha,k}$ from the left and the right by resolved identities

$$\hat{S}_{\alpha,k} = \sum_{a,b} \underbrace{|E_a\rangle\langle E_a| \hat{S}_{\alpha,k} |E_b\rangle\langle E_b|}_{\hat{S}_{\alpha,k}^j} \qquad \Rightarrow \qquad [\hat{S}_{\alpha,k}^j, \hat{H}_S] = \omega_j \hat{S}_{\alpha,k}^j, \tag{181}$$

with $\omega_j = E_b - E_a$.

With the help of Eq. (180), we may cast Eq. (144) into

$$\partial_t \tilde{\rho}_S(t) = \sum_{\alpha,k,k'} \sum_{j,j'} e^{i(\omega_j - \omega_{j'})t} \Gamma_{k,k'}^\alpha(\omega_{j'}) \left[ \hat{S}_{\alpha,k'}^{j'} \tilde{\rho}_S(t) \left(\hat{S}_{\alpha,k}^j\right)^\dagger - \left(\hat{S}_{\alpha,k}^j\right)^\dagger \hat{S}_{\alpha,k'}^{j'} \tilde{\rho}_S(t) \right] + H.c. \tag{182}$$

with

$$\Gamma_{k,k'}^\alpha(\omega) \equiv \int_0^\infty ds\, e^{i\omega s} C_{k,k'}^\alpha(s) = \frac{1}{2}\gamma_{k,k'}^\alpha(\omega) + i\Delta_{k,k'}^\alpha(\omega), \tag{183}$$

where $\gamma_{k,k'}^\alpha(\omega)$ and $\Delta_{k,k'}^\alpha(\omega)$ are both real.

In the remainder of this section, we will consider a time-independent Hamiltonian $\hat{H}_S$, as well as bath-correlation functions that obey $C_{k,k'}^\alpha \propto \delta_{k,k'}$ for simplicity. This will simplify the notation as well as the derivation of the laws of thermodynamics for the master equations we consider.

### 4.4.1 The secular approximation

The secular approximation is the most common approach for obtaining a master equation in GKLS form and can be found in many text-books (see for instance Ref. [52]). Let $\tau_S$ denote the time scale over which $\tilde{\rho}_S$ changes (remember that the Born-Markov approximations require $\tau_S \gg \tau_B$). The secular approximation is justified if

$$|\omega_j - \omega_{j'}|\tau_S \gg 1 \qquad\qquad \forall\, j \neq j'. \tag{184}$$

In this case, we may drop all terms in Eq. (182) with $j \neq j'$ because they average out over a time-scale shorter than $\tau_S$ due to the oscillating term $\exp[i(\omega_j - \omega_{j'})t]$.

Going back to the Schrödinger picture, we then find the GKLS master equation

$$\partial_t \hat{\rho}_S(t) = -i[\hat{H}_S + \hat{H}_{LS}, \hat{\rho}_S(t)] + \sum_{\alpha,k,j} \gamma_k^\alpha(\omega_j) \mathcal{D}[\hat{s}_{\alpha,k}^j]\hat{\rho}_S(t), \tag{185}$$

where $\gamma_k^\alpha(\omega_j) = \gamma_{k,k}^\alpha(\omega_j)$ and the Lamb-shift Hamiltonian is given by

$$\hat{H}_{LS} = \sum_{\alpha,k,j} \Delta_k^\alpha(\omega_j) \left(\hat{s}_{\alpha,k}^j\right)^\dagger \hat{s}_{\alpha,k}^j, \tag{186}$$

with $\Delta_k^\alpha(\omega_j) = \Delta_{k,k}^\alpha(\omega_j)$. We note that from Eq. (181), it follows that $[\hat{H}_S, \hat{H}_{LS}] = 0$.

Due to Eq. (184), the secular approximation works well for systems which have no small gaps in $\hat{H}_S$. As we will see, thermal machines may consist of multiple parts that are weakly coupled and have small energy gaps induced by the weak coupling. In such systems, the secular approximation breaks down. Note that in order to obtain the jump operators $\hat{s}_{\alpha,k}^j$, the Hamiltonian $\hat{H}_S$ needs to be diagonalized first. This implies that already obtaining the master equation may be a formidable task.

We note that in the secular approximation, for a non-degenerate system Hamiltonian, the populations decouple from the coherences. Indeed, the dissipative part of Eq. (185) describes classical jumps between the eigenstates of $\hat{H}_S$. Concretely, this means that in the energy-eigenbasis, the off-diagonal terms of the density matrix tend to zero and the dynamics can be described by a classical rate equation involving only the populations. While this may be a good approximation, there are many situations of interest where coherences between energy eigenstates are important in the presence of thermal reservoirs and the secular approximation is no longer justified [65–70].

A particularly appealing feature of the secular approximation is the fact that the laws of thermodynamics are ensured to hold [10]:

**0th law:** If all reservoirs are at the same inverse temperature $\beta$ and chemical potential $\mu$, then the steady state of Eq. (185) reduces to the Gibbs state

$$\hat{\rho}_S(t) \xrightarrow{t \to \infty} \frac{e^{-\beta(\hat{H}_S - \mu \hat{N}_S)}}{\text{Tr}\left\{e^{-\beta(\hat{H}_S - \mu \hat{N}_S)}\right\}}. \tag{187}$$

In equilibrium, the system is thus described by the same temperature and chemical potential as the environment.

**1st law:** Writing the master equation in Eq. (185) as

$$\partial_t \hat{\rho}_S(t) = -i[\hat{H}_S + \hat{H}_{LS}, \hat{\rho}_S(t)] + \sum_\alpha \mathcal{L}_\alpha \hat{\rho}_S(t), \tag{188}$$

we find the first law as

$$\partial_t \langle \hat{H}_S \rangle = -\sum_\alpha [J_\alpha(t) + P_\alpha(t)], \tag{189}$$

with

$$J_\alpha(t) = -\text{Tr}\left\{(\hat{H}_S - \mu_\alpha \hat{N}_S)\mathcal{L}_\alpha \hat{\rho}_S(t)\right\}, \qquad P_\alpha(t) = -\mu_\alpha \text{Tr}\left\{\hat{N}_S \mathcal{L}_\alpha \hat{\rho}_S(t)\right\}. \tag{190}$$

These definitions are to be compared with the definitions for the power and heat current in the general scenario, c.f. Eq. (111). There, we defined heat and work using the changes in the

energy and particle numbers in the reservoirs. When using a master equation for the reduced system state, we no longer have access to the properties of the reservoirs. However, we may still infer heat and work because the term $\mathcal{L}_\alpha$ in the master equation describes the exchange of energy and particles with reservoir $\alpha$. An increase in the particle number due to $\mathcal{L}_\alpha$ implies a decrease of the particle number in the reservoir $\alpha$ by the same amount, and similarly for energy. This is discussed in more detail below, see Sec. 4.5.

**2nd law:** The second law of thermodynamics also follows from Eq. (185) and it may be shown that [71, 72]

$$\dot{\Sigma} = k_B \partial_t S_{vN}[\hat{\rho}_S(t)] + \sum_\alpha \frac{J_\alpha(t)}{T_\alpha} \geq 0. \tag{191}$$

While we focused on a time-independent system Hamiltonian here, the secular approximation may analogously be applied for a time-dependent Hamiltonian and the laws of thermodynamics continue to hold in this case, see Ref. [10] and references therein.

### 4.4.2   The singular-coupling limit

The singular-coupling limit [52, 54] is another popular approach to obtain a master equation in GKLS form. It is justified when all Bohr frequencies $\omega_j$ are close to each other, i.e.,

$$|\omega_j - \omega_{j'}|\tau_B \ll 1 \qquad \qquad \forall \, j, j'. \tag{192}$$

In this case, we may write

$$\tilde{S}_{\alpha,k}(t-s) = \sum_j e^{-i\omega_j(t-s)} \hat{S}_{\alpha,k}^j \simeq e^{i\omega_{\alpha,k}s} \sum_j e^{-i\omega_j t} \hat{S}_{\alpha,k}^j = e^{i\omega_{\alpha,k}s} \tilde{S}_{\alpha,k}(t), \tag{193}$$

where $\omega_{\alpha,k}$ has to be chosen such that $|\omega_{\alpha,k} - \omega_j|\tau_B \ll 1$ for all $j$. Because of Eq. (192), one may for instance choose $\omega_{\alpha,k}$ to equal the average of all $\omega_j$. Equation (193) is justified for all $s \lesssim \tau_B$, i.e., exactly the values of $s$ which are relevant in the integral of Eq. (144). As a consequence of Eq. (193), we may replace $\Gamma_k^\alpha(\omega_{j'})$ by $\Gamma_k^\alpha(\omega_{\alpha,k})$ in Eq. (182) which, in the Schrödinger picture, results in the GKLS master equation

$$\partial_t \hat{\rho}_S(t) = -i[\hat{H}_S + \hat{H}_{LS}, \hat{\rho}_S(t)] + \sum_{\alpha,k} \gamma_k^\alpha(\omega_{\alpha,k}) \mathcal{D}[\hat{S}_{\alpha,k}] \hat{\rho}_S(t), \tag{194}$$

where the Lamb-shift Hamiltonian reads

$$\hat{H}_{LS} = \sum_{\alpha,k} \Delta_k^\alpha(\omega_{\alpha,k}) \hat{S}_{\alpha,k}^\dagger \hat{S}_{\alpha,k}, \tag{195}$$

which does not necessarily commute with $\hat{H}_S$.

Note that the jump operators $\hat{S}_{\alpha,k}$ entering Eq. (194) are the operators that enter the coupling Hamiltonian $\hat{V}_\alpha$. This implies that, in contrast to the secular approximation, the system Hamiltonian does not need to be diagonalized in order to write down the master equation. Since we are often interested in systems that are not explicitly solvable, this is very helpful.

We further note that the singular-coupling limit is always justified for a perfectly Markovian environment, i.e., an environment where $\tau_B \to 0$. More generally, the singular-coupling limit is justified when $\gamma_k^\alpha(\omega_j) \simeq \gamma_k^\alpha(\omega_{\alpha,k})$ for all $j$. This is often the case in quantum-optical systems, which is why the singular-coupling limit is widely applied in this community.

Showing that the laws of thermodynamics hold in the singular-coupling limit is a bit more difficult than in the secular approximation. We first need to introduce a *thermodynamic Hamiltonian* $\hat{H}_{TD}$, which is obtained by rescaling the gaps of $\hat{H}_S$ as $\omega_j \to \omega_{\alpha,k}$. The thermodynamic

Hamiltonian is then used to compute the internal energy of the system. As argued in Ref. [64], the mistake we make by replacing $\hat{H}_\text{S}$ with $\hat{H}_\text{TD}$ in the thermodynamic bookkeeping is smaller than the resolution of heat that the master equation in Eq. (194) ensures. Within the accuracy of our model, the replacement is thus completely justified and in many cases it can be compared to neglecting the system-bath coupling in the thermodynamic bookkeeping. For the thermodynamic Hamiltonian, we find

$$[\hat{H}_\text{TD}, \hat{H}_\text{S}] = [\hat{H}_\text{TD}, \hat{H}_\text{LS}] = 0, \qquad\qquad [\hat{S}_{\alpha,k}, \hat{H}_\text{TD}] = \omega_{\alpha,k}\hat{S}_{\alpha,k}. \tag{196}$$

The last equality implies that a jump $\hat{S}_{\alpha,k}\hat{\rho}_\text{S}\hat{S}^{\dagger}_{\alpha,k}$ reduces the internal energy by $\omega_{\alpha,k}$.

With the help of the thermodynamic Hamiltonian, the laws of thermodynamics may be shown to hold [64]:

**0th law:**  If all reservoirs are at the same inverse temperature $\beta$ and chemical potential $\mu$, then the steady state of Eq. (194) reduces to the Gibbs state

$$\hat{\rho}_\text{S}(t) \xrightarrow{t\to\infty} \frac{e^{-\beta(\hat{H}_\text{TD}-\mu\hat{N}_\text{S})}}{\text{Tr}\left\{e^{-\beta(\hat{H}_\text{TD}-\mu\hat{N}_\text{S})}\right\}}. \tag{197}$$

In equilibrium, the system is thus described by the same temperature and chemical potential as the environment.

**1st law:**  Writing the master equation in Eq. (194) as

$$\partial_t\hat{\rho}_\text{S}(t) = -i[\hat{H}_\text{S} + \hat{H}_\text{LS}, \hat{\rho}_\text{S}(t)] + \sum_{\alpha}\mathcal{L}_{\alpha}\hat{\rho}_\text{S}(t), \tag{198}$$

we find the first law as

$$\partial_t\langle\hat{H}_\text{TD}\rangle = -\sum_{\alpha}[J_{\alpha}(t) + P_{\alpha}(t)], \tag{199}$$

with

$$J_{\alpha}(t) = -\text{Tr}\left\{\left(\hat{H}_\text{TD} - \mu_{\alpha}\hat{N}_\text{S}\right)\mathcal{L}_{\alpha}\hat{\rho}_\text{S}(t)\right\}, \qquad P_{\alpha}(t) = -\mu_{\alpha}\text{Tr}\left\{\hat{N}_\text{S}\mathcal{L}_{\alpha}\hat{\rho}_\text{S}(t)\right\}. \tag{200}$$

**2nd law:**  The second law of thermodynamics also follows from Eq. (194) and it may be shown that

$$\dot{\Sigma} = k_\text{B}\partial_t S_\text{vN}[\hat{\rho}_\text{S}(t)] + \sum_{\alpha}\frac{J_{\alpha}(t)}{T_{\alpha}} \geq 0. \tag{201}$$

As for the secular approximation, the results from this section may be extended to a time-dependent Hamiltonian and the laws of thermodynamics continue to hold in this case [64].

We note that in particular in thermodynamic contexts, it is often the case that there are positive and negative Bohr frequencies, with each $\tilde{S}_{\alpha,k}(t)$ containing only frequencies of one sign. In this case, we may perform the singular-coupling limit for the positive and negative Bohr frequencies separately. In this case, the master equation in Eq. (194) is also valid if Eq. (192) is only respected for frequencies $\omega_j$ and $\omega_{j'}$ of the same sign. Below, we will discuss an example where this is the case, see Sec. 4.6.

### 4.4.3   The unified GKLS master equation

Finally, we briefly mention an approach to obtain GKLS form for systems where neither the secular approximation nor the singular-coupling limit is justified [64,73]. The problem is that for some values of $j$ and $j'$, we may find $|\omega_j - \omega_{j'}|\tau_S \lesssim 1$, rendering the secular approximation inapplicable, while for other values of $j$ and $j'$, we may have $|\omega_j - \omega_{j'}|\tau_B \gtrsim 1$, such that the singular-coupling limit may not be applied. The solution for this problem exploits the fact that $\tau_B \ll \tau_S$, otherwise the Born-Markov approximations are not justified in the first place. This inequality implies that for all values of $j$ and $j'$, we either have $|\omega_j - \omega_{j'}|\tau_S \gg 1$ or we have $|\omega_j - \omega_{j'}|\tau_B \ll 1$. One may then perform the following approximations to reach GKLS form:

1. Drop all terms with $j$ and $j'$ such that $|\omega_j - \omega_{j'}|\tau_S \gg 1$, in analogy to the secular approximation.

2. Perform a singular-coupling limit on the remaining cross terms with $\omega_j \neq \omega_{j'}$.

The resulting master equation also obeys the laws of thermodynamics for an appropriately chosen thermodynamic Hamiltonian. The thermodynamic consistency of this approach was recently also shown using the method of full counting statistics [74].

## 4.5   Heat and work in quantum master equations

Here we briefly connect the definitions of heat and work we use for master equations, as introduced in Eqs. (190) and (200), to the definitions introduced for the general scenario, see Eqs. (111). We first consider power. The conservation of the total number of particles results in

$$\partial_t \langle \hat{N}_S \rangle = -\partial_t \sum_\alpha \langle \hat{N}_\alpha \rangle, \tag{202}$$

where the averages are taken with respect to the exact, total density matrix $\hat{\rho}_{tot}(t)$. In a master equation written in the form of Eq. (198), we only have access to the left-hand side of the last equation, which is given by

$$\partial_t \langle \hat{N}_S \rangle = \sum_\alpha \text{Tr}\{\hat{N}_S \mathcal{L}_\alpha \hat{\rho}_S(t)\}. \tag{203}$$

Comparing Eq. (203) to Eq. (202), we may infer

$$P_\alpha(t) = \mu_\alpha \partial_t \langle \hat{N}_\alpha \rangle = -\mu_\alpha \text{Tr}\{\hat{N}_S \mathcal{L}_\alpha \hat{\rho}_S(t)\}, \tag{204}$$

connecting the definition in Eqs. (111) to Eqs. (190) and (200). Since the contributions of the different reservoirs to the Liouvillean are additive, we may infer the change of particle number induced by each reservoir and, by exploiting the conservation of particles, we may infer $\partial_t \langle \hat{N}_\alpha \rangle$ even though we only have access to the reduced system state.

   A similar analysis can be done for energy. From total energy conservation (assuming as above a time-independent Hamiltonian), we have

$$\partial_t \langle \hat{H}_S \rangle = -\partial_t \sum_\alpha \left[ \langle \hat{H}_\alpha \rangle + \langle \hat{V}_\alpha \rangle \right] \simeq -\partial_t \sum_\alpha \langle \hat{H}_\alpha \rangle, \tag{205}$$

where we neglected the energy stored in the coupling between system and environment because we assume it to be small in order to derive a Markovian master equation. From a master equation written in the form of Eq. (198), we may write the left-hand side of Eq. (205) as

$$\partial_t \langle \hat{H}_S \rangle = \sum_\alpha \text{Tr}\{\hat{H}_S \mathcal{L}_\alpha \hat{\rho}_S(t)\}, \tag{206}$$

which leads us to identify

$$\partial_t \langle \hat{H}_\alpha \rangle \simeq -\mathrm{Tr}\{\hat{H}_S \mathcal{L}_\alpha \hat{\rho}_S(t)\} \tag{207}$$

and therefore to define the heat current as

$$J'_\alpha = -\mathrm{Tr}\{(\hat{H}_S - \mu_\alpha N_S)\mathcal{L}_\alpha \hat{\rho}_S(t)\} \simeq \partial_t \langle \hat{H}_\alpha \rangle - \mu_\alpha \partial_t \langle \hat{N}_\alpha \rangle. \tag{208}$$

While this is indeed the appropriate heat current for master equations in the secular approximation, see Eq. (190), we introduced a different definition in Eq. (200) based on a thermodynamic Hamiltonian. To understand why, it is necessary to appreciate that Markovian master equations rely on assumptions on time-scales. In particular, the Markov approximation neglects the time-evolution of the system during the bath-correlation time, see Eq. (142). This neglects the finite life-time of the particles in the system. Through the time-energy uncertainty relation, this neglects the energy broadening of the states in the system. As a consequence, we no longer know the exact energy at which particles are exchanged with the environment. In the singular-coupling limit, this is exacerbated because we also neglect small differences in the Bohr frequencies of the system Hamiltonian, see Eq. (193).

The fact that we no longer know the exact energy at which particles enter the environment implies that we lose resolution in the energy exchanged with the environment. As a result, a naive application of Eq. (208) can result in violations of the laws of thermodynamics, see for instance Ref. [75]. It is important to stress however that any violations should remain absent or negligibly small as long as the approximations that result in the master equation are well justified [76, 77].

It is however desirable to have a framework that mathematically ensures the laws of thermodynamics. This can be obtained by introducing the thermodynamic Hamiltonian to quantify the internal energy of the system [64] which results in the definition for the heat currents [c.f. Eq. (200)]

$$J_\alpha \equiv -\mathrm{Tr}\{(\hat{H}_{TD} - \mu_\alpha N_S)\mathcal{L}_\alpha \hat{\rho}_S(t)\} \simeq \partial_t \langle \hat{H}_\alpha \rangle - \mu_\alpha \partial_t \langle \hat{N}_\alpha \rangle. \tag{209}$$

Importantly, the use of the thermodynamic Hamiltonian should only slightly affect the values of the heat currents, i.e., $\partial_t \langle \hat{H}_S \rangle \simeq \partial_t \langle \hat{H}_{TD} \rangle$ and $J'_\alpha \simeq J_\alpha$, such that both definitions of the heat current are acceptable. Should the two definitions result in substantial differences, then the approximations that went into the master equation are most likely no longer justified.

Since we will mainly consider master equations in the singular-coupling limit below, and since we prefer to have a framework that ensures the laws of thermodynamics exactly and not only approximately, we will use the heat current given in Eq. (209) and the power given in Eq. (204).

## 4.6 Example: A double quantum dot

We now consider an example provided by a double quantum dot, coupled to two fermionic reservoirs. We again consider spinless electrons, such that each dot can at maximum host one electron. Furthermore, we consider two dots in series, such that each dot is coupled to one of the reservoirs, see Fig. 10. The total Hamiltonian that describes this scenario is given by

$$\hat{H}_{tot} = \hat{H}_S + \hat{H}_L + \hat{H}_R + \hat{V}_L + \hat{V}_R, \tag{210}$$

with the system Hamiltonian

$$\hat{H}_S = \varepsilon \hat{d}^\dagger_L \hat{d}_L + \varepsilon \hat{d}^\dagger_R \hat{d}_R + g(\hat{d}^\dagger_L \hat{d}_R + \hat{d}^\dagger_R \hat{d}_L) = (\varepsilon + g)\hat{d}^\dagger_+ \hat{d}_+ + (\varepsilon - g)\hat{d}^\dagger_- \hat{d}_-, \tag{211}$$

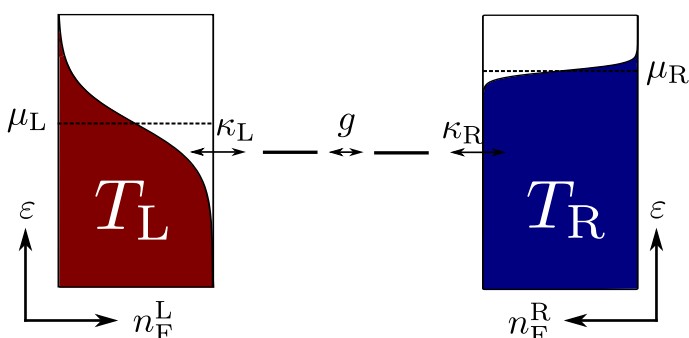

Figure 10: Double quantum dot coupled to two fermionic reservoirs. The reservoirs are characterized by temperatures $T_\alpha$ and chemical potentials $\mu_\alpha$. The couplings between the reservoirs and the double dot are given by $\kappa_\alpha$, and $g$ denotes the inter-dot tunnel coupling. A temperature bias ($T_L \neq T_R$) and/or a chemical potential bias ($\mu_L \neq \mu_R$) may result in particle and heat transport through the double dot in the steady state.

where we chose the on-site energies $\varepsilon$ of the two dots to be equal for simplicity and we introduced the eigenmodes

$$\hat{d}_\pm = \frac{1}{\sqrt{2}}\left(\hat{d}_R \pm \hat{d}_L\right), \qquad \hat{d}_R = \frac{1}{\sqrt{2}}\left(\hat{d}_+ + \hat{d}_-\right), \qquad \hat{d}_L = \frac{1}{\sqrt{2}}\left(\hat{d}_+ - \hat{d}_-\right). \qquad (212)$$

The reservoirs are again modeled by a collection of non-interacting electrons that are tunnel-coupled to the respective dots

$$\hat{H}_\alpha = \sum_q \varepsilon_{\alpha,q} \hat{c}^\dagger_{\alpha,q} \hat{c}_{\alpha,q}, \qquad \hat{V}_\alpha = \sum_q \left(g_{\alpha,q} \hat{d}_\alpha \hat{c}^\dagger_{\alpha,q} - g^*_{\alpha,q} \hat{d}^\dagger_\alpha \hat{c}_{\alpha,q}\right), \qquad (213)$$

with $\alpha = L, R$. As in the example of a single quantum dot, we may write the coupling Hamiltonian as $\hat{V}_\alpha = \hat{S}_{\alpha,0}\hat{B}_{\alpha,0} + \hat{S}_{\alpha,1}\hat{B}_{\alpha,1}$ and we find

$$\tilde{S}_{R,0}(t) = e^{i\hat{H}_S t}\hat{d}_R e^{-i\hat{H}_S t} = e^{-i(\varepsilon+g)t}\frac{\hat{d}_+}{\sqrt{2}} + e^{-i(\varepsilon-g)t}\frac{\hat{d}_-}{\sqrt{2}},$$
$$\tilde{S}_{L,0}(t) = e^{i\hat{H}_S t}\hat{d}_L e^{-i\hat{H}_S t} = e^{-i(\varepsilon+g)t}\frac{\hat{d}_+}{\sqrt{2}} - e^{-i(\varepsilon-g)t}\frac{\hat{d}_-}{\sqrt{2}}. \qquad (214)$$

Comparing these expressions to Eq. (180), we may read off the frequencies $\omega_j = \varepsilon \pm g$, as well as the corresponding operators $\hat{S}^j_{\alpha,0}$. Similarly, we find $\tilde{S}_{\alpha,1}(t) = \tilde{S}^\dagger_{\alpha,0}(t)$, which involve the frequencies $\omega_j = -(\varepsilon \pm g)$.

The bath-correlation functions are obtained in analogy to the single-dot case in Sec. 4.3 and read

$$C^\alpha_{0,0}(s) = \int_{-\infty}^\infty d\omega e^{-i\omega s}\rho_\alpha(\omega)[1 - n^\alpha_F(\omega)], \qquad C^\alpha_{1,1}(s) = \int_{-\infty}^\infty d\omega e^{i\omega s}\rho_\alpha(\omega)n^\alpha_F(\omega), \qquad (215)$$

with the Fermi-Dirac occupation

$$n^\alpha_F(\varepsilon) = \frac{1}{e^{\frac{\varepsilon-\mu_\alpha}{k_B T_\alpha}} + 1}. \qquad (216)$$

Furthermore, from Eq. (183) we find the relevant transition rates

$$\gamma^\alpha_0(\omega) = \kappa_\alpha[1 - n^\alpha_F(\omega)], \qquad \gamma^\alpha_1(-\omega) = \kappa_\alpha n^\alpha_F(\omega), \qquad (217)$$

where $\kappa_\alpha = 2\pi\rho_\alpha(\omega)$, as well as the energy shifts

$$\Delta_0^\alpha(\omega) = P \int_{-\infty}^{\infty} d\omega' \frac{\rho_\alpha(\omega')[1-n_F^\alpha(\omega')]}{\omega - \omega'}, \qquad \Delta_1^\alpha(-\omega) = -P \int_{-\infty}^{\infty} d\omega' \frac{\rho_\alpha(\omega')n_F^\alpha(\omega')}{\omega - \omega'}. \tag{218}$$

Note the different signs in the arguments of the functions with different subscripts. As we will see below, these functions will be evaluated at frequencies of opposite signs. From the bath-correlation functions, we conclude that the Born-Markov approximations are justified when

$$\rho_\alpha(\varepsilon \pm g \pm \kappa) \simeq \rho_\alpha(\varepsilon \pm g), \qquad n_F^\alpha(\varepsilon \pm g \pm \kappa) \simeq n_F^\alpha(\varepsilon \pm g), \tag{219}$$

where $\kappa = \max\{\kappa_L, \kappa_R\}$.

### 4.6.1 The secular approximation

Having identified all the quantities appearing in Eq. (185), we find the GKLS master equation in the secular approximation

$$\begin{aligned} \partial_t \hat{\rho}_S(t) = &-i \left[ \sum_{\sigma=\pm} \bar{\varepsilon}_\sigma \hat{d}_\sigma^\dagger \hat{d}_\sigma, \hat{\rho}_S(t) \right] \\ &+ \sum_{\alpha=L,R} \sum_{\sigma=\pm} \frac{\kappa_\alpha}{2} \left\{ n_F^\alpha(\varepsilon_\sigma)\mathcal{D}[\hat{d}_\sigma^\dagger] + [1 - n_F^\alpha(\varepsilon_\sigma)]\mathcal{D}[\hat{d}_\sigma] \right\} \hat{\rho}_S(t), \end{aligned} \tag{220}$$

with the energies

$$\varepsilon_\pm = \varepsilon \pm g, \qquad \bar{\varepsilon}_\pm = \varepsilon_\pm + P \int_{-\infty}^{\infty} d\omega \frac{\rho_L(\omega) + \rho_R(\omega)}{\varepsilon \pm g - \omega}. \tag{221}$$

The master equation in the secular approximation describes classical hopping into and out-of the delocalized eigenmodes described by the operators $\hat{d}_\pm$. Any coherences between these eigenmodes decay. We may thus interpret the secular master equation as a classical master equation. However, note that the eigenmodes themselves describe electrons being in a coherent superposition between the two dots.

Evaluating Eq. (190), we find the heat flow and power

$$J_\alpha = \sum_{\sigma=\pm} \frac{\varepsilon_\sigma - \mu_\alpha}{2} \kappa_\alpha \left[ \langle \hat{d}_\sigma^\dagger \hat{d}_\sigma \rangle - n_F^\alpha(\varepsilon_\sigma) \right], \qquad P_\alpha = \mu_\alpha \sum_{\sigma=\pm} \frac{\kappa_\alpha}{2} \left[ \langle \hat{d}_\sigma^\dagger \hat{d}_\sigma \rangle - n_F^\alpha(\varepsilon_\sigma) \right]. \tag{222}$$

From Eq. (184), we conclude that the secular approximation is justified when

$$\frac{1}{\tau_S} = \kappa_\alpha \ll g = \frac{1}{2}|\omega_j - \omega_{j'}|. \tag{223}$$

This implies that for strong coupling between the dots, the secular approximation can safely be applied. However, once $g$ becomes comparable to either coupling $\kappa_\alpha$, it is no longer justified. In this case, one should apply the singular-coupling limit.

### 4.6.2 The singular-coupling limit

In the singular-coupling limit, we make the replacement

$$\tilde{S}_{\alpha,0}(t-s) \simeq e^{i\varepsilon s}\tilde{S}_{\alpha,0}(t), \qquad \tilde{S}_{\alpha,1}(t-s) \simeq e^{-i\varepsilon s}\tilde{S}_{\alpha,1}(t). \tag{224}$$

This implies that the frequencies appearing in Eq. (194) are $\omega_{\alpha,0} = \varepsilon$ and $\omega_{\alpha,1} = -\varepsilon$. We note that there is no restriction on $|\omega_{\alpha,0} - \omega_{\alpha,1}|$ because $\tilde{S}_{\alpha,0}$ only involves the frequencies $\varepsilon \pm g$

while $\tilde{S}_{\alpha,1}$ only involves the frequencies $-(\varepsilon \pm g)$. With the substitution in Eq. (224), we find the GKLS master equation

$$\partial_t \hat{\rho}_S(t) = -i[\hat{H}_S + \hat{H}_{LS}, \hat{\rho}_S(t)] + \sum_{\alpha=L,R} \kappa_\alpha \left\{ n_F^\alpha(\varepsilon) \mathcal{D}[\hat{d}_\alpha^\dagger] + [1 - n_F^\alpha(\varepsilon)] \mathcal{D}[\hat{d}_\alpha] \right\} \hat{\rho}_S(t), \quad (225)$$

with the Lamb-shift Hamiltonian

$$\hat{H}_{LS} = \Delta_L \hat{d}_L^\dagger \hat{d}_L + \Delta_R \hat{d}_R^\dagger \hat{d}_R, \qquad \Delta_\alpha = P \int_{-\infty}^\infty d\omega \frac{\rho_\alpha(\omega)}{\varepsilon - \omega}. \quad (226)$$

The master equation in the singular-coupling limit is also known as a *local* master equation, because the jump operators act locally on the left and right quantum dots. In contrast to the secular master equation (also denoted *global* master equation), the populations and coherences do not decouple and we may find coherence and even entanglement between the two quantum dots [70]. We note that the local master equation may also be obtained heuristically by first setting $g = 0$, deriving the master equation as in Sec. 4.3, and then reinstating $g$ in the Hamiltonian. For certain scenarios, this heuristic approach may result in master equations that violate the laws of thermodynamics [75]. For this reason, it is recommended to perform the singular-coupling approximation as outlined above. We then obtain a thermodynamically consistent framework with the thermodynamic Hamiltonian

$$\hat{H}_{TD} = \varepsilon \left( \hat{d}_L^\dagger \hat{d}_L + \hat{d}_R^\dagger \hat{d}_R \right). \quad (227)$$

In the thermodynamic bookkeeping, we thus neglect the coupling between the dots, in analogy to how we neglect the system-bath coupling energy.

Evaluating Eq. (200), we find the heat flow and power

$$J_\alpha = \kappa_\alpha (\varepsilon - \mu_\alpha) \left[ \langle \hat{d}_\alpha^\dagger \hat{d}_\alpha \rangle - n_F^\alpha(\varepsilon) \right], \qquad P_\alpha = \kappa_\alpha \mu_\alpha \left[ \langle \hat{d}_\alpha^\dagger \hat{d}_\alpha \rangle - n_F^\alpha(\varepsilon) \right]. \quad (228)$$

In this approximation, each electron that enters reservoir $\alpha$ carries the heat $\varepsilon - \mu_\alpha$ and the power $\mu_\alpha$, resulting in the simple relation $J_\alpha/P_\alpha = (\varepsilon - \mu_\alpha)/\mu_\alpha$.

The singular-coupling limit is justified when $\rho_\alpha(\varepsilon \pm g) \simeq \rho_\alpha(\varepsilon)$ as well as $n_F^\alpha(\varepsilon \pm g) \simeq n_F^\alpha(\varepsilon)$. The second condition is obeyed when

$$g \ll \max\{k_B T_\alpha, |\varepsilon - \mu_\alpha|\}. \quad (229)$$

Note that $g \ll \kappa$ is *not* required for the singular-coupling limit to be justified. The secular approximation and the singular-coupling limit may thus be justified at the same time. Indeed, since $\tau_S \gg \tau_B$ this is what we expect from Eqs. (184) and (192).

# 5 Quantum thermal machines

In this section, we consider quantum thermal machines, i.e., machines that use reservoirs in local equilibrium to perform a useful task such as converting heat into work or producing entanglement. While in local equilibrium, these reservoirs have different temperatures and/or chemical potentials such that together, they describe an out-of-equilibrium scenario.

## 5.1 A quantum dot heat engine

The first machine we consider is a simplified version of the heat engine that was implemented experimentally in Ref. [78]. In contrast to a quantum dot coupled to a single reservoir, where

the only thing that happens is thermalization, we will find heat flows in the steady state and we will see how heat can be converted into work and how work can be used to refrigerate. The system we consider is a spinless, single-level quantum dot tunnel-coupled to two heat baths

$$\hat{H}_{\text{tot}} = \hat{H}_{\text{S}} + \hat{H}_{\text{c}} + \hat{H}_{\text{h}} + \hat{V}_{\text{c}} + \hat{V}_{\text{h}}, \tag{230}$$

with

$$\hat{H}_{\text{S}} = \varepsilon_{\text{d}}\hat{d}^\dagger\hat{d}, \qquad \hat{H}_\alpha = \sum_q \varepsilon_{\alpha,q}\hat{c}^\dagger_{\alpha,q}\hat{c}_{\alpha,q}, \qquad \hat{V}_\alpha = \hat{d}\sum_q g_{\alpha,q}\hat{c}^\dagger_{\alpha,q} - \hat{d}^\dagger\sum_q g^*_{\alpha,q}\hat{c}_{\alpha,q}, \tag{231}$$

where $\alpha =$ c, h labels the reservoirs according to their temperatures $T_{\text{c}} \leq T_{\text{h}}$. Just as for the quantum dot coupled to a single reservoir, Eq. (144) is already in GKLS form. Since the terms in the master equation corresponding to different reservoirs are additive, we find

$$\partial_t\hat{\rho}_{\text{S}} = -i[\hat{H}_{\text{S}}, \hat{\rho}_{\text{S}}] + \mathcal{L}_{\text{c}}\hat{\rho}_{\text{S}} + \mathcal{L}_{\text{h}}\hat{\rho}_{\text{S}}, \tag{232}$$

with

$$\mathcal{L}_\alpha\hat{\rho} = \kappa_\alpha[1 - n_{\text{F}}^\alpha(\varepsilon_{\text{d}})]\mathcal{D}[\hat{d}]\hat{\rho} + \kappa_\alpha n_{\text{F}}^\alpha(\varepsilon_{\text{d}})\mathcal{D}[\hat{d}^\dagger]\hat{\rho}, \tag{233}$$

where $n_{\text{F}}^\alpha$ is the Fermi-Dirac occupation with temperature $T_\alpha$ and chemical potential $\mu_\alpha$, see Eq. (216). Here we neglected the renormalization of $\varepsilon_{\text{d}}$ which is given by a straightforward generalization of Eq. (166).

### 5.1.1 Solving the master equation

The master equation can easily be solved by considering

$$\partial t\, p_1 = \text{Tr}\{\hat{d}^\dagger\hat{d}\,\partial_t\hat{\rho}_{\text{S}}\} = -\sum_{\alpha=\text{c,h}} \kappa_\alpha\left\{[1 - n_{\text{F}}^\alpha(\varepsilon_{\text{d}})]p_1 - n_{\text{F}}^\alpha(\varepsilon_{\text{d}})p_0\right\} = -\gamma(p_1 - \bar{n}), \tag{234}$$

where

$$\gamma = \kappa_{\text{c}} + \kappa_{\text{h}}, \qquad \bar{n} = \frac{\kappa_{\text{c}}n_{\text{F}}^{\text{c}}(\varepsilon_{\text{d}}) + \kappa_{\text{h}}n_{\text{F}}^{\text{h}}(\varepsilon_{\text{d}})}{\kappa_{\text{c}} + \kappa_{\text{h}}}. \tag{235}$$

Comparing to Eq. (171), we find that the quantum dot behaves just like a quantum dot coupled to a single heat bath with coupling strength $\gamma$ and mean occupation $\bar{n}$. The solution thus reads

$$p_1(t) = p_1(0)e^{-\gamma t} + \bar{n}(1 - e^{-\gamma t}). \tag{236}$$

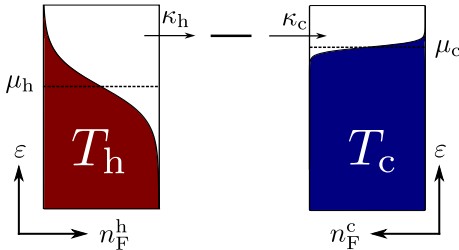

Figure 11: Quantum dot heat engine. A quantum dot is coupled both to a hot reservoir with temperature $T_{\text{h}}$ and chemical potential $\mu_{\text{h}}$ as well as a cold reservoir with temperature $T_{\text{c}}$ and chemical potential $\mu_{\text{c}}$. A temperature bias, $T_{\text{h}} > T_{\text{c}}$, allows for a particle current against a chemical potential bias ($\mu_{\text{h}} \neq \mu_{\text{c}}$, for electrons, this corresponds to a voltage bias). Thereby, heat is converted into chemical work.

### 5.1.2  The first law

From the master equation, we find the first law

$$\partial_t \langle \hat{H}_S \rangle = \varepsilon_d \text{Tr}\{\hat{d}^\dagger \hat{d} \mathcal{L}_c \hat{\rho}_S\} + \varepsilon_d \text{Tr}\{\hat{d}^\dagger \hat{d} \mathcal{L}_h \hat{\rho}_S\} = -J_c - P_c - J_h - P_h, \tag{237}$$

where the power and heat currents are defined in agreement with Eq. (190)

$$P_\alpha = -\mu_\alpha \text{Tr}\{\hat{d}^\dagger \hat{d} \mathcal{L}_\alpha \hat{\rho}_S\}, \qquad J_\alpha = -(\varepsilon_d - \mu_\alpha)\text{Tr}\{\hat{d}^\dagger \hat{d} \mathcal{L}_\alpha \hat{\rho}_S\}. \tag{238}$$

Explicitly, we find

$$P_\alpha = \mu_\alpha \kappa_\alpha e^{-\gamma t}[p_1(0) - \bar{n}] + \mu_\alpha \kappa_\alpha [\bar{n} - n_F^\alpha(\varepsilon_d)], \qquad J_\alpha = \frac{\varepsilon_d - \mu_\alpha}{\mu_\alpha} P_\alpha. \tag{239}$$

Just as for a single reservoir, there is a transient term in the power which decreases exponentially in time. In contrast to the single reservoir case, there is now also a time-independent term which remains in steady state.

In the steady state, the observables of the system do not change. We can use this fact to draw a number of conclusions without using the explicit solutions for the power and the heat currents. In particular, since the left-hand side of Eq. (237) vanishes, we find

$$\text{Tr}\{\hat{d}^\dagger \hat{d} \mathcal{L}_c \hat{\rho}_S\} = -\text{Tr}\{\hat{d}^\dagger \hat{d} \mathcal{L}_h \hat{\rho}_S\}. \tag{240}$$

From this, using Eqs. (238), follows

$$P = P_c + P_h = -(J_c + J_h), \tag{241}$$

which is nothing but the first law, as well as

$$\eta = \frac{P}{-J_h} = \frac{\mu_c - \mu_h}{\varepsilon_d - \mu_h} = 1 - \frac{\varepsilon_d - \mu_c}{\varepsilon_d - \mu_h}, \tag{242}$$

where we introduced the efficiency $\eta$ which is given by the ratio between the power (the output of the heat engine) and the heat current from the hot reservoir (the input of the heat engine). Using the explicit solution for the power in Eq. (239), we find

$$P = \frac{\kappa_c \kappa_h}{\kappa_c + \kappa_h}(\mu_c - \mu_h)[n_F^h(\varepsilon_d) - n_F^c(\varepsilon_d)]. \tag{243}$$

This quantity vanishes at zero voltage ($\mu_c = \mu_h$), as well as at the stopping voltage where $n_F^h(\varepsilon_d) = n_F^c(\varepsilon_d)$, see also Fig. 13 b).

Let us now consider under which conditions the system acts as a heat engine, i.e., heat from the hot reservoir is converted into power. From Eq. (243), we can identify different regimes depending on the signs of $P$, $J_c$, and $J_h$. These regimes are illustrate in Figs. 12 and 13 (a). For $\mu_c \geq \mu_h$ ($\mu_c \leq \mu_h$), we find that the quantum dot acts as a heat engine for large positive (negative) values of $\varepsilon_d$. In both cases, we find from Eq. (243) that power is positive as long as

$$\frac{\varepsilon_d - \mu_c}{\varepsilon_d - \mu_h} \geq \frac{T_c}{T_h} \quad \Rightarrow \quad \eta \leq 1 - \frac{T_c}{T_h} = \eta_C. \tag{244}$$

We thus find that the efficiency is bounded from above by the Carnot efficiency as long as the power output is non-negative.

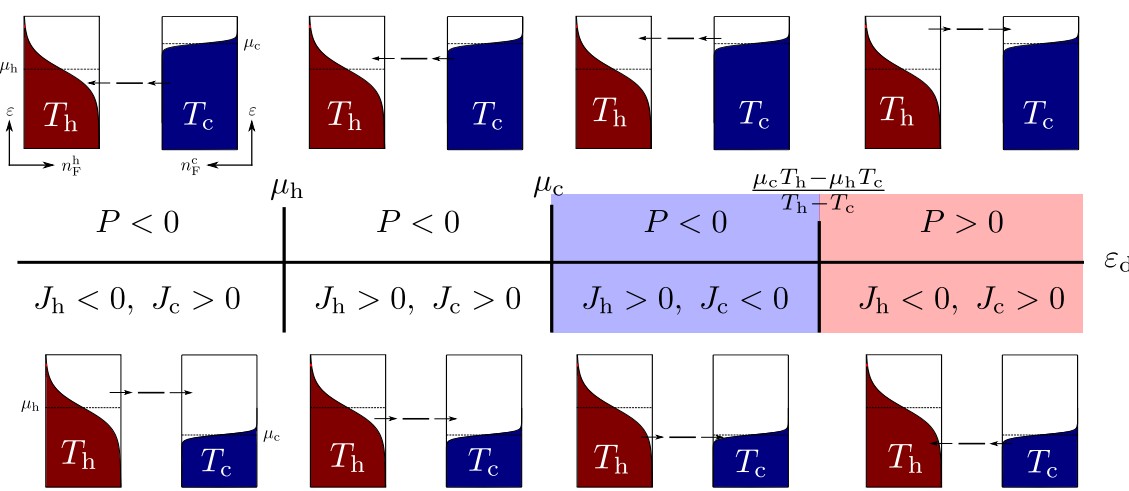

Figure 12: Different regimes determined by the signs of $P$, $J_c$, and $J_h$. For $\mu_c \geq \mu_h$, $\varepsilon_d$ increases from left to right (illustrated by the cartoons in the upper row). For $\mu_c \leq \mu_h$, we find the exact same regimes but $\varepsilon_d$ decreases from left to right (illustrated by the cartoons in the lower row). In the regime to the far left, heat flows out of the hot reservoir and into the cold reservoir and power is dissipated. If $\varepsilon_d$ lies in between the chemical potentials, both reservoirs are heated up by the dissipated power. Refrigeration is obtained in the blue shaded regime, where heat is extracted from the cold reservoir and dumped into the hot reservoir. Finally, in the red shaded regime the quantum dot acts as a heat engine where a heat flow from hot to cold drives a charge flow against the external voltage bias.

### 5.1.3 The second law

We first make some general statements about the second law in a two-terminal setup. These are very similar to the statements made in Sec. 1.2. The entropy production rate is given by

$$\dot{\Sigma} = k_B \partial_t S_{vN}[\hat{\rho}_S] + \frac{J_c}{T_c} + \frac{J_h}{T_h} \geq 0. \tag{245}$$

In the steady state, the first term vanishes and we immediately find that at least one of the heat currents has to be positive. This implies that it is impossible to cool down all reservoirs at the same time (in the steady state). Furthermore, for equal temperatures ($T_c = T_h$), we find $P = -(J_c + J_h) \leq 0$ which implies that it is not possible to convert heat into work with reservoirs at a single temperature. This is known as the Kelvin-Planck statement of the second law. Finally, we can use the first law in Eq. (241) to eliminate $J_c$, resulting in

$$\dot{\Sigma} = \frac{P}{T_c}\frac{\eta_C - \eta}{\eta} \quad \Rightarrow \quad 0 \leq \eta \leq \eta_C, \tag{246}$$

where the last inequality holds for $P \geq 0$. The fact that the efficiency is upper bounded by the Carnot efficiency is thus a direct consequence of the second law.

In our system, the entropy of the quantum dot is given in Eq. (176). Using

$$\partial_t p_1 = -\frac{J_c}{\varepsilon_d - \mu_c} - \frac{J_h}{\varepsilon_d - \mu_h}, \tag{247}$$

we can write the entropy production rate as

$$\dot{\Sigma} = k_B \sum_{\alpha=c,h} \kappa_\alpha \Big( p_1[1 - n_F^\alpha(\varepsilon_d)] - [1 - p_1]n_F^\alpha(\varepsilon_d) \Big) \ln\left( \frac{p_1[1 - n_F^\alpha(\varepsilon_d)]}{[1 - p_1]n_F^\alpha(\varepsilon_d)} \right) \geq 0, \tag{248}$$

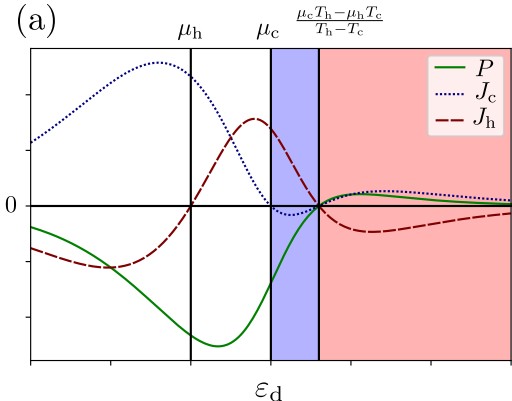
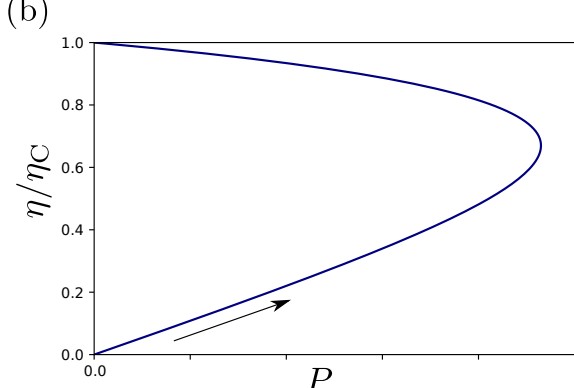

Figure 13: Performance of the quantum dot heat engine. (a) Power and heat currents as a function of $\varepsilon_d$. The regimes illustrated in Fig. 12 can clearly be identified. (b) "Lasso" diagram. Along the curve, the voltage bias is increased in the direction of the arrow. For $\mu_c = \mu_h$, both the power as well as the efficiency vanishes. Increasing $\mu_c$ then results in a finite power and efficiency, until the power vanishes again at the stopping voltage, where the efficiency reaches $\eta_C$. Such plots are called Lasso diagrams because the curve usually goes back to the origin at the stopping voltage, see also Fig. 14. Parameters: $\kappa_c = \kappa_h$, $\varepsilon_d = 2\kappa_c$, $k_B T_c = 0.3\kappa_c$, $k_B T_h = 0.8\kappa_c$, $\mu_c = \kappa_c$, $\mu_h = 0$.

which is positive since each term in the sum is positive in complete analogy to Eq. (178). In the steady state, we find

$$\dot{\Sigma} = k_B \frac{\kappa_c \kappa_h}{\kappa_c + \kappa_h}[n_F^h(\varepsilon_d) - n_F^c(\varepsilon_d)][\beta_c(\varepsilon_d - \mu_c) - \beta_h(\varepsilon_d - \mu_h)] \geq 0, \tag{249}$$

which vanishes at the Carnot point, where $n_F^h(\varepsilon_d) = n_F^c(\varepsilon_d)$, $\eta = \eta_C$, and both the power as well as the heat currents vanish. We stress that while an equilibrium situation (i.e., $T_c = T_h$ and $\mu_c = \mu_h$) ensures $n_F^h(\varepsilon_d) = n_F^c(\varepsilon_d)$, the Carnot point can also be reached out of equilibrium.

The interplay between power and efficiency is illustrated in Fig. 13. We find that at maximum power, the efficiency reaches above **60%** of the Carnot efficiency. Similar values where found experimentally in Ref. [78]. As mentioned above, the Carnot efficiency is obtained at the stopping voltage. This is a consequence of the fact that there is only a single energy at which transport happens. At the stopping voltage, all transport is blocked implying that both the charge as well as the heat currents vanish. This implies that there is no dissipation ($\dot{\Sigma} = 0$) and the efficiency takes on the Carnot value (see also Ref. [79]). In reality, as well as in the experiment of Ref. [78], this ideal filtering effect is spoiled by the broadening of the energy level which is neglected in the Markovian master equation, see Fig. 14. Including this energy broadening, one finds that at some energies, charges move from hold to cold while at other energies, they move in the other direction. At the stopping voltage, this results in a vanishing of power but still a net heat current, such that the efficiency vanishes. In this case, Fig. 13 (b) takes on the shape of a lasso.

### 5.1.4 Refrigeration

As discussed above, the quantum dot can also act as a refrigerator in the regime where $P < 0$, $J_h > 0$, and $J_c < 0$. In this case, electrical power is used to reverse the natural flow of heat, resulting in a heat flow out of the cold reservoir and into the hot reservoir. The efficiency of

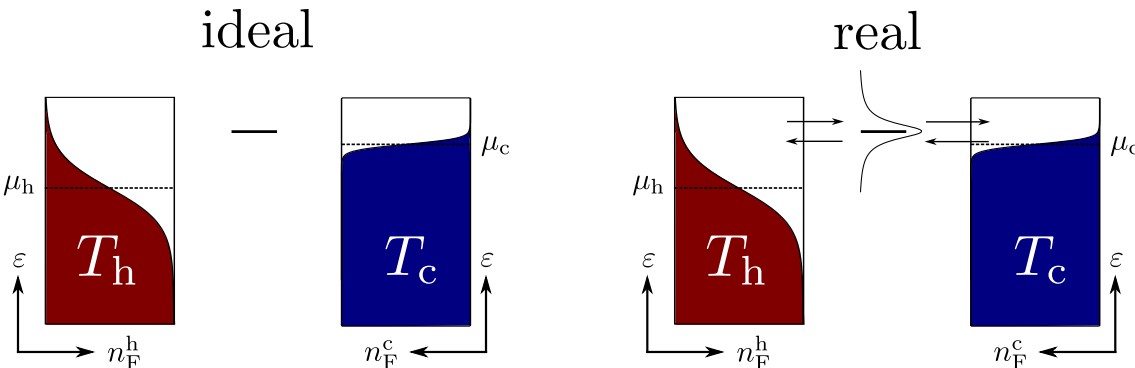

Figure 14: Effect of level broadening. At the Carnot point, where $n_F^h(\varepsilon_d) = n_F^c(\varepsilon_d)$, our Markovian treatment predicts $P = J_h = 0$ and $\eta = \eta_C$. This occurs because the quantum dot is approximated as being a perfect energy filter and no transport occurs at the Carnot point as illustrated in the left panel. In reality, the energy level of the quantum dot is broadened. Above $\varepsilon_d$, $n_F^h(\varepsilon) > n_F^c(\varepsilon)$ and we get a particle current from hot to cold. Below $\varepsilon_d$, $n_F^h(\varepsilon) < n_F^c(\varepsilon)$ resulting in a particle current from cold to hot. These particle currents cancel, resulting in $P = 0$. However, since the particles above and below $\varepsilon_d$ have a different energy, their contribution to the heat current does not cancel resulting in $-J_h > 0$ which implies $\eta = 0$. If the Markovian approximation is justified, $J_h$ is small. However, it remains finite which always implies $\eta = 0$ at the stopping voltage.

this process is usually characterized by the coefficient of performance (COP)

$$\eta^{COP} = \frac{-J_c}{-P}, \tag{250}$$

where we left the minus signs to stress that this performance quantifier is relevant in the regime where both $P$ as well as $J_c$ are negative under our sign convention. We can use the first law in Eq. (241) to eliminate $J_h$ and write the entropy production rate as

$$\dot{\Sigma} = \frac{-J_c}{T_h} \frac{\eta_C^{COP} - \eta^{COP}}{\eta_C^{COP} \eta^{COP}} \quad \Rightarrow \quad 0 \leq \eta^{COP} \leq \eta_C^{COP}, \tag{251}$$

where the second inequality holds for $J_c \leq 0$ and we introduced the Carnot value for the COP

$$\eta_C^{COP} = \frac{T_c}{T_h - T_c}. \tag{252}$$

We note that as $T_c \to T_h$, $\eta_C^{COP}$ diverges. This reflects the fact that in principle, it is possible to move heat in between two reservoirs with equal temperature without investing any work.

In our system, we find from Eqs. (238) and (240)

$$\eta^{COP} = \frac{\varepsilon_d - \mu_c}{\mu_c - \mu_h}. \tag{253}$$

From this, we find that $\eta^{COP}$ vanishes when $\varepsilon_d = \mu_c$ and takes on the Carnot value at $\varepsilon_d = \frac{\mu_c T_h - \mu_h T_c}{T_h - T_c}$, which is exactly the point where the regime of the refrigerator meets the regime of the heat engine, see Fig. 12. Interestingly, both the COP as well as the efficiency reach their maximum value at this point, where no transport takes place. The Carnot point is often called the point of *reversibility*. At this point nothing happens but it can be seen as the limit of converting heat into work infinitely slowly and without wasting any energy (thus, reversibly). Equivalently, taking the limit from the other side, it can be seen as the limit of cooling the cold reservoir reversibly.

## 5.2 Entanglement generator

In this section, we consider a thermal machine that uses a temperature gradient in order to produce entanglement between two quantum dots. The original idea goes back to Ref. [80], where qubits instead of quantum dots are considered. Here we focus on the scenario investigated in Ref. [70], i.e., a double quantum dot coupled to two fermionic reservoirs, just like in Sec. 4.6 (see Fig. 10).

### 5.2.1 Entanglement

Before we consider the thermal machine itself, we provide a brief introduction to entanglement which is one of the most intriguing features of quantum mechanics. For more information, see the Book by Nielsen and Chuang [17]. To consider entanglement, we require a bi-partite Hilbert space $\mathcal{H} = \mathcal{H}_A \otimes \mathcal{H}_B$, where $\mathcal{H}_A$ denotes the Hilbert space of Alice and $\mathcal{H}_B$ is Bob's Hilbert space. A quantum state on this bi-partite Hilbert space is then said to be a product state if it reads $\hat{\rho} = \hat{\rho}_A \otimes \hat{\rho}_B$. This corresponds to the scenario where Alice and Bob have access to their respective states $\hat{\rho}_A$ and $\hat{\rho}_B$, without any correlations between them. A classical mixture of product states is called a separable state

$$\hat{\rho} = \sum_j p_j \hat{\rho}_A^j \otimes \hat{\rho}_B^j, \qquad p_j \geq 0, \qquad \sum_j p_j = 1. \qquad (254)$$

Such a state contains classical correlations. For instance, it can describe the rather trivial example where Alice and Bob by fruit together. With equal probabilities, they buy two apples or two oranges. If Alice has an apple, we know Bob has an apple as well and similarly for oranges. Obviously, no fruits are entangled in this example.

A state is entangled iff it is not separable, i.e., if it cannot be written in the form of Eq. (254). In this case, the correlations between Alice and Bob are no longer classical in nature. Entanglement may thus be seen as a form of correlation that goes beyond classical correlations. Here we illustrate this with two simple examples. First, we consider the state

$$\hat{\rho}_{cl} = \frac{1}{2} \left( |00\rangle\langle00| + |11\rangle\langle11| \right) = \frac{1}{2} |0\rangle\langle0| \otimes |0\rangle\langle0| + \frac{1}{2} |1\rangle\langle1| \otimes |1\rangle\langle1|, \qquad (255)$$

which is evidently separable. It corresponds to the apple-orange scenario above, where $|00\rangle$ could denote an apple for both Alice and Bob. As an example for an entangled state, we consider one of the Bell states

$$|\Phi^+\rangle = \frac{1}{\sqrt{2}} \left( |00\rangle + |11\rangle \right), \qquad |\Phi^+\rangle\langle\Phi^+| = \frac{1}{2} \left( |00\rangle\langle00| + |11\rangle\langle11| + |00\rangle\langle11| + |11\rangle\langle00| \right). \qquad (256)$$

This state cannot be written in the form of Eq. (254) due to the off-diagonal elements. Indeed, it is the maximally entangled state for two qubits.

The amount of entanglement can be quantified by the *entanglement of formation* [81]. Loosely speaking, it is determined by the number of Bell states that are required to prepare the given state using only local operations and classical communication (LOCC). For two qubits, the entanglement of formation is a number between zero and one, where zero is obtained for separable states and one for Bell states.

**Concurrence:** Determining if a given state is entangled is in general a highly non-trivial task. However, for two qubits the low dimensionality of the problem considerably facilitates the task. A common measure for entanglement in this scenario is the *concurrence* [82, 83],

which is monotonically related to the entanglement of formation. Just as the latter, the concurrence ranges from zero, obtained for separable states, to one, reached for Bell states. The concurrence of a state $\hat{\rho}$ can be computed with the help of the auxiliary state

$$\tilde{\rho} = \hat{\sigma}_y \otimes \hat{\sigma}_y \hat{\rho}^* \hat{\sigma}_y \otimes \hat{\sigma}_y, \qquad \hat{\sigma}_y = -i\,|0\rangle\langle 1| + i\,|1\rangle\langle 0|, \tag{257}$$

where the star denotes complex conjugation in the computational basis ($|0\rangle, |1\rangle$). Let $\lambda_j$ denote the eigenvalues of $\hat{\rho}\tilde{\rho}$ in decreasing order, i.e., $\lambda_1 \geq \lambda_2 \geq \lambda_3 \geq \lambda_4$. The concurrence may then be written as

$$C[\hat{\rho}] = \max\left\{0, \sqrt{\lambda_1} - \sqrt{\lambda_2} - \sqrt{\lambda_3} - \sqrt{\lambda_4}\right\}. \tag{258}$$

This quantity lies between zero and one, where $C[\hat{\rho}] = 0$ holds for separable states and $C[\hat{\rho}] > 0$ for entangled states, with unity reached only for Bell states.

**Fermionic entanglement:** The discussion above assumed that the Hilbert space has the tensor-product structure $\mathcal{H} = \mathcal{H}_A \otimes \mathcal{H}_B$. For fermions, this is not the case because fermionic operators corresponding to Alice and Bob *anti*-commute. This implies that any definition for entanglement needs to be reconsidered for fermions [84]. To make progress, we denote by $|00\rangle$ the vacuum state, where no fermion is present. Using the creation operators on the dots $\hat{d}_\alpha^\dagger$, we may then define the states

$$|10\rangle = \hat{d}_L^\dagger |00\rangle, \qquad |01\rangle = \hat{d}_R^\dagger |00\rangle, \qquad |11\rangle = \hat{d}_L^\dagger \hat{d}_R^\dagger |00\rangle = -\hat{d}_R^\dagger \hat{d}_L^\dagger |00\rangle. \tag{259}$$

Note the minus sign in the last equation, which forces us to associate to the state $|11\rangle$ a specific order for the fermionic operators. It turns out that in our system, we may compute the concurrence as if these states were two-qubit states. This is the case because we will only find a single off-diagonal element corresponding to a coherent superposition of a single electron between the two modes (i.e., between $|01\rangle$ and $|10\rangle$). In general, more care has to be taken when evaluating the entanglement of fermionic systems (even for two modes) [84].

### 5.2.2 The master equation

We will use the master equation in the singular-coupling limit (local master equation) given in Eq. (225), neglecting any Lamb-shift. For convenience, we reproduce this equation here

$$\partial_t \hat{\rho}_S(t) = -i[\hat{H}_S, \hat{\rho}_S(t)] + \mathcal{L}_L \hat{\rho}_S(t) + \mathcal{L}_R \hat{\rho}_S(t), \tag{260}$$

with the local dissipators

$$\mathcal{L}_\alpha = \kappa_\alpha n_F^\alpha(\varepsilon)\mathcal{D}[\hat{d}_\alpha^\dagger] + \kappa_\alpha[1 - n_F^\alpha(\varepsilon)]\mathcal{D}[\hat{d}_\alpha], \tag{261}$$

and the Hamiltonian

$$\hat{H}_S = \varepsilon\left(\hat{d}_L^\dagger \hat{d}_L + \hat{d}_R^\dagger \hat{d}_R\right) + g\left(\hat{d}_L^\dagger \hat{d}_R + \hat{d}_R^\dagger \hat{d}_L\right). \tag{262}$$

As discussed in Sec. 4.6, this restricts our analysis to $g \ll \max\{k_B T_\alpha, |\varepsilon - \mu_\alpha|\}$. For a discussion of the entanglement generator outside of this regime, see Ref. [70].

Since this master equation is bi-linear and contains one annihilation and one creation operator per term, the time-derivatives $\partial_t \langle \hat{d}_\alpha^\dagger \hat{d}_\beta \rangle$ form a closed set of equations. Furthermore, the quantities $\langle \hat{d}_\alpha^\dagger \hat{d}_\beta \rangle$ completely determine the quantum state of the system at all times (given that this is true for the initial state). The quantum state is said to be Gaussian. Expectation values including more than two annihilation or creation operators can then always be reduced using Wick's theorem [85, 86]. In particular, for Gaussian states it holds that

$$\langle \hat{d}_i^\dagger \hat{d}_j^\dagger \hat{d}_k \hat{d}_l \rangle = \langle \hat{d}_i^\dagger \hat{d}_l \rangle \langle \hat{d}_j^\dagger \hat{d}_k \rangle - \langle \hat{d}_i^\dagger \hat{d}_k \rangle \langle \hat{d}_j^\dagger \hat{d}_l \rangle. \tag{263}$$

For more information on solving master equations for Gaussian processes, see Ref. [18].

From the master equation in Eq. (260), one may derive

$$\partial_t \vec{v} = A\vec{v} + \vec{b}, \tag{264}$$

with

$$\vec{v} = \begin{pmatrix} \langle \hat{d}_{\mathrm{L}}^\dagger \hat{d}_{\mathrm{L}} \rangle \\ \langle \hat{d}_{\mathrm{R}}^\dagger \hat{d}_{\mathrm{R}} \rangle \\ \langle \hat{d}_{\mathrm{L}}^\dagger \hat{d}_{\mathrm{R}} \rangle \\ \langle \hat{d}_{\mathrm{R}}^\dagger \hat{d}_{\mathrm{L}} \rangle \end{pmatrix}, \quad A = \begin{pmatrix} -\kappa_{\mathrm{L}} & 0 & -ig & ig \\ 0 & -\kappa_{\mathrm{R}} & ig & -ig \\ -ig & ig & -\frac{\kappa_{\mathrm{L}}+\kappa_{\mathrm{R}}}{2} & 0 \\ ig & -ig & 0 & -\frac{\kappa_{\mathrm{L}}+\kappa_{\mathrm{R}}}{2} \end{pmatrix}, \quad \vec{b} = \begin{pmatrix} \kappa_{\mathrm{L}} n_{\mathrm{F}}^{\mathrm{L}}(\varepsilon) \\ \kappa_{\mathrm{R}} n_{\mathrm{F}}^{\mathrm{R}}(\varepsilon) \\ 0 \\ 0 \end{pmatrix}. \tag{265}$$

At steady state, we may set the LHS of Eq. (264) to zero and we find

$$0 = A\vec{v}_{\mathrm{ss}} + \vec{b}, \quad \Rightarrow \quad \vec{v}_{\mathrm{ss}} = -A^{-1}\vec{b}. \tag{266}$$

In order to connect the averages in $\vec{v}$ to the state of the double quantum dot, we write

$$\hat{\rho}_{\mathrm{S}} = \sum_{\substack{n_{\mathrm{L}}, n_{\mathrm{R}} \\ n_{\mathrm{L}}', n_{\mathrm{R}}'}} \rho_{n_{\mathrm{L}}', n_{\mathrm{R}}'}^{n_{\mathrm{L}}, n_{\mathrm{R}}} |n_{\mathrm{L}}, n_{\mathrm{R}}\rangle\langle n_{\mathrm{L}}', n_{\mathrm{R}}'|. \tag{267}$$

With the basis states given in Eq. (259), the matrix elements may be cast into

$$\begin{aligned} \rho_{n_{\mathrm{L}}', n_{\mathrm{R}}'}^{n_{\mathrm{L}}, n_{\mathrm{R}}} &= \langle n_{\mathrm{L}}, n_{\mathrm{R}}| \hat{\rho}_{\mathrm{S}} |n_{\mathrm{L}}', n_{\mathrm{R}}'\rangle = \mathrm{Tr}\left\{ \left(\hat{d}_{\mathrm{L}}^\dagger\right)^{n_{\mathrm{L}}'} \left(\hat{d}_{\mathrm{R}}^\dagger\right)^{n_{\mathrm{R}}'} |00\rangle\langle 00| \hat{d}_{\mathrm{R}}^{n_{\mathrm{R}}} \hat{d}_{\mathrm{L}}^{n_{\mathrm{L}}} \hat{\rho}_{\mathrm{S}} \right\} \\ &= \left\langle \left(\hat{d}_{\mathrm{L}}^\dagger\right)^{n_{\mathrm{L}}'} \left(\hat{d}_{\mathrm{R}}^\dagger\right)^{n_{\mathrm{R}}'} \left(1 - \hat{d}_{\mathrm{L}}^\dagger \hat{d}_{\mathrm{L}}\right) \left(1 - \hat{d}_{\mathrm{R}}^\dagger \hat{d}_{\mathrm{R}}\right) \hat{d}_{\mathrm{R}}^{n_{\mathrm{R}}} \hat{d}_{\mathrm{L}}^{n_{\mathrm{L}}} \right\rangle. \end{aligned} \tag{268}$$

Here we used $|00\rangle\langle 00| = \left(1 - \hat{d}_{\mathrm{L}}^\dagger \hat{d}_{\mathrm{L}}\right)\left(1 - \hat{d}_{\mathrm{R}}^\dagger \hat{d}_{\mathrm{R}}\right)$ which may be verified by applying this operator to all basis states. Note that due to the fermionic anti-commutation relations, and because there is no coherence between states with a different total number of electrons, the only averages required to compute the density matrix that are not of the form $\langle \hat{d}_\alpha^\dagger \hat{d}_\beta \rangle$ can be reduced by Eq. (263).

A lengthy but straightforward calculation then results in the steady state

$$\begin{aligned} \hat{\rho}_{\mathrm{ss}} = \frac{1}{4g^2 + \kappa_{\mathrm{L}}\kappa_{\mathrm{R}}} \Bigg\{ &\kappa_{\mathrm{L}}\kappa_{\mathrm{R}} \frac{e^{-\beta_{\mathrm{L}}(\varepsilon - \mu_{\mathrm{L}})\hat{d}_{\mathrm{L}}^\dagger \hat{d}_{\mathrm{L}}} e^{-\beta_{\mathrm{R}}(\varepsilon - \mu_{\mathrm{R}})\hat{d}_{\mathrm{R}}^\dagger \hat{d}_{\mathrm{R}}}}{Z_{\mathrm{L}} Z_{\mathrm{R}}} + 4g^2 \frac{e^{-\bar{\beta}(\varepsilon - \bar{\mu})(\hat{d}_{\mathrm{L}}^\dagger \hat{d}_{\mathrm{L}} + \hat{d}_{\mathrm{R}}^\dagger \hat{d}_{\mathrm{R}})}}{\bar{Z}^2} \\ &- \frac{2g\,\kappa_{\mathrm{L}}\kappa_{\mathrm{R}}(n_{\mathrm{F}}^{\mathrm{L}} - n_{\mathrm{F}}^{\mathrm{R}})}{\kappa_{\mathrm{L}} + \kappa_{\mathrm{R}}} i\left(\hat{d}_{\mathrm{L}}^\dagger \hat{d}_{\mathrm{R}} - \hat{d}_{\mathrm{R}}^\dagger \hat{d}_{\mathrm{L}}\right) \Bigg\}, \end{aligned} \tag{269}$$

where we omitted the argument of the Fermi functions for brevity and the barred quantities are defined by the equation

$$\bar{n} = \frac{1}{e^{\bar{\beta}(\varepsilon - \bar{\mu})} + 1} = \frac{\kappa_{\mathrm{L}} n_{\mathrm{F}}^{\mathrm{L}}(\varepsilon) + \kappa_{\mathrm{R}} n_{\mathrm{F}}^{\mathrm{R}}(\varepsilon)}{\kappa_{\mathrm{L}} + \kappa_{\mathrm{R}}}, \tag{270}$$

and we abbreviated the partition functions

$$Z_\alpha = \mathrm{Tr}\left\{ e^{-\beta_\alpha(\varepsilon - \mu_\alpha)\hat{d}_\alpha^\dagger \hat{d}_\alpha} \right\}, \qquad \bar{Z} = \mathrm{Tr}\left\{ e^{-\bar{\beta}(\varepsilon - \bar{\mu})\hat{d}_\alpha^\dagger \hat{d}_\alpha} \right\}, \tag{271}$$

where in the last equality, the choice of $\alpha$ does not matter. Note that the terms in the first line of Eq. (269) are both product states. Thus, it is the second line, including the coherence factor $\hat{d}_{\mathrm{L}}^\dagger \hat{d}_{\mathrm{R}} - \hat{d}_{\mathrm{R}}^\dagger \hat{d}_{\mathrm{L}} = |10\rangle\langle 01| - |01\rangle\langle 10|$, which is responsible for any entanglement we might get.

**Limiting cases:** It is instructive to consider the steady state in some limiting cases. First, we consider an equilibrium scenario where $\beta_L = \beta_R = \bar{\beta}$ and $\mu_L = \mu_R = \bar{\mu}$. In this case, the second line of Eq. (269) vanishes and the first two terms become proportional to each other. We thus find, as expected from Eq. (197)

$$\hat{\rho}_{ss} = \frac{e^{-\bar{\beta}(\hat{H}_{TD} - \bar{\mu}\hat{N}_S)}}{Z}, \tag{272}$$

with the thermodynamic Hamiltonian [64] and particle-number operator

$$\hat{H}_{TD} = \varepsilon \left( \hat{d}_L^\dagger \hat{d}_L + \hat{d}_R^\dagger \hat{d}_R \right), \qquad \hat{N}_S = \hat{d}_L^\dagger \hat{d}_L + \hat{d}_R^\dagger \hat{d}_R. \tag{273}$$

Note that this is a product state due to the additive nature of $\hat{H}_{TD}$. In the singular-coupling master equation considered here, the thermal state does not exhibit any entanglement. This is different at large coupling $g$, where the secular approximation should be used instead [70].

Second, we consider the limit where $\kappa_L, \kappa_R \gg g$. In this case, only the first term in Eq. (269) is relevant. This term describes two quantum dots, each in equilibrium with the reservoir it couples to. The state is thus a product of Gibbs states with the local reservoir temperatures and chemical potentials.

The third case we consider is the case where $\kappa_L = 0$. In this case, we find that only the second term in Eq. (269) survives with $\bar{\beta} = \beta_R$ and $\bar{\mu} = \mu_R$. The system thus equilibrates with the right reservoir, taking the form of Eq. (272).

Finally, we consider the limit $g \gg \kappa_L, \kappa_R$. Just as in the last case, the second term in Eq. (269) dominates and the system equilibrates to the average occupation $\bar{n}$ with the state reducing to Eq. (272). In all these limiting cases, no entanglement is found as the term involving coherence always drops out.

**Concurrence:** To quantify the entanglement in Eq. (269), we consider the concurrence. The steady state given in Eq. (269) is of the form

$$\hat{\rho}_{ss} = \begin{pmatrix} p_0 & 0 & 0 & 0 \\ 0 & p_L & \alpha & 0 \\ 0 & \alpha^* & p_R & 0 \\ 0 & 0 & 0 & p_d \end{pmatrix}, \tag{274}$$

where the basis states are $|00\rangle$, $|10\rangle$, $|01\rangle$, $|11\rangle$. For such a state, the concurrence reduces to $C[\hat{\rho}_{ss}] = \max\{0, 2|\alpha| - 2\sqrt{p_0 p_d}\}$. From Eq. (269), we find

$$2|\alpha| - 2\sqrt{p_0 p_d} = \frac{2\kappa_L \kappa_R}{4g^2 + \kappa_L \kappa_R} \left\{ \frac{2|g||n_F^R - n_F^L|}{\kappa_L + \kappa_R} \right. $$
$$\left. - \sqrt{\left[ (1 - n_F^L)(1 - n_F^R) + \frac{4g^2}{\kappa_L \kappa_R}(1 - \bar{n})^2 \right] \left[ n_F^L n_F^R + \frac{4g^2}{\kappa_L \kappa_R}\bar{n}^2 \right]} \right\}. \tag{275}$$

We thus find that entanglement is indeed generated. Interestingly, the entanglement is generated due to reservoirs which are out of equilibrium, as can be inferred from the term $|n_F^R - n_F^L|$. This is a surprising result because usually, any coupling to the environment only reduces coherence and entanglement in the system. This is because entanglement is *monogamous*, which means that if A is strongly entangled with B, it cannot at the same time be strongly entangled with C [87]. When a system couples to its environment, this results in entanglement between the system and the reservoirs, which generally reduces the inter-system entanglement. Here we encounter a different behavior, where an out-of-equilibrium environment induces entanglement within a system.

**Heat current:** It is instructive to consider the heat current, which is given by

$$J_\text{R} = -\text{Tr}\left\{\left(\hat{H}_\text{TD} - \mu_\text{R}\hat{N}_\text{S}\right)\mathcal{L}_\text{R}\hat{\rho}_\text{ss}\right\} = (\varepsilon - \mu_\text{R})\frac{4g^2\kappa_\text{L}\kappa_\text{R}(n_\text{F}^\text{L} - n_\text{F}^\text{R})}{(\kappa_\text{L} + \kappa_\text{R})(4g^2 + \kappa_\text{L}\kappa_\text{R})}, \tag{276}$$

with the thermodynamic Hamiltonian given in Eq. (273). Noting the similarity of this expression with the first term in the concurrence, see Eq. (275), we find that the system is entangled when the heat current exceeds a critical value [88]

$$|J_\text{R}| \geq |\varepsilon - \mu_\text{R}||g|\frac{2\kappa_\text{L}\kappa_\text{R}}{4g^2 + \kappa_\text{L}\kappa_\text{R}}\sqrt{\left[\left(1 - n_\text{F}^\text{L}\right)\left(1 - n_\text{F}^\text{R}\right) + \frac{4g^2}{\kappa_\text{L}\kappa_\text{R}}\left(1 - \bar{n}\right)^2\right]\left[n_\text{F}^\text{L}n_\text{F}^\text{R} + \frac{4g^2}{\kappa_\text{L}\kappa_\text{R}}\bar{n}^2\right]}. \tag{277}$$

When we can trust the master equation we employ to be a valid description, the heat current may thus serve as an indicator for entanglement.

**Maximal entanglement:** The concurrence in Eq. (275) can be maximized over all parameters, which results in

$$C[\hat{\rho}_\text{ss}] = \frac{\sqrt{5} - 1}{4} \simeq 0.309. \tag{278}$$

This value is obtained by the parameters $n_\text{F}^\text{L} = 1$, $n_\text{F}^\text{R} = 0$, $\kappa_\text{L} = \kappa_\text{R}$, $g/\kappa_\text{L} = (\sqrt{5} - 1)/4$. The Fermi functions should thus be maximally different, which results in the largest current and off-diagonal density matrix element. The couplings to the reservoirs should be equal and interestingly, the ratio between the interdot coupling and the system-bath coupling is determined by the golden ratio $\varphi$. Indeed, we find

$$\varphi \equiv \frac{1 + \sqrt{5}}{2} = \frac{\kappa_\text{L}}{2g} = \frac{1}{2C} = \frac{|\alpha|}{2\sqrt{p_0 p_\text{d}}}. \tag{279}$$

As shown in Ref. [89], the steady state for these parameters is not only entangled but can also be used for quantum teleportation. The entanglement is further enhanced when including electron-electron interactions, which may even result in nonlocal states that violate Bell inequalities [70].

## 5.3 Absorption refrigerator

As a final example of a thermal machine, we consider a quantum absorption refrigerator, which is a refrigerator that uses heat as its energy source [14, 90, 91]. Quantum absorption refrigerators have been implemented experimentally, using trapped ions [92] and very recently using superconducting qudits [93]. The system we consider here consists of three two-level systems (qubits), coupled to three reservoirs respectively, see Fig. 15, with temperatures $T_\text{c} \leq T_\text{r} \leq T_\text{h}$. Here the subscript r denotes an intermediate *room* temperature. The basic principle behind the absorption refrigerator is that the tendency from heat to flow from the hot reservoir to the room reservoir is exploited to drive a heat current from the cold reservoir to the room reservoir. Thereby, a hot thermal reservoir is used to cool down the coldest reservoir. At the same time, the qubit coupled to the cold reservoir is cooled below $T_\text{c}$. The absorption refrigerator may thereby achieve two goals:

1. Cooling the cold reservoir.

2. Cooling the qubit coupled to the cold reservoir.

Which of these goals is more relevant may depend on the specific circumstances. As we see below, different figures of merit can be used to describe the performance of the refrigerator in reaching the different goals.

We stress that in contrast to the previous examples, we do not consider quantum dots where fermions are exchanged with the environment. Instead, we consider qubits that can exchange photons with the environment. We therefore do not have chemical potentials in this section (since the $\mu = 0$ for photons).

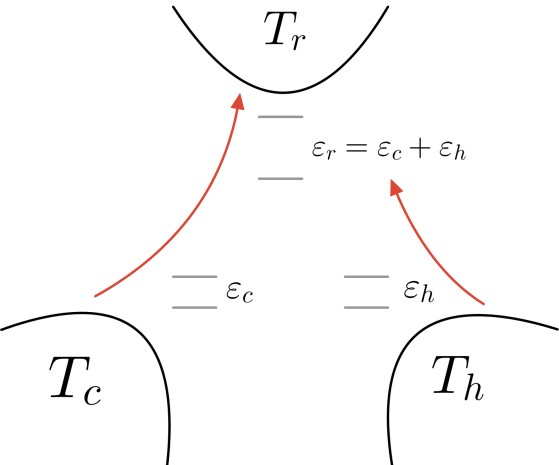

Figure 15: Sketch of a three-qubit absorption refrigerator. Three two-level systems (qubits) are coupled to three thermal reservoirs at temperatures $T_c \leq T_r \leq T_h$. A heat flow from the hot reservoir to the room reservoir induces a heat flow from the cold reservoir to the room reservoir, thereby extracting heat from the coldest reservoir. At the same time, the qubit coupled to the cold reservoir is cooled to a temperature below $T_c$.

### 5.3.1 The master equation

The three qubits are described by the Hamiltonian

$$\hat{H}_S = \hat{H}_0 + \hat{H}_{\text{int}}, \tag{280}$$

where the first term describes the individual qubits

$$\hat{H}_0 = \varepsilon_c \hat{\sigma}_c^\dagger \hat{\sigma}_c + \varepsilon_h \hat{\sigma}_h^\dagger \hat{\sigma}_h + \varepsilon_r \hat{\sigma}_r^\dagger \hat{\sigma}_r, \tag{281}$$

and the second term describes their interaction

$$\hat{H}_{\text{int}} = g \left( \hat{\sigma}_r^\dagger \hat{\sigma}_c \hat{\sigma}_h + \hat{\sigma}_c^\dagger \hat{\sigma}_h^\dagger \hat{\sigma}_r \right) = g \left( |001\rangle\langle 110| + |110\rangle\langle 001| \right). \tag{282}$$

Here we introduced the lowering operators

$$\hat{\sigma}_c = |0\rangle\langle 1| \otimes \mathbb{1}_2 \otimes \mathbb{1}_2, \quad \hat{\sigma}_h = \mathbb{1}_2 \otimes |0\rangle\langle 1| \otimes \mathbb{1}_2, \quad \hat{\sigma}_c = \mathbb{1}_2 \otimes \mathbb{1}_2 \otimes |0\rangle\langle 1|. \tag{283}$$

From the interaction Hamiltonian in Eq. (282), we may anticipate the cooling mechanism: Two excitations, one in the hot qubit and one in the cold qubit, are turned into an exciation in the room qubit by the term $\hat{\sigma}_r^\dagger \hat{\sigma}_c \hat{\sigma}_h$. This excitation is then dissipated into the room reservoir. The hot qubit is then re-excited by the hot reservoir, preparing to remove any excitation in the

cold qubit coming from the cold reservoir. The interaction in Eq. (282) is most effective when it is in *resonance*, i.e., when the states $|001\rangle$ and $|110\rangle$ have the same energy. We thus demand

$$\varepsilon_{\mathrm{r}} = \varepsilon_{\mathrm{c}} + \varepsilon_{\mathrm{h}}. \tag{284}$$

We note that the interaction in Eq. (282) would be unphysical for fermions, where it would imply that one fermion is turned into two fermions.

We describe the coupling to the environment using the master equation in the singular-coupling limit

$$\partial_t \hat{\rho}_{\mathrm{S}}(t) = -i[\hat{H}_{\mathrm{S}}, \hat{\rho}_{\mathrm{S}}(t)] + \mathcal{L}_{\mathrm{c}}\hat{\rho}_{\mathrm{S}}(t) + \mathcal{L}_{\mathrm{h}}\hat{\rho}_{\mathrm{S}}(t) + \mathcal{L}_{\mathrm{r}}\hat{\rho}_{\mathrm{S}}(t), \tag{285}$$

with the dissipators

$$\mathcal{L}_\alpha = \tilde{\kappa}_\alpha n_{\mathrm{B}}^\alpha \mathcal{D}[\hat{\sigma}_\alpha^\dagger] + \tilde{\kappa}_\alpha(n_{\mathrm{B}}^\alpha + 1)\mathcal{D}[\hat{\sigma}_\alpha] = \kappa_\alpha n_{\mathrm{F}}^\alpha \mathcal{D}[\hat{\sigma}_\alpha^\dagger] + \kappa_\alpha(1 - n_{\mathrm{F}}^\alpha)\mathcal{D}[\hat{\sigma}_\alpha], \tag{286}$$

where the Bose-Einstein occupation and the Fermi-Dirac occupation are given by

$$n_{\mathrm{B}}^\alpha = \frac{1}{e^{\varepsilon_\alpha/k_{\mathrm{B}}T_\alpha} - 1} \qquad n_{\mathrm{F}}^\alpha = \frac{1}{e^{\varepsilon_\alpha/k_{\mathrm{B}}T_\alpha} + 1}. \tag{287}$$

This master equation can be derived using reservoirs described by noninteracting bosons (with $\mu = 0$)

$$\hat{H}_\alpha = \sum_q \varepsilon_{\alpha,q} \hat{a}_{\alpha,q}^\dagger \hat{a}_{\alpha,q}, \qquad \hat{V}_\alpha = \sum_k g_{\alpha,q}\left(\hat{a}_{\alpha,q}^\dagger \hat{\sigma}_\alpha + \hat{\sigma}_\alpha^\dagger \hat{a}_{\alpha,q}\right). \tag{288}$$

This naturally results in the dissipators given by the first equality in Eq. (286), including the Bose-Einstein occupations. In the second equality in Eq. (286), the dissipators are written in terms of the Fermi-Dirac occupations by using

$$\kappa_\alpha = \tilde{\kappa}_\alpha \frac{n_{\mathrm{B}}^\alpha}{n_{\mathrm{F}}^\alpha}, \qquad \frac{n_{\mathrm{B}}^\alpha + 1}{n_{\mathrm{B}}^\alpha} = \frac{1 - n_{\mathrm{F}}^\alpha}{n_{\mathrm{F}}^\alpha} = e^{\varepsilon_\alpha/k_{\mathrm{B}}T_\alpha}. \tag{289}$$

The last equality ensures that the ratio between the rates of absorbing and emitting energy is given by the Boltzmann factor. This is known as *local detailed balance* and it ensures that the equilibrium state is a Gibb's state. Note that while $\tilde{\kappa}_\alpha$ is determined by the reservoir's spectral density and is temperature independent, $\kappa_\alpha$ depends on temperature $T_\alpha$.

**Heat currents:**    From the master equation, we may extract the heat currents that enter the different reservoirs. As discussed in Sec. 4.4.2, the singular-coupling limit requires the use of a thermodynamic Hamiltonian for a consistent thermodynamic bookkeeping. The correct choice for the thermodynamic Hamiltonian depends on the details of the Hamiltonian. Given the resonance condition in Eq. (284), the appropriate thermodynamic Hamiltonian is given by $\hat{H}_{\mathrm{TD}} = \hat{H}_0$, i.e., we neglect the interaction between the qubits in the thermodynamic book-keeping. The heat currents then read

$$J_\alpha = \mathrm{Tr}\left\{\hat{H}_0 \mathcal{L}_\alpha \hat{\rho}_{\mathcal{S}}\right\} = -\varepsilon_\alpha \kappa_\alpha \left(n_{\mathrm{F}}^\alpha - \langle \hat{\sigma}_\alpha^\dagger \hat{\sigma}_\alpha \rangle\right). \tag{290}$$

Similarly to what we found for the systems based on quantum dots, the sign of the heat currents are determined by which occupation is larger, the occupation of the reservoir $n_{\mathrm{F}}^\alpha$ or the occupation of the qubit $\langle \hat{\sigma}_\alpha^\dagger \hat{\sigma}_\alpha \rangle$.

To make progress, it is instructive to consider the time-evolution of the qubit occupation

$$\partial_t \langle \hat{\sigma}_\alpha^\dagger \hat{\sigma}_\alpha \rangle = \mathrm{Tr}\left\{\hat{\sigma}_\alpha^\dagger \hat{\sigma}_\alpha \partial_t \hat{\rho}_{\mathrm{S}}(t)\right\} = -i\langle [\hat{\sigma}_\alpha^\dagger \hat{\sigma}_\alpha, \hat{H}_{\mathrm{int}}]\rangle + \kappa_\alpha\left(n_{\mathrm{F}}^\alpha - \langle \hat{\sigma}_\alpha^\dagger \hat{\sigma}_\alpha \rangle\right). \tag{291}$$

The second term on the right-hand side can be identified as $-J_\alpha/\varepsilon_\alpha$ by comparing to Eq. (290). The first term can be related to the following quantity

$$I \equiv 2g \, \text{Im} \langle \hat{\sigma}_r^\dagger \hat{\sigma}_c \hat{\sigma}_h \rangle = i \langle [\hat{\sigma}_c^\dagger \hat{\sigma}_c, \hat{H}_{\text{int}}] \rangle = i \langle [\hat{\sigma}_h^\dagger \hat{\sigma}_h, \hat{H}_{\text{int}}] \rangle = -i \langle [\hat{\sigma}_r^\dagger \hat{\sigma}_r, \hat{H}_{\text{int}}] \rangle. \qquad (292)$$

In the steady state, we have $\partial_t \langle \hat{\sigma}_\alpha^\dagger \hat{\sigma}_\alpha \rangle = 0$. From Eqs. (291) and (292), we may then infer

$$J_c = -\varepsilon_c I, \qquad J_h = -\varepsilon_h I, \qquad J_r = \varepsilon_r I. \qquad (293)$$

This remarkably simple relation between the heat currents arises due to the simple structure of the Hamiltonian. Heat may only traverse the system by exchanging an excitation in the room qubit with two excitations, one in the cold and one in the hot qubit.

**The laws of thermodynamics:** In the steady state, Eq. (292) allows us to express the laws of thermodynamics in a particularly simple form. The first law of thermodynamics may be written as

$$J_c + J_h + J_r = I(\varepsilon_r - \varepsilon_c - \varepsilon_h) = 0, \qquad (294)$$

which holds because of the resonance condition in Eq. (284). Obviously, the first law still holds if this resonance condition is not met. However, in that case the thermodynamic Hamiltonian should be chosen differently which would modify Eq. (290).

The second law of thermodynamics reads

$$\frac{J_c}{T_c} + \frac{J_h}{T_h} + \frac{J_r}{T_r} = I \left( \frac{\varepsilon_c + \varepsilon_h}{T_r} - \frac{\varepsilon_c}{T_c} - \frac{\varepsilon_h}{T_h} \right) \geq 0. \qquad (295)$$

In contrast to the first law, we may not verify the second law without explicitly solving the master equation. However, knowing that it holds, the second law can tell us when the device operates as a refrigerator. From Eq. (292), we see that the cold reservoir is cooled when $I \geq 0$. From the second law, we can infer that this is the case if the term in brackets in Eq. (295) is positive. This results in the condition for cooling

$$\frac{\varepsilon_c}{\varepsilon_h} \leq \frac{T_c(T_h - T_r)}{T_h(T_r - T_c)} \equiv \eta_C^{\text{COP}}, \qquad (296)$$

where $\eta_C^{\text{COP}}$ denotes the Carnot value for the coefficient of performance of an absorption refrigerator, see also the next subsection.

### 5.3.2 Figures of merit

As discussed above, we may pursue two distinct goals with the absorption refrigerator: Cooling the cold reservoir (goal 1), or cooling the qubit coupled to the cold reservoir (goal 2). For these two goals, we introduce different figures of merit.

**Goal 1:** For this goal, the desired output is a large heat current out of the cold reservoir. The natural figure of merit is then the coefficient of performance (COP)

$$\eta^{\text{COP}} \equiv \frac{-J_c}{-J_h} = \frac{\varepsilon_c}{\varepsilon_h} \leq \eta_C^{\text{COP}}. \qquad (297)$$

As for the heat engine in Sec. 5.1, the coefficient of performance can be understood as the desired output divided by the consumed resource, which in this case is the heat provided by the hot reservoir. For the absorption refrigerator under consideration, the COP is determined only by the qubit energy splittings, due to Eq. (292). As shown above, the second law of thermodynamics provides an upper bound on the COP, under the assumption that the device indeed operates as a refrigerator. This bound diverges when $T_r \to T_c$, as heat can in principle be moved in between two reservoirs of the same temperature without consuming any resources.

**Goal 2:**   When cooling the qubit coupled to the cold reservoir, the natural figure of merit is its occupation. It may be quantified by an effective temperature

$$\theta = \frac{\varepsilon_{\text{c}}}{k_{\text{B}} \ln\left(\frac{1+\langle \hat{\sigma}_{\text{c}}^{\dagger}\hat{\sigma}_{\text{c}}\rangle}{\langle \hat{\sigma}_{\text{c}}^{\dagger}\hat{\sigma}_{\text{c}}\rangle}\right)} \qquad \Leftrightarrow \qquad \langle \hat{\sigma}_{\text{c}}^{\dagger}\hat{\sigma}_{\text{c}}\rangle = \frac{1}{e^{\frac{\varepsilon_{\text{c}}}{k_{\text{B}}\theta}}+1}. \tag{298}$$

From Eq. (291), we find (in the steady state)

$$\langle \hat{\sigma}_{\text{c}}^{\dagger}\hat{\sigma}_{\text{c}}\rangle = n_{\text{F}}^{\text{c}} - I/\kappa_{\text{c}} > 0, \tag{299}$$

which depends on the quantity $I$ encountered before. The occupation may asymptotically reach zero in the limit

$$\frac{g}{\kappa_{\text{r}}} \to 0, \qquad \frac{\kappa_{\text{r}}}{\kappa_{\text{c}}} = \frac{\kappa_{\text{h}}}{\kappa_{\text{c}}} \to \infty, \qquad \frac{\varepsilon_{\text{h}}}{k_{\text{B}}T_{\text{h}}} \to 0, \qquad \frac{\varepsilon_{\text{r}}}{k_{\text{B}}T_{\text{r}}} \to \infty. \tag{300}$$

These conditions are very demanding and can obviously never be reached exactly. This can be understood as an implication of the third law of thermodynamics, which prevents cooling a system down to the ground state.

### 5.3.3   Perturbation theory

As we have seen above, the quantity $I$ determines most of the observables that we are interested in. In the model we consider, $I$ can be calculate analytically [this is how the limit in Eq. (300) was obtained]. However, this is a rather tedious calculation. Here we consider a perturbative calculation, which is simpler but nevertheless provides some insight. To this end, we consider the interacting part of the Hamiltonian $\hat{H}_{\text{int}}$ [c.f. Eq. (282)] as a small perturbation to the dynamics. This is justified, as long as $g \ll \kappa_{\text{c}} + \kappa_{\text{h}} + \kappa_{\text{r}}$. We then write

$$\hat{\rho}_{\text{S}} = \hat{\rho}_{\text{S}}^{(0)} + \hat{\rho}_{\text{S}}^{(1)} + \mathcal{O}(g^2), \tag{301}$$

where $\hat{\rho}_{\text{S}}^{(j)}$ is proportional to $g^j$. We note that the Hamiltonian is already written as the sum of a $g$ independent part, $\hat{H}_0$, and the interaction Hamiltonian which is linear in $g$. We then write the master equation in Eq. (285) order by order in $g$, starting with the zeroth order

$$\partial_t \hat{\rho}_{\text{S}}^{(0)}(t) = -i[\hat{H}_0, \hat{\rho}_{\text{S}}^{(0)}(t)] + \mathcal{L}_{\text{c}}\hat{\rho}_{\text{S}}^{(0)}(t) + \mathcal{L}_{\text{h}}\hat{\rho}_{\text{S}}^{(0)}(t) + \mathcal{L}_{\text{r}}\hat{\rho}_{\text{S}}^{(0)}(t). \tag{302}$$

This master equation describes three independent qubits, each coupled to its respective thermal reservoir. The steady state is thus simply a product of three Gibbs states (at different temperatures) and can be written as

$$\hat{\rho}^{(0)} = \begin{pmatrix} 1-n_{\text{F}}^{\text{c}} & 0 \\ 0 & n_{\text{F}}^{\text{c}} \end{pmatrix} \otimes \begin{pmatrix} 1-n_{\text{F}}^{\text{h}} & 0 \\ 0 & n_{\text{F}}^{\text{h}} \end{pmatrix} \otimes \begin{pmatrix} 1-n_{\text{F}}^{\text{r}} & 0 \\ 0 & n_{\text{F}}^{\text{r}} \end{pmatrix}. \tag{303}$$

The master equation to first order may be written as

$$\partial_t \hat{\rho}_{\text{S}}^{(1)}(t) = -i[\hat{H}_{\text{int}}, \hat{\rho}_{\text{S}}^{(0)}(t)] - i[\hat{H}_0, \hat{\rho}_{\text{S}}^{(1)}(t)] + \mathcal{L}_{\text{c}}\hat{\rho}_{\text{S}}^{(1)}(t) + \mathcal{L}_{\text{h}}\hat{\rho}_{\text{S}}^{(1)}(t) + \mathcal{L}_{\text{r}}\hat{\rho}_{\text{S}}^{(1)}(t). \tag{304}$$

The first term on the right-hand side gives

$$[\hat{H}_{\text{int}}, \hat{\rho}_{\text{S}}^{(0)}(t)] = g\,\delta n\,(|001\rangle\langle 110| - |110\rangle\langle 001|), \tag{305}$$

where we abbreviated

$$\delta n = n_{\text{F}}^{\text{c}} n_{\text{F}}^{\text{h}}(1-n_{\text{F}}^{\text{r}}) - (1-n_{\text{F}}^{\text{c}})(1-n_{\text{F}}^{\text{h}})n_{\text{F}}^{\text{r}}. \tag{306}$$

Note that $\delta n$ provides the population of the state $|110\rangle$ minus the population of the state $|001\rangle$ at $g = 0$. The term in Eq. (305) acts like a source term in the differential equation in Eq. (304). This suggests the ansatz

$$\hat{\rho}_{\rm S}^{(1)} = x\,|001\rangle\langle 110| + x^*\,|110\rangle\langle 001|\,. \tag{307}$$

This ansatz is Hermitian and trace-less, which is important to keep the density matrix in Eq. (301) Hermitian and with trace one. With this ansatz, we find

$$[\hat{H}_0, \hat{\rho}_{\rm S}^{(1)}] = (\varepsilon_{\rm r} - \varepsilon_{\rm c} - \varepsilon_{\rm h})x\,|001\rangle\langle 110| - (\varepsilon_{\rm r} - \varepsilon_{\rm c} - \varepsilon_{\rm h})x^*\,|110\rangle\langle 001|\,, \tag{308}$$

and

$$\mathcal{L}_j \hat{\rho}_{\rm S}^{(1)} = -\frac{\kappa_j}{2}\hat{\rho}_{\rm S}^{(1)}\,. \tag{309}$$

These identities can be inserted into Eq. (304) to obtain

$$\hat{\rho}_{\rm S}^{(1)} = -\frac{2ig\,\delta n}{\kappa_{\rm c} + \kappa_{\rm h} + \kappa_{\rm r}}(|001\rangle\langle 110| - |110\rangle\langle 001|)\,, \tag{310}$$

which allows us to determine

$$I = 2g\,{\rm Im}\langle\hat{\sigma}_{\rm r}^\dagger\hat{\sigma}_{\rm c}\hat{\sigma}_{\rm h}\rangle = 2g\,{\rm Im}\,\langle 110|\,\hat{\rho}_{\rm S}\,|001\rangle = \frac{4g^2\delta n}{\kappa_{\rm c} + \kappa_{\rm h} + \kappa_{\rm r}}\,. \tag{311}$$

From this expression, we find that for small $g$, the heat currents are quadratic in $g$. Furthermore, we obtain cooling whenever $\delta n > 0$. One may show that this inequality is equivalent to $\varepsilon_{\rm c}/\varepsilon_{\rm h} < \eta_{\rm C}^{\rm COP}$. This perturbative calculation thus reproduces the cooling condition for all values of $g$.

### 5.3.4 Coherence-enhanced cooling

Finally, we consider a feature of the absorption refrigerator that exploits quantum coherence in the transient dynamics. To this end, we assume that we have a means to turn the refrigerator on and off. In an implementation based on a superconducting circuit, this can for instance be achieved using a magnetic field [94]. When the refrigerator is off, the qubits will tend to the state given in Eq. (303), i.e., each qubit will thermalize with its respective reservoir. When the refrigerator is turned on, the effective temperature $\theta$ defined in Eq. (298) will start to oscillate, see Fig. 16. These oscillations will eventually damp out towards the steady-state value. Interestingly however, $\theta$ may dip below its steady-state value. If the refrigerator is switched off when $\theta$ reaches its first minimum, it can stay below its steady-state value for a significant amount of time if the couplings to the environment, $\kappa_j$, are sufficiently small. Quantum coherence thus allows for better cooling. This effect was first discussed in Ref. [95] and recently verified experimentally [92].

The damped oscillations in Fig. 16 are a consequence of the interplay between the unitary and the dissipative part in the master equation. The unitary part results in oscillations between the states $|110\rangle$ and $|001\rangle$, while the dissipative part induces an exponential time-dependence towards the steady state. To get some insight into the oscillations, we isolate the effect of the Hamiltonian. To this end, we consider the von Neumann equation

$$\begin{aligned}\partial_t \hat{\rho}_{\rm S}(t) &= -[\hat{H}_{\rm int}, \hat{\rho}_{\rm S}(t)]\\ &= -ig\,x(t)(|001\rangle\langle 110| - |110\rangle\langle 001|) + g\,y(t)(|110\rangle\langle 110| - |001\rangle\langle 001|)\,,\end{aligned} \tag{312}$$

where

$$x(t) = \langle 110|\,\hat{\rho}_{\rm S}(t)\,|110\rangle\,, \qquad\qquad y(t) = 2\,{\rm Im}\,\langle 110|\,\hat{\rho}_{\rm S}(t)\,|001\rangle\,. \tag{313}$$

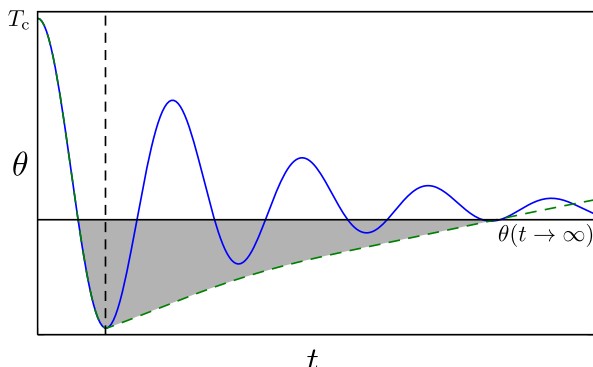

Figure 16: Coherence-enhanced cooling. The temperature in resonator c, $\theta$, is shown as a function of time. At $t = 0$, the refrigerator is switched on. The temperature then oscillates before it reaches its steady-state value (blue, solid line). Switching the refrigerator off at the time the temperature is at its first minimum (dashed vertical line) allows for $\theta$ to remain below the steady-state value for a substantial amount of time (green, dashed line) as illustrated by the grey shading.

In Eq. (312), we dropped $\hat{H}_0$ because it commutes both with $\hat{H}_{\text{int}}$ as well as with the initial state, which we choose to be given by Eq. (303). Equation (312) can be cast into the coupled differential equations

$$\partial_t x(t) = -2g\, y(t), \qquad\qquad \partial_t y(t) = 2g\, x(t), \qquad\qquad (314)$$

with the initial conditions $x(t = 0) = \delta n$ and $y(t = 0) = 0$. These equations may easily be solved, resulting in

$$x(t) = \delta n \cos(2g\, t), \qquad\qquad y(t) = \delta n \sin(2g\, t). \qquad\qquad (315)$$

The solution to Eq. (312) then reads

$$
\begin{aligned}
\hat{\rho}_S(t) = \hat{\rho}_S(t = 0) &- \delta n \sin^2(g\, t)(|110\rangle\langle110| - |001\rangle\langle001|) \\
&- \frac{i\delta n}{2}\sin(2g\, t)(|001\rangle\langle110| - |110\rangle\langle001|).
\end{aligned}
\qquad (316)
$$

These oscillations are known as (generalized) Rabi-oscillations and they appear whenever two states are coherently coupled to each other. From the quantum state, we may easily obtain the time-dependent occupation of the cold qubit

$$\langle\hat{\sigma}_c^\dagger\hat{\sigma}_c\rangle = n_F^c - \delta n \sin^2(g\, t). \qquad\qquad (317)$$

We note that $\delta n \leq n_F^c$, ensuring that the population remains positive at all times.

# 6   Fluctuations

In this section, we are exploring a key difference between macroscopic systems and small quantum systems: fluctuations. While in macroscopic systems, fluctuations can generally be neglected because they are negligible compared to mean values (c.f. Sec. 2.2), fluctuations do play an important role in small, nano-scale systems. The laws of thermodynamics derived

in Sec. 3 only include average values and they hold irrespective of the relevance of fluctuations. In this section, we discuss new laws that describe the behavior of heat and work in the presence of fluctuations. These new laws include fluctuation theorems [6, 96, 97] and thermodynamic uncertainty relations [98]. While fluctuation theorems generalize the second law of thermodynamics to fluctuating systems, thermodynamic uncertainty relations provide a trade-off between a signal-to-noise ratio and entropy production, and they do not have any counterpart in macroscopic systems.

## 6.1 Fluctuation theorems for an isolated system

Before we consider the general scenario introduced in Sec. 3.1, we focus on an isolated system that undergoes unitary time-evolution governed by a time-dependent Hamiltonian $\hat{H}_S(t)$. We consider an initial state that is diagonal in the instantaneous energy eigenbasis, i.e.,

$$\hat{\rho}_S(0) = \sum_n p_n |E_n^0\rangle\langle E_n^0|, \qquad\qquad \hat{H}_S(t) = \sum_n E_n^t |E_n^t\rangle\langle E_n^t|. \tag{318}$$

In this case, the time-evolution operator is given by

$$\hat{U}_S(t) = \mathcal{T}e^{-i\int_0^t dt' \hat{H}_S(t')}. \tag{319}$$

Since the system is isolated, no heat is exchanged with the environment. However, due to the time-dependence of the Hamiltonian, work will be performed on the system. The average work performed on the system after time $\tau$ reads (see also Sec. 3.3)

$$W_S = \int_0^\tau dt\, \mathrm{Tr}\left\{\left[\partial_t \hat{H}_S(t)\right]\hat{\rho}_S(t)\right\}. \tag{320}$$

### 6.1.1 The two-point measurement scheme

In order to go beyond average values, we introduce measurements according to the so-called *two-point measurement scheme* [99]:

1. Projectively measure the system energy at $t = 0$. This results in an outcome $E_n^0$.

2. Let the system evolve for time $\tau$ according to the time-evolution in Eq. (319).

3. Projectively measure the system energy at $t = \tau$. This results in an outcome $E_m^\tau$.

The outcomes of the measurements define a *trajectory* as the system evolved from state $|E_n^0\rangle \to |E_m^\tau\rangle$. The change in energy for such a trajectory is given by $W_{m\leftarrow n} = E_m^\tau - E_n^0$. Since there is no heat exchange, we interpret this energy change as work. The probability for observing a trajectory corresponds to the joint probability of measuring the initial and final energies and it reads

$$p(m \leftarrow n) = p_n p_{m|n} = p_n |\langle E_m^\tau | \hat{U}_S(\tau) | E_n^0\rangle|^2, \tag{321}$$

where $p_n$ is the initial probability of finding the system in state $|E_n^0\rangle$ and $p_{m|n}$ denotes the conditional probability of finding the system in state $|E_m^\tau\rangle$ at time $\tau$ given that it started out in state $|E_n^0\rangle$ at $t = 0$.

The average energy change may then be evaluated as

$$\langle W_{m\leftarrow n}\rangle = \sum_{n,m} p(m \leftarrow n) W_{m\leftarrow n} = \mathrm{Tr}\left\{\sum_m E_m^\tau |E_m^\tau\rangle\langle E_m^\tau| \hat{U}_S(\tau) \sum_n p_n |E_n^0\rangle\langle E_n^0| \hat{U}_S^\dagger(\tau)\right\}$$

$$- \mathrm{Tr}\left\{\hat{U}_S(\tau) \sum_n p_n E_n^0 |E_n^0\rangle\langle E_n^0| \hat{U}_S^\dagger(\tau)\right\} \tag{322}$$

$$= \mathrm{Tr}\left\{\hat{H}_S(\tau)\hat{\rho}_S(\tau)\right\} - \mathrm{Tr}\left\{\hat{H}_S(0)\hat{\rho}_S(0)\right\} = \int_0^\tau dt\, \partial_t \mathrm{Tr}\left\{\hat{H}_S(t)\hat{\rho}_S(t)\right\} = W_S.$$

This further justifies the interpretation of $W_{m \leftarrow n}$ as the work performed on the system during the trajectory $|E_n^0\rangle \rightarrow |E_m^\tau\rangle$, i.e., in a single experimental run when $E_n^0$ and $E_m^\tau$ are measured. Note that we use the same brackets to denote averages over trajectories $\langle f(n, m) \rangle = \sum_{n,m} p(m \leftarrow n) f(n, m)$ as we do for ensemble averages $\langle \hat{O} \rangle = \text{Tr}\{\hat{O}\hat{\rho}\}$. The object that is averaged over usually makes it clear which average is meant.

Having defined the work along trajectories allows us to investigate the fluctuations in the performed work. For instance, we may be interested in the higher moments of work defined as

$$\langle W_{m \leftarrow n}^k \rangle = \sum_{n,m} p(m \leftarrow n) W_{m \leftarrow n}^k. \tag{323}$$

### 6.1.2 The backward experiment

In fluctuation theorems, time-reversal plays an important role. For this reason, we introduce the time-reversal operator $\hat{\Theta}$ [6, 100]. Time-reversal in quantum mechanics is described by an *anti*-unitary operator. If we define $|\tilde{a}\rangle = \hat{\Theta} |a\rangle$ and $|\tilde{b}\rangle = \hat{\Theta} |b\rangle$, the defining properties of anti-unitarity are

$$\langle \tilde{b}|\tilde{a} \rangle = \langle b|a \rangle^* = \langle a|b \rangle, \qquad\qquad \hat{\Theta} (\alpha |\psi\rangle + \beta |\phi\rangle) = \alpha^* \hat{\Theta} |\psi\rangle + \beta^* \hat{\Theta} |\phi\rangle. \tag{324}$$

Note that these relations imply that the time-reversal operator anti-commutes with the imaginary unit: $\hat{\Theta} i = -i \hat{\Theta}$. Being anti-unitary, some rules that we take for granted do not apply to the time-reversal operator. For instance, the cyclic invariance of the trace has to be adapted as

$$\text{Tr}\{\hat{\Theta}\hat{A}\hat{\Theta}^{-1}\} = \text{Tr}\{\hat{A}\}^*. \tag{325}$$

Another useful identity we will employ reads

$$\langle \tilde{a}|\hat{A}|\tilde{b} \rangle = \langle b|\Theta^{-1}\hat{A}^\dagger \hat{\Theta}|a \rangle. \tag{326}$$

We note that in some works, a daggered time-reversal operator appears. Here we follow Refs. [6, 100] and avoid such a notation, using instead the inverse of the time-reversal operator $\hat{\Theta}^{-1}$. Furthermore, we always consider the time-reversal operator (and its inverse) to act on the right.

With the help of the time-reversal operator, we define a backward experiment. In this experiment, the time-evolution is determined by the time-reversed Hamiltonian

$$\tilde{H}_S(t) = \hat{\Theta}\hat{H}_S(\tau - t)\hat{\Theta}^{-1}, \qquad\qquad \tilde{U}_S(t) = \mathcal{T} e^{-i \int_0^t dt' \tilde{H}_S(t')}. \tag{327}$$

The work protocol discussed above, where time-evolution is governed by $\hat{H}_S(t)$ is denoted as the *forward experiment*. In addition to transforming the Hamiltonian by the time-reversal operator, the time argument is inverted in Eq. (327). Any external parameters that are changed during the forward experiment thus have their time-dependence reversed in the backward experiment. For instance, if an electric field is ramped up in the forward experiment, it is ramped down in the backward experiment. A special role is played by external magnetic fields. In the presence of such fields we have $\tilde{H}_S(\vec{B}, t) = \hat{\Theta}\hat{H}_S(-\vec{B}, \tau - t)\hat{\Theta}^{-1}$. The reason for this is that if we included the charges that create the magnetic field in the description, the time-reversal operator would reverse the momenta of these charges which changes the sign of the resulting magnetic fields. The time-evolution along the forward and backward experiments are linked by the so-called microversibility condition

$$\hat{\Theta}\hat{U}_S(\tau)\hat{\Theta}^{-1} = \tilde{U}_S^{-1}(\tau) = \tilde{U}_S^\dagger(\tau), \tag{328}$$

which may be derived from Eq. (327) and the properties of the time-reversal operator. As an initial state for the backward experiment, we consider a state that is diagonal in the basis of the time-reversed Hamiltonian at $t = 0$

$$\tilde{\rho}_S(0) = \sum_m q_m \hat{\Theta}|E_m^\tau\rangle\langle E_m^\tau|\hat{\Theta}^{-1}, \qquad \tilde{H}_S(t) = \sum_m E_m^{\tau-t}\hat{\Theta}|E_m^{\tau-t}\rangle\langle E_m^{\tau-t}|\hat{\Theta}^{-1}. \qquad (329)$$

The backward experiment is then defined similarly to the forward experiment:

1. Projectively measure the system energy at $t = 0$. This results in an outcome $E_m^\tau$.

2. Let the system evolve for time $\tau$ according to the time-evolution in Eq. (327).

3. Projectively measure the system energy at $t = \tau$. This results in an outcome $E_n^0$.

The outcomes of the measurements again define a trajectory $|\tilde{E}_m^\tau\rangle = \hat{\Theta}|E_m^\tau\rangle \rightarrow \hat{\Theta}|E_n^0\rangle = |\tilde{E}_n^0\rangle$. The probability for this trajectory reads

$$\tilde{p}(n \leftarrow m) = q_m|\langle\tilde{E}_n^0|\tilde{U}_S(\tau)|\tilde{E}_m^\tau\rangle|^2 = q_m|\langle E_m^\tau|\Theta^{-1}\tilde{U}_S^\dagger(\tau)\hat{\Theta}|E_n^0\rangle|^2$$
$$= q_m|\langle E_m^\tau|\hat{U}_S(\tau)|E_n^0\rangle|^2, \qquad (330)$$

where we employed Eqs. (326) and (328).

### 6.1.3 Fluctuation theorems

We may now derive fluctuation theorems by taking the ratios of observing time-reversed trajectories in the forward and backward experiment

$$\frac{\tilde{p}(n \leftarrow m)}{p(m \leftarrow n)} = \frac{q_m}{p_n}. \qquad (331)$$

Due to microreversibility, the transition probabilities drop out and we are left with the ratio of initial probabilities. This innocuous expression is actually very powerful. To see this, let us consider thermal initial states

$$\hat{\rho}_S(0) = \frac{e^{-\beta\hat{H}_S(0)}}{Z_0}, \qquad \tilde{\rho}_S(0) = \hat{\Theta}\frac{e^{-\beta\hat{H}_S(\tau)}}{Z_\tau}\hat{\Theta}^{-1}, \qquad (332)$$

where the partition function reads

$$Z_t = \sum_n e^{-\beta E_n^t} = e^{-\beta F_t}, \qquad (333)$$

with $F_t$ denoting the free energy for a thermal state at inverse temperature $\beta$ and Hamiltonian $\hat{H}_S(t)$, see Eq. (102). For thermal states, the probabilities reduce to

$$p_n = e^{-\beta(E_n^0 - F_0)}, \qquad q_m = e^{-\beta(E_m^\tau - F_\tau)}. \qquad (334)$$

**Crooks fluctuation theorem:**  Plugging Eq. (334) into Eq. (331) results in

$$\frac{\tilde{p}(n \leftarrow m)}{p(m \leftarrow n)} = e^{-\beta(W_{m \leftarrow n} - \Delta F)}, \qquad (335)$$

with $\Delta F = F_\tau - F_0$. This relation is known as Crooks fluctuation theorem [101]. It is an instance of a *detailed* fluctuation theorem because it involves the probabilities for single trajectories.

**Jarzynski relation:** Multiplying Eq. (335) with $p(m \leftarrow n)$ and summing over all $n$ and $m$, we find the Jarzynski relation [102]

$$\sum_{n,m} \tilde{p}(n \leftarrow m) = \sum_{n,m} p(m \leftarrow n) e^{-\beta(W_{m\leftarrow n} - \Delta F)} \quad \Rightarrow \quad \left\langle e^{-\beta W_{m\leftarrow n}} \right\rangle = e^{-\beta \Delta F}. \tag{336}$$

This relation is known as an *integral* fluctuation theorem since it involves an average rather than individual trajectories. The Jarzynski relation is remarkable because it relates an equilibrium quantity, the difference in equilibrium free energies on the right hand side, to an out-of-equilibrium quantity, the work performed in the forward experiment that appears on the left-hand side. Importantly, there is no requirement of remaining in or close to equilibrium during the experiment. Since equilibrium free energies are difficult to measure, the Jarzynski relation has been used to determine free energy landscapes, in particular in single-molecule pulling experiments [103]. In practice, evaluating the average on the left-hand side may be challenging as trajectories with exponentially small probabilities may contribute [104].

Finally, we may apply Jensen's inequality to the Jarzynski relation. Jensen's inequality states that for a convex function $f(ax_1 + (1-a)x_2) \leq af(x_1) + (1-a)f(x_2)$, the inequality

$$\langle f(X) \rangle \geq f(\langle X \rangle), \tag{337}$$

holds for a random variable $X$. Using this inequality, we find

$$e^{-\beta \Delta F} = \left\langle e^{-\beta W_{m\leftarrow n}} \right\rangle \geq e^{-\beta \langle W_{m\leftarrow n} \rangle} \quad \Rightarrow \quad \langle W_{m\leftarrow n} \rangle = W_{\mathrm{S}} \geq \Delta F. \tag{338}$$

The final inequality states that the work performed on the system always exceeds the difference in equilibrium free energies. This inequality is equivalent to the second law of thermodynamics, when both the initial and final states are thermal states (which can always be achieved by letting the system equilibrate after the experiment). Thus, we find that the Jarzynski relation and the Crooks fluctuation theorem generalize the second law of thermodyanmics, as they constrain not only the average value of the performed work but also its fluctuations.

## 6.2 Fluctuation theorems for the general scenario

We now return to the general scenario introduced in Sec. 3.1. To define trajectories, we write the initial state [c.f. Eq. (122)] as

$$\hat{\rho}_{\mathrm{tot}}(0) = \hat{\rho}_{\mathrm{S}}(0) \bigotimes_\alpha \hat{\tau}_\alpha = \sum_{i,\vec{k}} p_0(i,\vec{k}) \left| \psi_i(0), \vec{k} \right\rangle\!\left\langle \psi_i(0), \vec{k} \right|, \tag{339}$$

with the states

$$\left| \psi_i(0), \vec{k} \right\rangle = \left| \psi_i(0) \right\rangle \bigotimes_\alpha \left| E^\alpha_{k_\alpha}, N^\alpha_{k_\alpha} \right\rangle, \tag{340}$$

where

$$\hat{H}_\alpha \left| E^\alpha_{k_\alpha}, N^\alpha_{k_\alpha} \right\rangle = E^\alpha_{k_\alpha} \left| E^\alpha_{k_\alpha}, N^\alpha_{k_\alpha} \right\rangle, \qquad \hat{N}_\alpha \left| E^\alpha_{k_\alpha}, N^\alpha_{k_\alpha} \right\rangle = N^\alpha_{k_\alpha} \left| E^\alpha_{k_\alpha}, N^\alpha_{k_\alpha} \right\rangle. \tag{341}$$

The sub- and superscripts may be confusing here. The eigenvalues of $\hat{H}_\alpha$ are labeled by $E^\alpha_j$ and the vector $\vec{k}$ has elements $k_\alpha$. The probabilities in Eq. (339) read

$$p_0(i,\vec{k}) = p_i(0) \prod_\alpha \frac{e^{-\beta_\alpha \left( E^\alpha_{k_\alpha} - \mu_\alpha N^\alpha_{k_\alpha} \right)}}{Z_\alpha}. \tag{342}$$

Finally, we introduced the eigenvalues and eigenstates of the reduced state of the system through

$$\hat{\rho}_{\mathrm{S}}(t) = \mathrm{Tr}_{\mathrm{B}} \{\hat{\rho}_{\mathrm{tot}}(t)\} = \sum_i p_i(t) \left| \psi_i(t) \right\rangle\!\left\langle \psi_i(t) \right|. \tag{343}$$

### 6.2.1   Forward trajectories

We now introduce trajectories by

1. Projectively measure all reservoir energies and particle numbers $\{\hat{H}_\alpha, \hat{N}_\alpha\}$ and the system in its eigenbasis $|\psi_i(0)\rangle$. This results in outcomes: $i, \vec{k}$.

2. Let the system evolve for time $\tau$ according to the time-evolution operator $\hat{U}(\tau)$ given in Eq. (110).

3. Projectively measure all reservoir energies and particle numbers $\{\hat{H}_\alpha, \hat{N}_\alpha\}$ and the system in its eigenbasis $|\psi_j(\tau)\rangle$. This results in outcomes: $j, \vec{l}$.

We denote the corresponding trajectory by $\gamma : \left|\psi_i(0), \vec{k}\right\rangle \rightarrow \left|\psi_j(\tau), \vec{l}\right\rangle$. The probability to observe such a trajectory is given by

$$P(\gamma) = p_0(i, \vec{k}) \left|\left\langle\psi_j(\tau), \vec{l}\right|\hat{U}(\tau)\left|\psi_i(0), \vec{k}\right\rangle\right|^2. \tag{344}$$

We may now define various thermodynamic quantities along these trajectories.

**Stochastic heat:**   The stochastic heat exchanged with reservoir $\alpha$ is defined as

$$Q_\alpha(\gamma) = \left(E^\alpha_{l_\alpha} - \mu_\alpha N^\alpha_{l_\alpha}\right) - \left(E^\alpha_{k_\alpha} - \mu_\alpha N^\alpha_{k_\alpha}\right). \tag{345}$$

The average of the stochastic heat reduces to

$$\langle Q_\alpha(\gamma)\rangle = \sum_\gamma P(\gamma) Q_\alpha(\gamma) = \mathrm{Tr}\left\{\left(\hat{H}_\alpha - \mu_\alpha \hat{N}_\alpha\right)\left[\hat{\rho}_{\mathrm{tot}}(\tau) - \hat{\rho}_{\mathrm{tot}}(0)\right]\right\} = Q_\alpha, \tag{346}$$

which is equal to the definition for heat introduced in Eq. (124). In Eq. (346), the sum over all trajectories is given by $\sum_\gamma = \sum_{i,j,\vec{k},\vec{l}}$ and the equality may be proven using the identities

$$\hat{H}_\alpha = \sum_{j,\vec{l}} E^\alpha_{l_\alpha} \left|\psi_j(\tau), \vec{l}\right\rangle\!\left\langle\psi_j(\tau), \vec{l}\right|, \quad \hat{N}_\alpha = \sum_{j,\vec{l}} N^\alpha_{l_\alpha} \left|\psi_j(\tau), \vec{l}\right\rangle\!\left\langle\psi_j(\tau), \vec{l}\right|,$$

$$\mathbb{1} = \sum_{j,\vec{l}} \left|\psi_j(\tau), \vec{l}\right\rangle\!\left\langle\psi_j(\tau), \vec{l}\right|. \tag{347}$$

**Stochastic (chemical) work:**   Analogously, the stochastic chemical work is defined as

$$W_\alpha(\gamma) = \mu_\alpha \left(N^\alpha_{l_\alpha} - N^\alpha_{k_\alpha}\right). \tag{348}$$

As the stochastic heat, its average reduces to the expected value

$$\langle W_\alpha(\gamma)\rangle = \mu_\alpha \mathrm{Tr}\left\{\hat{N}_\alpha\left[\hat{\rho}_{\mathrm{tot}}(\tau) - \hat{\rho}_{\mathrm{tot}}(0)\right]\right\}. \tag{349}$$

**Stochastic entropy:**   Following Ref. [105], we may use the self-information introduced in Sec. 1.3 to define a stochastic system entropy change

$$\Delta S(\gamma) = -\ln p_j(\tau) + \ln p_i(0), \tag{350}$$

which averages to the change in von Neumann entropy

$$\langle \Delta S(\gamma)\rangle = S_{\mathrm{vN}}[\hat{\rho}_S(\tau)] - S_{\mathrm{vN}}[\hat{\rho}_S(0)]. \tag{351}$$

**Stochastic entropy production:**   Finally, we can introduce the stochastic entropy production along a trajectory as

$$\Sigma(\gamma) = k_{\text{B}} \Delta S(\gamma) + \sum_\alpha \frac{Q_\alpha(\gamma)}{T_\alpha}, \tag{352}$$

which averages to the entropy production introduced in Eq. (123), $\langle \Sigma(\gamma) \rangle = \Sigma$ as can be easily shown using Eqs. (346) and (351). Importantly, while the average entropy production $\Sigma$ is always nonnegative, the stochastic entropy production $\Sigma(\gamma)$ can become negative.

We note that in contrast to Sec. 6.1, we do not define a stochastic version of the work associated to the time-dependence of the Hamiltonian. The reason for this is that if the initial state does not commute with the Hamiltonian, there is no unique way to define stochastic work that obeys all desired properties [106, 107].

### 6.2.2   Backward trajectories

To derive fluctuation theorems, we need to introduce backward trajectories. For these, we consider the initial state

$$\tilde{\rho}_{\text{tot}}(0) = \hat{\Theta} \hat{\rho}_{\text{S}}(\tau) \bigotimes_\alpha \hat{\tau}_\alpha \hat{\Theta}^{-1} = \sum_{j,\vec{l}} p_\tau(j, \vec{l}) \hat{\Theta} \left| \psi_j(\tau), \vec{l} \right\rangle \left\langle \psi_j(\tau), \vec{l} \right| \hat{\Theta}^{-1}, \tag{353}$$

where $p_\tau(j, \vec{l})$ is obtained from Eq. (342) by replacing $p_j(0) \to p_j(\tau)$, i.e.,

$$p_\tau(j, \vec{l}) = p_j(\tau) \prod_\alpha \frac{e^{-\beta_\alpha \left( E_{l_\alpha}^\alpha - \mu_\alpha N_{l_\alpha}^\alpha \right)}}{Z_\alpha}. \tag{354}$$

Note that the initial state of the backward trajectory is quite different from what we considered in Sec. 6.1. For the system state, we consider the (time-reversed) final state of the forward experiment. The reservoirs on the other hand are initialized in the same Gibbs states as in the beginning of the forward experiment. This is consistent with the paradigm that we have control over the system, but not over the environment which we only describe by constant temperatures and chemical potentials.

This state is then evolved by the operator

$$\tilde{U}(t) = \mathcal{T} e^{-i \int_0^t dt' \hat{\Theta} \hat{H}_{\text{tot}}(t') \hat{\Theta}^{-1}}, \tag{355}$$

which obeys the micro-reversibility condition

$$\hat{\Theta} \hat{U}(\tau) \hat{\Theta}^{-1} = \tilde{U}^\dagger(\tau). \tag{356}$$

The backward trajectories are now defined by

1. Projectively measure all reservoir energies and particle numbers $\{\hat{H}_\alpha, \hat{N}_\alpha\}$ and the system in its eigenbasis $\hat{\Theta} | \psi_j(\tau) \rangle$. This results in outcomes: $j, \vec{l}$.

2. Let the system evolve for time $\tau$ according to the time-evolution operator $\tilde{U}(\tau)$ given in Eq. (355).

3. Projectively measure all reservoir energies and particle numbers $\{\hat{H}_\alpha, \hat{N}_\alpha\}$ and the system in its eigenbasis $\hat{\Theta} | \psi_i(0) \rangle$. This results in outcomes: $i, \vec{k}$.

We denote the corresponding trajectory by $\tilde{\gamma} : \hat{\Theta} \left| \psi_j(\tau), \vec{l} \right\rangle \to \hat{\Theta} \left| \psi_i(0), \vec{k} \right\rangle$. The probability to observe such a trajectory is given by, using Eq. (326)

$$\begin{aligned}
\tilde{P}(\tilde{\gamma}) &= p_\tau(j, \vec{l}) \left| \left\langle \psi_j(\tau), \vec{l} \right| \hat{\Theta}^{-1} \tilde{U}^\dagger(\tau) \hat{\Theta} \left| \psi_i(0), \vec{k} \right\rangle \right|^2 \\
&= p_\tau(j, \vec{l}) \left| \left\langle \psi_j(\tau), \vec{l} \right| \hat{U}(\tau) \left| \psi_i(0), \vec{k} \right\rangle \right|^2.
\end{aligned} \tag{357}$$

### 6.2.3 Fluctuation theorems

A detailed fluctuation theorem may now be obtained by taking the ratio of probabilities for time-reversed trajectories in the forward and in the backward experiment

$$\frac{\tilde{P}(\tilde{\gamma})}{P(\gamma)} = \frac{p_\tau(j, \vec{l})}{p_0(i, \vec{k})} = e^{-\sum_\alpha \frac{Q_\alpha(\gamma)}{T_\alpha} + \ln \frac{p_j(\tau)}{p_i(0)}} = e^{-\Sigma(\gamma)/k_B}. \tag{358}$$

As for the isolated system, c.f. Eq. (335), the transition probabilities drop out and the right-hand side is determined solely by the initial probabilities. The Boltzmann factors in Eq. (342) result in the stochastic heat, c.f. (345), and together with the initial probabilities of the system, the exponent adds up to the stochastic entropy production, c.f. Eq. (352).

Equation (358) provides a generalization of the second law of thermodynamics to nano-scale systems, where fluctuations matter. It is instructive, to consider what happens if Eq. (358) is applied to a macroscopic process, for instance a glass shattering. In such a process, that we denote by the trajectory $\gamma$, roughly $10^{23}$ degrees of freedom are involved (Avogadro constant), resulting in an entropy production of the order of $10^{23} k_B$. The probability to observe the time-reversed process $\tilde{\gamma}$ is thus exponentially suppressed by an extremely large number. It is thus safe to assume that macroscopic processes with negative entropy production do not happen as their probabilities are virtually zero. For a nano-scale object, the entropy production along a trajectory may be as low as a few $k_B$, and indeed we may observe the time-reversed trajectory.

This has interesting consequences for the thermodynamic arrow of time. For macroscopic systems, such an arrow of time is given by the fact that entropy is produced. For nano-scale systems, fluctuations prevent us from identifying the direction of time with certainty: if you see a video of a fluctuating system, you may not with certainty identify if the video was played forward or backward. Recently, machine learning was employed to learn the arrow of time and the algorithm rediscovered the fluctuation theorem as the underlying thermodynamic principle [108].

From the detailed fluctuation theorem, we may derive the Crooks fluctuation theorem by summing over all trajectories that have the same entropy production

$$\frac{\tilde{P}(-\Sigma)}{P(\Sigma)} = e^{-\Sigma/k_B}, \quad P(\Sigma) = \sum_\gamma \delta(\Sigma - \Sigma(\gamma)) P(\gamma), \quad \tilde{P}(\Sigma) = \sum_\gamma \delta(\Sigma - \Sigma(\gamma)) \tilde{P}(\gamma). \tag{359}$$

Furthermore, an integral fluctuation theorem can be derived

$$\left\langle e^{-\Sigma(\gamma)} \right\rangle = 1, \tag{360}$$

from which the second law, $\langle \Sigma(\gamma) \rangle \geq 0$, follows through Jensen's inequality in complete analogy to Eqs. (336) and (338). This implies that the fluctuation theorem in Eq. (358) contains strictly more information than the second law, and therefore can be understood as a generalization thereof.

## 6.3 Full counting statistics

In the last subsection, we defined fluctuations along single trajectories for the general scenario. Here we show how we can extract these fluctuations from an approximate description based on Markovian master equations. To this end, we consider the method of full counting statistics (FCS) which allows us to calculate probability distributions for heat and work, as well as their associated moments and cumulants. A more detailed introduction can be found in Refs. [18, 54].

### 6.3.1 Counting particles

We start by considering a system that can exchange particles with a fermionic reservoir as illustrated in Fig. 17. In addition, it might be coupled to other reservoirs. We are interested in the net number of particles $n$ that entered the fermionic reservoir after time $t$. Whenever a fermion enters the reservoir, $n$ is increased by one. When a fermion leaves the reservoir (enters the system), $n$ is reduced by one. The quantity $n$ is thus a stochastic variable that is different in each experimental run. We consider the master equation

$$\partial_t \hat{\rho}_S(t) = \tilde{\mathcal{L}} \hat{\rho}_S(t) + \kappa(1 - n_F)\mathcal{D}[\hat{c}]\hat{\rho}_S(t) + \kappa n_F \mathcal{D}[\hat{c}^\dagger]\hat{\rho}_S(t) \equiv \mathcal{L}\hat{\rho}_S(t), \tag{361}$$

where $\tilde{\mathcal{L}}$ denotes the time-evolution due to the Hamiltonian as well as all reservoirs except the one we are interested in and the annihilation operator $\hat{c}$ destroys a fermion in the system. Now we identify which terms in the master equation change the values of $n$. To this end, we write out the superoperators

$$\mathcal{D}[\hat{c}^\dagger]\hat{\rho} = \hat{c}^\dagger \hat{\rho} \hat{c} - \frac{1}{2}\{\hat{c}\hat{c}^\dagger, \hat{\rho}\}, \qquad \mathcal{D}[\hat{c}]\hat{\rho} = \hat{c}\hat{\rho}\hat{c}^\dagger - \frac{1}{2}\{\hat{c}^\dagger \hat{c}, \hat{\rho}\}. \tag{362}$$

The term $\hat{c}^\dagger \hat{\rho} \hat{c}$ increases the number of fermions in the system by one. This fermion comes from the reservoir, thus this term decreases $n$ by one. Similarly, the term $\hat{c}\hat{\rho}\hat{c}^\dagger$ increases $n$ by one.

We now introduce the $n$-resolved density matrix $\hat{\rho}_S(n)$ as the density matrix if $n$ fermions entered the reservoir. From this quantity, we may recover the full density matrix as

$$\hat{\rho}_S = \sum_{n=-\infty}^{\infty} \hat{\rho}_S(n). \tag{363}$$

Having identified the terms in the master equation which change $n$, we may write the time-evolution of the $n$-resolved density matrix as

$$\partial_t \hat{\rho}_S(n) = \mathcal{L}_0 \hat{\rho}_S(n) + \mathcal{J}_+ \hat{\rho}_S(n-1) + \mathcal{J}_- \hat{\rho}_S(n+1), \tag{364}$$

where the superoperators that increase and decrease $n$ are written as

$$\mathcal{J}_+ \hat{\rho} = \kappa(1 - n_F)\hat{c}\hat{\rho}\hat{c}^\dagger, \qquad \mathcal{J}_- \hat{\rho} = \kappa n_F \hat{c}^\dagger \hat{\rho}\hat{c}, \tag{365}$$

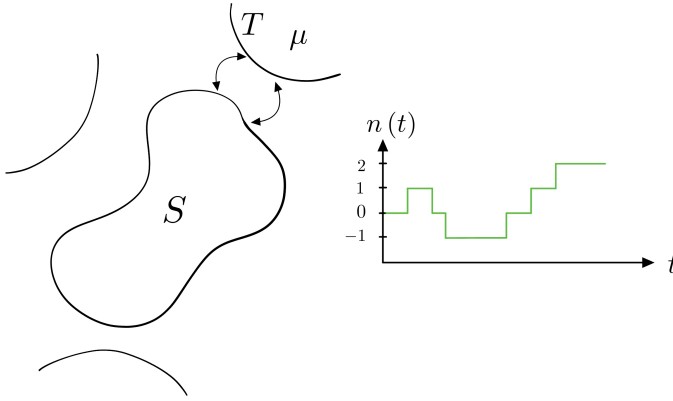

Figure 17: Illustration of full counting statistics. A quantum system can exchange particles with a reservoir. The net number of particles that enter the reservoir during time $t$ is denoted by $n(t)$. Whenever a particle enters the reservoir, $n$ is increased by one. Whenever a particle leaves the reservoir (and enters the system) $n$ is decreased by one.

and $\mathcal{L}_0$ denotes the time-evolution that does not change $n$ such that $\mathcal{L} = \mathcal{L}_0 + \mathcal{J}_+ + \mathcal{J}_-$. Summing Eq. (363) over $n$, we thus recover the master equation in Eq. (361).

From the $n$-resolved density matrix, we can obtain the probability that $n$ fermions entered the reservoir during time $t$ as

$$p(n) = \text{Tr}\{\hat{\rho}_S(n)\}. \tag{366}$$

Often we are interested in the moments of this distribution

$$\langle n^k \rangle = \sum_{n=-\infty}^{\infty} n^k p(n), \tag{367}$$

with the first moment $\langle n \rangle$ being the average.

Keeping the information on $n$ comes at a price. Comparing Eqs. (364) to Eq. (361), we turned a single master equation into infinitely many coupled master equations, one for each $\hat{\rho}_S(n)$. This problem can be simplified by Fourier transforming

$$\hat{\rho}_S(\chi) = \sum_{n=-\infty}^{\infty} e^{in\chi} \hat{\rho}_S(n), \tag{368}$$

where $\chi$ is known as the *counting field*. Fourier transforming Eq. (364), we find

$$\partial_t \hat{\rho}_S(\chi) = \mathcal{L}_0 \sum_{n=-\infty}^{\infty} e^{in\chi} \hat{\rho}_S(n) + \mathcal{J}_+ \sum_{n=-\infty}^{\infty} e^{in\chi} \hat{\rho}_S(n-1) + \mathcal{J}_- \sum_{n=-\infty}^{\infty} e^{in\chi} \hat{\rho}_S(n+1)$$
$$= \mathcal{L}_0 \hat{\rho}_S(\chi) + e^{i\chi} \mathcal{J}_+ \hat{\rho}_S(\chi) + e^{-i\chi} \mathcal{J}_- \hat{\rho}_S(\chi) \equiv \mathcal{L}(\chi) \hat{\rho}_S(\chi). \tag{369}$$

This equation has the formal solution

$$\hat{\rho}_S(\chi) = e^{\mathcal{L}(\chi)t} \hat{\rho}_S(\chi, t = 0). \tag{370}$$

To find the initial condition, we note that we start counting at $t = 0$, implying that $n = 0$ at $t = 0$ and therefore

$$\hat{\rho}_S(n, t = 0) = \hat{\rho}_S(t = 0)\delta_{n,0} \quad \Leftrightarrow \quad \hat{\rho}_S(\chi, t = 0) = \hat{\rho}_S(t = 0). \tag{371}$$

The trace of the $\chi$-dependent density matrix provides the characteristic function

$$\Lambda(\chi) \equiv \sum_{n=-\infty}^{\infty} e^{in\chi} p(n) = \text{Tr}\{\hat{\rho}_S(\chi)\}, \qquad \Lambda(0) = \sum_{n=-\infty}^{\infty} p(n) = 1. \tag{372}$$

In the field of FCS, the characteristic function is often called the moment-generating function since the moments of $n$ can be obtained by taking derivatives as

$$\langle n^k \rangle = (-i)^k \partial_\chi^k \Lambda(\chi)|_{\chi=0}. \tag{373}$$

Note however that in probability theory, the moment-generating function is the Laplace transform of the probability distribution. A particularly useful quantity is the cumulant-generating function, given by the logarithm of the characteristic function

$$S(\chi) = \ln \Lambda(\chi), \qquad \langle\!\langle n^k \rangle\!\rangle = (-i)^k \partial_\chi^k S(\chi)|_{\chi=0}, \tag{374}$$

which produces the cumulants $\langle\!\langle n^k \rangle\!\rangle$ upon differentiation. The first cumulant is equal to the mean

$$\langle\!\langle n \rangle\!\rangle = -i\partial_\chi S(\chi)|_{\chi=0} = -i \left.\frac{\partial_\chi \Lambda(\chi)}{\Lambda(\chi)}\right|_{\chi=0} = \langle n \rangle. \tag{375}$$

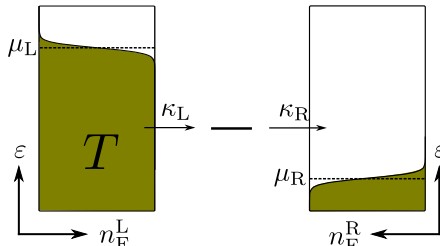

Figure 18: Quantum dot in the high bias regime. Due to a large voltage bias particles can only enter the dot from the left reservoir and leave to the right reservoir. We are interested in the net number of particles that enter the right reservoir.

The second cumulant is equal to the variance

$$\langle\langle n^2 \rangle\rangle = -\partial_\chi^2 S(\chi)|_{\chi=0} = \frac{-\partial_\chi^2 \Lambda(\chi)}{\Lambda(\chi)}\bigg|_{\chi=0} - \frac{\left[-i\partial_\chi \Lambda(\chi)\right]^2}{\Lambda^2(\chi)}\bigg|_{\chi=0} = \langle n^2 \rangle - \langle n \rangle^2. \qquad (376)$$

The third cumulant $\langle\langle n^3 \rangle\rangle$ is related to the skewness of the distribution and the fourth cumulant to its kurtosis ("tailedness").

Often, we are interested in the long-time behavior of $n$. In this case, it is useful to use the spectral decomposition to write

$$e^{\mathcal{L}(\chi)t} = \sum_j e^{\nu_j t} \mathcal{P}_j, \qquad (377)$$

where $\nu_j$ denote the eigenvalues of the Liouvillean $\mathcal{L}$ and the projectors $\mathcal{P}_j$ can be obtained from its eigenvectors. The characteristic function may then be written as

$$\Lambda(\chi) = \sum_j e^{\nu_j t} \text{Tr}\{\mathcal{P}_j \hat{\rho}_S(t=0)\} \xrightarrow{t\to\infty} e^{\nu_{\max} t} \text{Tr}\{\mathcal{P}_{\max} \hat{\rho}_S(t=0)\}, \qquad (378)$$

where $\nu_{\max}$ is the $\chi$-dependent eigenvalue of the Liouvillean with the largest real part. Similarly, the cumulant generating function may be written as

$$S(\chi) \xrightarrow{t\to\infty} \nu_{\max} t + \ln \text{Tr}\{\mathcal{P}_{\max} \hat{\rho}_S(t=0)\} \simeq \nu_{\max} t. \qquad (379)$$

The cumulant generating function is thus fully determined by the eigenvalue of the Liouvillean with the largest real part. Furthermore, all cumulants become linear in time, since $S(\chi) \propto t$.

### 6.3.2 Example: transport through a quantum dot

As an example, we return to the quantum dot coupled to two fermionic reservoirs, see Fig. 18, as discussed in Sec. 5.1 in the context of a heat engine. Here we consider the so-called large bias regime for simplicity, where $\mu_L - \epsilon_d, \epsilon_d - \mu_R \gg k_B T$. In this case, $n_F^L \simeq 1$ and $n_F^R \simeq 0$, such that particles can only enter the dot from the left reservoir and leave to the right reservoir. We want to describe the statistics of the net number of particles that enter the right reservoir. This system was experimentally investigated in Ref. [109], where the tunneling electrons could be observed and counted one by one.

The master equation for this scenario is given by

$$\partial_t \hat{\rho}_S = \kappa_L \mathcal{D}[\hat{d}^\dagger]\hat{\rho}_S + \kappa_R \mathcal{D}[\hat{d}]\hat{\rho}_S = \mathcal{L}\hat{\rho}_S. \qquad (380)$$

Since we only count the particles entering the right reservoir, we may introduce the counting field by the replacement

$$\kappa_R \hat{d} \hat{\rho}_S \hat{d}^\dagger \quad \rightarrow \quad e^{i\chi} \kappa_R \hat{d} \hat{\rho}_S \hat{d}^\dagger. \qquad (381)$$

This results in the counting-field dependent master equation

$$\partial_t \hat{\rho}_S(\chi) = \mathcal{L}\hat{\rho}_S(\chi) + \kappa_R \left(e^{i\chi} - 1\right)\hat{d}\hat{\rho}_S\hat{d}^\dagger. \tag{382}$$

Noting that the density matrix remains diagonal, and denoting the diagonal elements by $p_0(\chi)$ and $p_1(\chi)$, we may cast this master equation into

$$\partial_t \begin{pmatrix} p_0(\chi) \\ p_1(\chi) \end{pmatrix} = \underbrace{\begin{pmatrix} -\kappa_L & \kappa_R e^{i\chi} \\ \kappa_L & -\kappa_R \end{pmatrix}}_{L(\chi)} \begin{pmatrix} p_0(\chi) \\ p_1(\chi) \end{pmatrix}, \tag{383}$$

where $L(\chi)$ denotes the Liouvillean (acting on a vector instead of on a matrix). Its eigenvalues can easily be calculated and read

$$\nu_\pm = -\frac{\kappa_L + \kappa_R}{2} \pm \frac{1}{2}\sqrt{(\kappa_L - \kappa_R)^2 + 4e^{i\chi}\kappa_L\kappa_R}. \tag{384}$$

The cumulant generating function in the long-time limit is thus [c.f. Eq. (379)]

$$S(\chi)/t = \nu_+ = -\frac{\kappa_L + \kappa_R}{2} + \frac{1}{2}\sqrt{(\kappa_L - \kappa_R)^2 + 4e^{i\chi}\kappa_L\kappa_R}, \tag{385}$$

resulting in the average

$$\langle n \rangle = -i\partial_\chi S(\chi)|_{\chi=0} = \frac{\kappa_L\kappa_R}{\kappa_L + \kappa_R}t = \langle I \rangle t, \tag{386}$$

where $\langle I \rangle$ denotes the average particle current, see also the power given in Eq. (243). The variance is given by

$$\langle\!\langle n^2 \rangle\!\rangle = -\partial_\chi^2 S(\chi)|_{\chi=0} = \langle n \rangle \frac{\kappa_L^2 + \kappa_R^2}{\kappa_L + \kappa_R^2}. \tag{387}$$

Transport becomes particularly simple if one of the couplings far exceeds the other, for instance if $\kappa_L \gg \kappa_R$. In this case, the cumulant generating function reduces to

$$S(\chi) = \kappa_R t \left(e^{i\chi} - 1\right). \tag{388}$$

Note that it is the smaller of the couplings that features in the cumulant generating function. This is the case because transport at long times is dominated by bottlenecks. In this case, the bottleneck is the smaller coupling to the right reservoir. The cumulants obtained from Eq. (388) are all equal and read $\langle\!\langle n^k \rangle\!\rangle = \kappa_R t$. This is the hallmark of Poissonian transport, where each particle is independent of the others. Indeed, we may write the characteristic function as

$$\Lambda(\chi) = e^{-\kappa_R t} \exp\left(\kappa_R t e^{i\chi}\right) = e^{-\kappa_R t} \sum_{n=0}^{\infty} \frac{(\kappa_R t)^n}{n!} e^{in\chi}, \tag{389}$$

from which we may read off, with the help of Eq. (372),

$$p(n) = \frac{(\kappa_R t)^n}{n!} e^{-\kappa_R t}, \tag{390}$$

which is indeed a Poisson distribution.

### 6.3.3 Counting heat and work

So far, we focused on counting particles. We now illustrate how the same technique can be applied to evaluate the moments and cumulants of heat and power exchanged with the environment. To this end, we consider the singular-coupling master equation given in Eq. (194) which is here restated for convenience

$$\partial_t \hat{\rho}_S(t) = -i[\hat{H}_S, \hat{\rho}_S(t)] + \sum_{\alpha,k} \gamma_k^\alpha(\omega_{\alpha,k}) \mathcal{D}[\hat{S}_{\alpha,k}]\hat{\rho}_S(t) = \mathcal{L}\hat{\rho}_S(t), \tag{391}$$

where the Hamiltonian may include a Lamb shift. The jump operators obey

$$[\hat{S}_{\alpha,k}, \hat{H}_{TD}] = \omega_{\alpha,k}\hat{S}_{\alpha,k}, \qquad\qquad [\hat{S}_{\alpha,k}, \hat{N}_S] = n_{\alpha,k}\hat{S}_{\alpha,k}, \tag{392}$$

where the thermodynamic Hamiltonian is introduced in Sec. 4.4.2. From these relations, we may conclude that a jump $\hat{S}_{\alpha,k}\hat{\rho}_S\hat{S}_{\alpha,k}^\dagger$ reduces the particle number in the system by $n_{\alpha,k}$ and the internal energy by $\omega_{\alpha,k}$.

We are interested in the joint distribution of the heat and work exchanged with the different reservoirs $P(\{Q_\alpha, W_\alpha\})$. This distribution can be obtained from a characteristic function

$$\Lambda(\{\chi_\alpha, \lambda_\alpha\}) = \left(\prod_\alpha \int dQ_\alpha e^{iQ_\alpha\chi_\alpha} \int dW_\alpha e^{iW_\alpha\lambda_\alpha}\right) P(\{Q_\alpha, W_\alpha\}) = \text{Tr}\{\hat{\rho}_S(\{\chi_\alpha, \lambda_\alpha\})\}. \tag{393}$$

The counting field dependent density matrix can again be obtained by a modified master equation

$$\partial_t \hat{\rho}_S(\{\chi_\alpha, \lambda_\alpha\}) = \mathcal{L}(\{\chi_\alpha, \lambda_\alpha\})\hat{\rho}_S(\{\chi_\alpha, \lambda_\alpha\}), \tag{394}$$

where the counting field dependent Liouvillean is obtained from the Liouvillean $\mathcal{L}$ in Eq. (391) by the replacement

$$\hat{S}_{\alpha,k}\hat{\rho}_S\hat{S}_{\alpha,k}^\dagger \quad \rightarrow \quad e^{i\chi_\alpha(\omega_{\alpha,k} - \mu_\alpha n_{\alpha,k})} e^{i\lambda_\alpha \mu_\alpha n_{\alpha,k}} \hat{S}_{\alpha,k}\hat{\rho}_S\hat{S}_{\alpha,k}^\dagger. \tag{395}$$

In contrast to Eq. (369), the counting fields are weighted by the heat and work associated to each jump. While Eq. (395) provides a simple recipe, we note that we can rigorously obtain $\mathcal{L}(\{\chi_\alpha, \lambda_\alpha\})$ by introducing counting fields on the unitary evolution with $\hat{H}_{tot}$ and subsequently tracing out the environment using Born-Markov approximations [64].

We also note that the fluctuation theorem in Eq. (358) may be derived from the master equation using the method of FCS. A trajectory $\gamma$ is then determined by outcomes of initial and final measurements on the system, as well as the times and types of all jumps that occur [18, 97].

## 6.4 The Thermodynamic uncertainty relation

The thermodynamic uncertainty relation (TUR) is an inequality that bounds fluctuations in a current by the entropy production [98, 110]

$$\frac{\langle\!\langle I^2 \rangle\!\rangle}{\langle I \rangle^2} \geq \frac{2k_B}{\dot{\Sigma}}. \tag{396}$$

Here $\langle\!\langle I^2 \rangle\!\rangle$ and $\langle I \rangle$ denote the steady-state current fluctuations and average current respectively (see below), and $\dot{\Sigma}$ denotes the entropy production rate.

The TUR can be understood as an upper bound on the signal-to-noise ratio $\langle I \rangle^2 / \langle\!\langle I^2 \rangle\!\rangle$, provided by dissipation, as expressed through the entropy production rate: to achieve a high signal-to-noise ratio, a sufficient amount of entropy must be produced. Equivalently, the TUR

implies that it is not possible to simultaneously achieve a high current, low fluctuations, and low dissipation.

In contrast to fluctuation theorems, the TUR has a smaller range of validity. It applies to classical, Markovian systems. It may therefore be violated in quantum-coherent systems, see for instance Refs. [70, 111–114]. Markovian quantum systems thus have the potential to achieve higher signal-to-noise ratios for a given amount of dissipation than their classical counterparts. We note however that the TUR may also be violated in classical, non-Markovian systems [115].

### 6.4.1 Current and current noise

The average and the noise of a current enter the TUR. Here we briefly connect these quantities to the cumulants obtained from FCS as discussed in Sec. 6.3. Let $n$ count the number of transferred *quanta* (e.g., particles or photons exchanged with a reservoir). The current cumulants are then defined as the time-derivatives of the cumulants of $n$

$$\langle\!\langle I^k \rangle\!\rangle = \partial_t \langle\!\langle n^k \rangle\!\rangle \xrightarrow{t \to \infty} \frac{\langle\!\langle n^k \rangle\!\rangle}{t}, \tag{397}$$

where we used that all cumulants become linear in time in the long-time limit. The first cumulant of the current is simply the average current $\langle I \rangle$, while the second cumulant characterizes the noise of the current.

The relation between current and the counting variable $n$ can be motivated by introducing a stochastic current, which is the time-derivative of $n$, i.e.,

$$n(t) = \int_0^t dt' I(t'). \tag{398}$$

Since $n$ typically increases or decreases by one if a particle is exchanged, the stochastic current $I(t)$ consists of a series of Dirac deltas at these times. With this definition, one may show

$$\langle n \rangle = \int_0^t dt' \langle I(t) \rangle, \qquad \lim_{t \to \infty} \langle\!\langle n \rangle\!\rangle = t \int_{-\infty}^{\infty} d\tau \langle \delta I(\tau) \delta I(0) \rangle = t S_{\delta I}, \tag{399}$$

where $\delta I(t) = I(t) - \langle I(t) \rangle$ denotes the deviation of the current from its mean and $S_{\delta I}$ is known as the zero-frequency power spectrum. More information on noise and fluctuations can be found for instance in Refs. [18, 116].

### 6.4.2 Application: heat engine

As an illustrative example, let us consider the TUR applied to a heat engine operating between a hot and a cold reservoir, as discussed in Sec. 5.1. In this case, Eq. (396) may be re-written as [117]

$$P \frac{\eta}{\eta - \eta_C} \frac{k_B T_c}{\langle\!\langle P^2 \rangle\!\rangle} \leq \frac{1}{2}, \tag{400}$$

where $P$ denotes the average output power, $\eta$ the efficiency and $\eta_C$ the Carnot efficiency [c.f. Eq. (242)]. The TUR thus implies that we cannot at the same time achieve high power, an efficiency close to the Carnot efficiency, and low fluctuations in the output power (sometimes called high *constancy*). In other words, a high output power at efficiencies close to the Carnot efficiency is only possible if fluctuations are large.

Equation (400) may be obtained from Eq. (396) by using the expression for the entropy production rate in Eq. (246) together with the identities

$$\langle I \rangle = \frac{P}{\mu_c - \mu_h}, \qquad \langle\!\langle I \rangle\!\rangle = \frac{\langle\!\langle I \rangle\!\rangle}{(\mu_c - \mu_h)^2}, \tag{401}$$

which relate the power and its fluctuations to the particle current. The TUR thus implies that the well-known trade-off between high power and an efficiency close to the Carnot efficiency is actually a trade-off between three quantities, including the power fluctuations. This sheds light onto previous works that found finite power at Carnot efficiency close to a phase transition, where fluctuations diverge [118, 119].

Interesting additional applications of the TUR include the estimation of efficiencies of molecular motors [120], as well as investigating the precision of biological clocks [121].

## 7 Summary

These lecture notes provide an introduction to the vast and growing field of quantum thermodynamics. In particular, the following key questions were addressed:

**What is thermodynamic equilibrium?** This question is addressed in Sec. 2, where we introduced the Gibbs state as well as the concepts of temperature and chemical potential. To highlight the relevance of the Gibbs state, we provided three motivations: The maximum entropy principle, considering a small part of a system with well defined energy and particle number, as well as the concept of global passivity.

**How do the laws of thermodynamics emerge from quantum mechanics?** In Sec. 3, we tackled this key question that lies at the heart of quantum thermodynamics and we introduced the general scenario that was used throughout the lecture notes. We discussed how energy conservation results in the first law of thermodynamics, we introduced the concept of entropy and entropy production, and we discussed how the second law of thermodynamics can be understood in terms of loss of information.

**How can we model open quantum systems?** In Sec. 4 we derived Markovian master equations that provide a powerful tool to produce approximations for observables in the general scenario. We discussed how these equations can be derived in a thermodynamically consistent way and highlighted their limitations.

**What can we do with open quantum systems?** Section 5 introduced three different quantum thermal machines, a heat engine, an entanglement generator, and a refrigerator. These devices illustrate how open quantum systems coupled to multiple thermal reservoirs can be used to perform useful tasks.

**How do small systems differ from large ones?** In Sec. 6, we focused on a key difference between microscopic and macroscopic systems: fluctuations. We illustrated how they become relevant and how they result in new thermodynamic rules including fluctuation theorems, which generalize the second law of thermodynamics, and the thermodynamic uncertainty relation.

## 8 Outlook

While a number of topics in the field of quantum thermodynamics is addressed in these lecture notes, there is much more that has not been touched upon. Here I provide a subjective selection of exciting research avenues that build on the concepts discussed in these notes.

## 8.1   The thermodynamics of Information

Already in the 19th century, J. C. Maxwell realized, by considering a thought experiment, that having access to microscopic degrees of freedom allows for performing tasks that are otherwise forbidden by the second law of thermodynamics [122]. Maxwell considered a "neat-fingered being", commonly referred to as Maxwell's demon, that can see individual particles in a gas. Using a trap-door, the demon could separate fast from slow particles and thereby create a temperature gradient out of equilibrium without performing any work, seemingly reducing entropy and thus violating the second law of thermodynamics. More generally, using measurement and feedback control, thermal fluctuations can be rectified which may result in negative entropy production. This seeming contradiction of the laws of thermodynamics is resolved by appreciating that the demon itself is a physical system. In order to function, it needs to produce entropy. Including this contribution to the entropy production, the laws of thermodynamics are restored. Experimental progress has enabled the implementation of Maxwell's thought experiment in the laboratory, both in classical [123,124], and more recently in quantum systems [125–127].

Interestingly, to understand the thermodynamics of feedback-controlled systems, it is not required to model the physical implementation of feedback control (i.e., the demon). Instead, it is sufficient to include the information obtained by the demon in the thermodynamic bookkeeping [12,128–130]. Since feedback-controlled systems exploit fluctuations (different measurement outcomes result in different feedback protocols), and since the fluctuation theorem provides a generalization of the second law of thermodynamics to fluctuating systems, information can be incorporated in the thermodynamic bookkeeping by accordingly modified fluctuation theorems [131–135]. A better understanding of the thermodynamics of measurement and feedback promises to shed light onto how such systems can be exploited for emerging nano- and quantum technologies, similar to how a better understanding of thermodynamics allowed for the refinement of steam engines and refrigerators in the industrial revolution.

## 8.2   Quantum thermo-kinetic uncertainty relation

The TUR discussed in Sec. 6.4 shows how entropy production bounds the signal-to-noise ratio of any current. In addition to the TUR, another bound exists known as the kinetic uncertainty relation (KUR) [136]. The KUR looks very similar to the TUR but instead of the entropy production rate, the average rate of transitions between different states, the dynamical activity enters. While the TUR is tight close to equilibrium (a consequence of the fluctuation-dissipation theorem), it becomes very losoe far from equilibrium because the entropy production rate increases in this regime. In contrast, the KUR provides a relevant bound far from equilibrium and can also be applied to irreversible dynamics, such as predator-prey models. Very recently, a tighter version of the KUR was discovered which is known as the clock uncertainty relation (CUR), due to its connection to the problem of time estimation [137].

Both the TUR as well as the KUR hold for classical Markovian systems and can be violated in systems described by a quantum Markovian master equation (see for instance Ref. [70]). This has motivated efforts in finding similar bounds that constrain signal-to-noise ratios in quantum systems. Over the last few years, a number of extensions of the TUR and KUR to the quantum regime where discovered, see for instance Refs. [138–142]. These works provide insights into the fascinating avenue of research that aims at understanding the fundamental limitations of precision in open quantum systems, as well as the differences between quantum stochastic systems.

## 8.3 Finite-size environments

In quantum thermodynamics, the environment is often treated as large and memoryless, being well described by fixed temperatures and chemical potentials. This is however not always the case. Indeed in recent experiments, temperature fluctuations were measured [143], and a single two-level system was used to purify the state of its environment [144–146], resulting in enhanced coherence times. There is no unique approach to describe the thermodynamics of such scenarios. Indeed multiple approaches exist, including the extended micro-canonical master equation [147], where the energy of the environment is treated dynamically and classical correlations between system and environment are taken into account. Furthermore, recently a thermodynamic description of energy exchanges between arbitrary quantum systems was put forward [148].

In Ref. [149], a single-level quantum dot coupled to a finite-size reservoir that can be described by a fluctuating temperature was investigated. It was found that the thermodynamics of the quantum dot is described by an entropic temperature instead of the actual, fluctuating temperature. In particular, the quantum dot equilibrates to the entropic temperature, in analogy to the constant temperature case considered in Sec. 4.3.

There still remain many open questions related to finite-size reservoirs. For instance, when can they be described using a fluctuating temperature and when is a non-equilibrium description needed? This research direction is closely related to understanding systems that are strongly coupled to their environment [44–47].

## 8.4 Dissipative phase transitions

Phase transitions are a central concept in macroscopic thermodynamics. A phase transition denotes a drastic change in a systems' property upon changing an external variable. Common examples include the transition of liquid water to ice at zero degrees Celcius, or the onset of magnetization in a ferromagnet. These phase transitions occur as temperature is changed. In quantum systems, phase transitions may occur at zero temperature, as quantum fluctuations result in a drastic state of the ground state properties. These phase transitions are termed quantum phase transitions.

Phase transitions are heavily exploited in technological applications, including refrigerators and sensors, e.g. single photon detectors [150]. Indeed, phase transitions are very promising for sensing applications due to the drastic change that can be induced by a small perturbation.

More recently, dissipative phase transitions between out-of-equilibrium steady states have been investigated [151]. These transitions occurring far from equilibrium provide a rich behavior and can be exploited for sensing applications [152]. The thermodynamics of these transitions remains rather unexplored and provides a promising research avenue.

## 8.5 Thermodynamics of multiple conserved quantities

Throughout these lecture notes, we considered the grand canonical (and sometimes the canonical) ensemble. In this ensemble, there are two conserved quantities, energy and particle number. One can imagine adding additional conserved quantities, for instance angular momentum or any other observable that commutes with the total Hamiltonian. This motivates the extension of the thermal state to the so-called generalized Gibbs state [153]

$$\hat{\rho}_{\mathrm{G}} = \frac{e^{-\beta(\hat{H} - \sum_k \mu_k \hat{Q}_k)}}{Z}, \tag{402}$$

where $\hat{Q}_k$ denotes the conserved charges. In this case, thermodynamic forces corresponding to different conserved quantities can be can be traded off with each other, similar to how the heat

engine in Sec. 5.1 exploits a temperature gradient to overcome a voltage bias. In contrast to a heat engine, which performs one useful task, hybrid thermal machines perform multiple useful tasks at the same time [154]. For instance, a three-terminal device can be used to produce work and cool down the coldest reservoir at the same time.

Of particular interest is the study of thermodynamics when the conserved quantities no longer commute, as is the case for instance with the different components of angular momentum. In this case, established concepts in thermodynamics are challenged and many fascinating questions are raised [155].

# Acknowledgmenets

These lecture notes were created for a Master course at the University of Basel where they are accompanied by exercise sheets (available upon request). They are based on a previous set of notes prepared for an 8-hour course at Lund University. I thank all participants of these courses whose feedback has been crucial for the development of these notes. I also thank Peter Samuelsson for providing the opportunity to develop my own course at Lund University. Many thanks also to the teaching assistants in Basel including Marcelo Janovitch, Matteo Brunelli, Kacper Prech, and Aaron Daniel. Special thanks to Aaron Daniel for illustrating Figs. 1, 2, 3, 5, 6, 7, 15, and 17, and to Marcelo Janovitch for illustrating Fig. 4. Special thanks also to Aaron Daniel and Nicolas Hunziker for pointing out the appearance of the golden ratio in Sec. 5.2. I acknowledge funding from the Swiss National Science Foundation (Eccellenza Professorial Fellowship PCEFP2_194268).

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
