# Peer review of "Quantum Thermodynamics"

_SciPost Physics Lecture Notes_

## Round 1 · Referee Report · Anonymous (Referee 1) · 2024-8-13

Report

The manuscript titled "Quantum Thermodynamics" by Patrick P. Potts offers a comprehensive introduction to the thermodynamics of small quantum systems. It successfully bridges theoretical concepts with practical applications, making it a valuable resource for students and researchers in the field.
The manuscript provides a robust introduction to quantum thermodynamics. With minor revisions, particularly in the references and grammar, it will serve as an excellent educational resource. I recommend acceptance with minor revisions and publication in SciPost Physics Lecture Notes.

Requested changes

  1. It is recommended to provide a more detailed explanation of the derivation from Eq. (120) to (121), specifically addressing how the term Tr{H_tot(t)∂_t ρ_tot} vanishes. This explanation could include a discussion on the commutation relationship between the Hamiltonian and the density matrix and how equilibrium conditions within the grand canonical ensemble, as referenced in Eq. (72), imply that the time derivative of the density matrix vanishes. Incorporating this detailed explanation will help students understand the underlying physical principles and mathematical steps more effectively, improving their grasp of quantum thermodynamics.
  2. It is recommended that the explanation between Eq. (135) and Eq. (136) be expanded. The manuscript could greatly benefit from including intermediate steps or underlying assumptions that lead to this solution.
  3. It is recommended to explicitly state the assumption that the system and the bath are uncorrelated before introducing Eq.(138).

  4. To further enhance the understanding and application of superoperator and projection operator techniques within the lecture notes, it is recommended to include the following references in the bibliography: i) Mukamel, S. (1995). "Principles of Nonlinear Optical Spectroscopy," Oxford University Press. • This book provides an extensive overview of nonlinear optical spectroscopy and discusses how projection operator techniques can be utilized to understand the dynamics of quantum systems. It is a resource for theoretical foundations and practical applications relevant to the course material. ii) Levitov, L., Lee, H., and Lesovik, G. B. (1996). "Electron Counting Statistics and Coherent States of Electric Current," Journal of Mathematical Physics. • This article explores superoperators and projection operator techniques in analyzing electron transport statistics and quantum noise. It offers a detailed examination suitable for advanced studies and research in quantum mechanics and mesoscopic physics.

Including these references will provide students and readers with additional insights into the practical applications of the theoretical concepts discussed in the lecture notes. These resources are particularly valuable for deepening our understanding of quantum dynamical systems and the mathematical methods used to describe them.

Attachment

Recommendation

Publish (easily meets expectations and criteria for this Journal; among top 50%)

---

## Round 2 · Author Response

Dear Editor, Dear Referee,

Thank you for the time and effort spent on my manuscript. I am happy that the Referee recommends acceptance with minor revisions. In particular, the Referee requests four changes which are fully addressed in the updated manuscript. I believe that with these changes, the manuscript is ready for publication in SciPost Physics Lecture Notes.

Yours sincerely,
Patrick Potts

---

## Round 2 · List of Changes

All equation and reference numbers refer to the updated manuscript.

Requests by the Referee:

  1. It is recommended to provide a more detailed explanation of the derivation from Eq. (120) to (122), specifically addressing how the term Tr{H_tot(t)∂_t ρ_tot} vanishes. This explanation could include a discussion on the commutation relationship between the Hamiltonian and the density matrix and how equilibrium conditions within the grand canonical ensemble, as referenced in Eq. (72), imply that the time derivative of the density matrix vanishes. Incorporating this detailed explanation will help students understand the underlying physical principles and mathematical steps more effectively, improving their grasp of quantum thermodynamics.

    Response: I added the new Eq. (121) that clarifies why the corresponding term vanishes. However, this is a direct consequence of the unitary dynamics and is not related to any equilibrium conditions. No assumptions on the initial state are required for this.

  2. It is recommended that the explanation between Eq. (136) and Eq. (137) be expanded. The manuscript could greatly benefit from including intermediate steps or underlying assumptions that lead to this solution.

    Response: I added text below Eq. (138), stating how it can be verified that Eq. (137) is indeed a solution to the second equation in Eq. (136). No assumptions need to be made. This solution can be obtained using standard methods for solving differential equations, like variation of constants. I did not provide an explicit derivation since I think it might be distracting and since there is an abundance of pedagogical resources covering differential equations.

  3. It is recommended to explicitly state the assumption that the system and the bath are uncorrelated before introducing Eq.(139).

    Response: In the updated manuscript, this assumption is explicitly stated.

  4. To further enhance the understanding and application of superoperator and projection operator techniques within the lecture notes, it is recommended to include the following references in the bibliography: i) Mukamel, S. (1995). "Principles of Nonlinear Optical Spectroscopy," Oxford University Press. • This book provides an extensive overview of nonlinear optical spectroscopy and discusses how projection operator techniques can be utilized to understand the dynamics of quantum systems. It is a resource for theoretical foundations and practical applications relevant to the course material. ii) Levitov, L., Lee, H., and Lesovik, G. B. (1996). "Electron Counting Statistics and Coherent States of Electric Current," Journal of Mathematical Physics. • This article explores superoperators and projection operator techniques in analyzing electron transport statistics and quantum noise. It offers a detailed examination suitable for advanced studies and research in quantum mechanics and mesoscopic physics.

    Response: I included a reference to the book by S. Mukamel (Ref. [62] in the updated manuscript). I did not include a reference to "Electron Counting Statistics and Coherent States of Electric Current" since this paper, while being a seminal contribution, does not explore superoperators and projector operator techniques.

Small changes: I fixed a number of typos and made some minor modifications. In particular, I slightly changed the presentation around Eq. (313), to improve accessibility. I also added the sentence "Figure adapted from Ref. [98]" to Fig. 16, I added Refs. [75,76] and updated references to preprints that have been published since the appearance of the first version, and I updated the acknowledgments.

---

## Editorial Decision

in_refereeing